# Enhanced P-TEFb activity compromises dentate gyrus neurogenesis in mice

Yin Fang [1,3], Tong Qiu [1,3], Ping Wang [2,3], Shujun Bai [1], Min Wang [1], Chao Yang [1], Yan Wang [1], Peixuan Zhang [1], He Wang [2], Shanling Liu [2✉], Xue Xiao [1✉] & Qintong Li [1✉]

## Abstract

**Enhanced P-TEFb activity is thought to promote cell proliferation by increasing the transcriptional output of RNA polymerase II. The 7SK snRNP complex, which contains LARP7 and HEXIM1, sequesters and inhibits most cellular P-TEFb to prevent premature transcription elongation. Paradoxically, instead of exerting overgrowth effects, biallelic inactivation of *LARP7* is linked to Alazami syndrome, a human neurodevelopmental disorder characterized by growth restriction and cognitive impairment. Here, we report that conditional ablation of either *Larp7* or *Hexim1* in the murine brain reduces the size and impairs the function of the hippocampal dentate gyrus during the neonatal period. Functional analyses reveal that increased P-TEFb activity enhances self-renewal transcriptional programs in transit-amplifying neuronal progenitor cells to limit neurogenesis in developing dentate gyri. These results demonstrate that dysregulated subtissular stem cell dynamics can reconcile increased P-TEFb activity with reduced organ growth, and suggest a translational opportunity for repurposing P-TEFb inhibitors to treat medical conditions affecting dentate gyrus size and function.**

**Keywords** Alazami Syndrome; LARP7; Transcription Regulation; P-TEFb; Stem Cell
**Subject Categories** Chromatin, Transcription & Genomics; Development; Neuroscience

## Introduction

Precise regulation of gene expression is essential for normal mammalian development (Cardoso-Moreira et al, 2019), and its dysregulation is associated with human diseases (Unger Avila et al, 2024). In metazoans, the transcription of most protein-coding genes by RNA polymerase II is controlled at the step of transcription elongation (Chen et al, 2018). After transcription initiation, the movement of RNA polymerase II is paused at the promoter-proximal region by negative elongation factors, DRB sensitivity-inducing factor (DSIF) and negative elongation factor (NELF) (Guo and Price, 2013). The release of paused RNA polymerase II into productive transcription elongation requires the kinase activity of positive transcription elongation factor b (P-TEFb), composed of CDK9 catalytic subunit and cyclin T regulatory subunit (Marshall and Price, 1995). Transcription factors, such as c-MYC, can recruit P-TEFb to the promoter region to reverse the effect of DSIF and NELF, and to phosphorylate the carboxy-terminal domain (CTD) of the largest subunit of RNA polymerase II (Peterlin and Price, 2006). Because CTD phosphorylation serves as a loading pad for RNA processing factors in capping and splicing, P-TEFb plays a central role in the coordination of RNA transcription and co-transcriptional events (Carrocci and Neugebauer, 2024).

It has been known for more than two decades that most cellular P-TEFb is sequestered and inactivated by 7SK small nuclear ribonucleoprotein complex (snRNP) (Nguyen et al, 2001; Yang et al, 2001). However, the physiological and pathological implications of the regulation of P-TEFb by 7SK snRNP remain elusive. The inhibitory 7SK snRNP consists of 7SK snRNA, HEXIM1 or HEXIM2 (HEXIM1/2) (Byers et al, 2005; Michels et al, 2003; Yik et al, 2003), and LARP7 (Krueger et al, 2008; Markert et al, 2008). We and others have shown that 7SK snRNA first interacts with HEXIM1/2 and then recruit P-TEFb (Michels et al, 2004). In vitro, 7SK snRNA and HEXIM1/2 are sufficient to inhibit the kinase activity of P-TEFb (Byers et al, 2005; Li et al, 2005). LARP7 is dispensable for 7SK snRNA to interact with HEXIM1/2 or P-TEFb in vitro, but acts as a chaperon to stabilize and maintain 7SK snRNA level in cells (Krueger et al, 2008; Markert et al, 2008). Knockdown or knockout of LARP7 renders 7SK snRNA undetectable in cells, and thus cellular P-TEFb activities are elevated. Because P-TEFb is required to activate transcription of most protein-coding genes, it is generally assumed that enhanced P-TEFb activity would increase cellular proliferation and growth (Fujinaga et al, 2023), and this assumption motivates the pharmaceutical industry to develop P-TEFb inhibitors to treat cancer (Zhang et al, 2024).

However, there is no definitive experimental evidence supporting the causal relationship between cell proliferation and the

[1]Departments of Obstetrics & Gynecology and Pediatrics, Cancer Center, West China Second University Hospital, Key Laboratory of Birth Defects and Related Diseases of Women and Children, Ministry of Education, Development and Related Diseases of Women and Children Key Laboratory of Sichuan Province, Sichuan University, Chengdu, China. [2]Department of Medical Genetics, Prenatal Diagnostic Center, West China Second University Hospital, Sichuan University, Chengdu, China. [3]These authors contributed equally: Yin Fang, Tong Qiu, Ping Wang. ✉E-mail: liuxl@scu.edu.cn; xiaoxuela@scu.edu.cn; liqintong@scu.edu.cn

amount of active P-TEFb in cells, that is, P-TEFb free of 7SK snRNP. In all cell types examined so far, 50–90% of P-TEFb is sequestered and inactivated by 7SK snRNP. Even among the most rapidly proliferating cells, such as mouse embryonic stem cells, more than 80% of P-TEFb is associated with 7SK snRNP (Dai et al, 2014). In fact, knockdown or knockout of LARP7 does not alter the proliferative rate of mouse embryonic stem cells or human HAP1 cells (Dai et al, 2014; Studniarek et al, 2021). Adding further complications, human genetics studies have linked inactivating variants in *LARP7* to both hypo- and hyper-proliferative diseases. On the one hand, loss-of-function variants of *LARP7* are thought to cause Alazami syndrome (OMIM, #615071) (Alazami et al, 2012), a human neurodevelopmental disorder characterized by global growth restriction and intellectual disability (Al-Hinai et al, 2022; Amelie et al, 2025; Das et al, 2021; Dateki et al, 2018; Hollink et al, 2016; Imbert-Bouteille et al, 2019; Ling and Sorrentino, 2016; Soengas-Gonda et al, 2023; Thouqan et al, 2025; Wojcik et al, 2019). On the other hand, both *LARP7* downregulation (Cheng et al, 2012) and overexpression (Zheng and Pan, 2025) are associated with tumorigenesis in human patients. However, the role of *LARP7* in neither neurodevelopmental disorder nor tumorigenesis has been substantiated by animal models.

Several adverse conditions at all stages of human life are believed to reduce the size of the dentate gyrus and impair cognitive functioning. To name a few, these include intrauterine growth restriction during embryogenesis (Gilchrist et al, 2018), child abuse (Teicher et al, 2012), post-traumatic stress disorder (Gilbertson et al, 2002) and aging (Fotuhi et al, 2012). The dentate gyrus is a subfield of the hippocampus involved in learning and memory (Borzello et al, 2023), and one of the few locations capable of generating new neurons throughout adulthood (Denoth-Lippuner and Jessberger, 2021). The hippocampus is more vulnerable to neurotoxic insults than other regions of the brain (Davidson and Stevenson, 2024). The most abundant cell type within the dentate gyrus is granule cells, thus the quantity of granule cells is a major determinant of the size of the dentate gyrus (Amaral et al, 2007). In mice, most pallial structures originate from Emx1+ neural stem and progenitor cells during embryogenesis (Gorski et al, 2002). Between embryonic day (E) 13.5 and E18.5, a small subset of Emx1+ stem and progenitor cells migrate out of the medial pallium region, and proliferate while they move along the dentate migratory stream to the future anatomic site of dentate gyrus (Nelson et al, 2020). The peak of neurogenesis within the dentate gyrus, that is, the differentiation of stem and progenitor cells to generate granule cells, occurs between postnatal day (P) 0 and P14 (Li and Pleasure, 2014). During this neonatal period, neural stem cells named radial glia-like cells (RGLs) give rise to Tbr2+ transit-amplifying neuronal intermediate progenitor cells (nIPCs). nIPCs further differentiate into neuroblasts, which in turn produce granule cells. Among these cell types, only nIPCs are highly proliferative (Hochgerner et al, 2018). Thus, in theory, the balance between nIPC proliferation and differentiation during the neonatal period may affect the final number of granule cells, and eventually the size of the dentate gyrus. Nevertheless, it remains incompletely understood how human neurodevelopmental disorder genes regulate neurogenesis in the dentate gyrus.

Here, we used multiple genetically engineered mouse models to investigate the cellular and molecular mechanisms underlying Alazami syndrome, a human neurodevelopmental disorder characterized by global growth delay and impaired cognition. We find that knockout of *Larp7* in the central nervous system by nestin-Cre resulted in growth retardation and cognitive deficits in mice, reminiscent of salient features in children with Alazami syndrome. The size of hippocampal dentate gyrus is disproportionally reduced during the neonatal period. Importantly, hypoplasia of the dentate gyrus can be separated from overall growth retardation. Knockout of *Larp7* in Emx1+ neural stem and progenitor cells does not cause any growth defects in mice, but still results in an equally shrunken dentate gyrus as well as cognitive deficits. Consistent with cell-intrinsic defects during neurogenesis, upregulated P-TEFb activity enhances existing self-renewal transcriptional programs in nIPCs to impair their differentiation, limiting neurogenic output and hence reducing the size of dentate gyrus. These results reveal a previously underappreciated role of subtissular stem cell dynamics in the phenotypic manifestation of dysregulated transcription elongation. Because the transcriptomic profiles are highly alike in nIPCs during perinatal, juvenile, and adult neurogenesis (Hochgerner et al, 2018), these results suggest that therapeutic inhibition of P-TEFb may favor neurogenesis to ameliorate the effect of various adverse conditions known to compromise dentate gyrus volume and cognition.

## Results

### Conditional knockout of *Larp7* by nestin-Cre recapitulates salient features of patients with Alazami syndrome

The genetic diagnosis of Alazami syndrome is defined by the identification of inactivating variants in *LARP7*, containing frame-shift or premature stop codons. The common threads among these patients are short stature and cognitive deficits (Al-Hinai et al, 2022; Alazami et al, 2012; Amelie et al, 2025; Das et al, 2021; Dateki et al, 2018; Hollink et al, 2016; Imbert-Bouteille et al, 2019; Ling and Sorrentino, 2016; Soengas-Gonda et al, 2023; Thouqan et al, 2025; Wojcik et al, 2019). Thus, Alazami syndrome is a human neurodevelopmental disorder, based on the criteria of the fifth edition of the Diagnostic and Statistical Manual of Mental Disorders (DSM-5) (Gidziela et al, 2023).

However, it remains unknown how lack of LARP7 function results in neurodevelopmental disorders. Conventional or germline knockout of *Larp7* causes lethality between E17.5 and the time of birth for unknown reasons (Okamura et al, 2012). Thus, we generated *Larp7*-floxed mice (*Larp7*f/f; Fig. 1A), and crossed them with nestin-Cre mice (Tronche et al, 1999) to delete *Larp7* in the developing central nervous system (*Larp7*f/f;nestin-Cre). Total protein lysates extracted from E18.5 whole brains were analyzed to determine the knockout efficiency. Compared to the wild-type embryos (hereafter *Larp7*-floxed mice were used as wild-type control), Larp7 protein expression was reduced about 50% in heterozygotes (*Larp7*f/+;nestin-Cre), and was largely eliminated in homozygotes (*Larp7*f/f;nestin-Cre) (Fig. 1B). Heterozygous and homozygous pups were born at expected ratios, and had a normal lifespan (Fig. 1C). At postnatal day (P) 7, the body length of wild-type and heterozygous mice were comparable, but was reduced in homozygous mice with complete penetrance (Fig. 1D). The bodyweight was also reduced in both male and female knockout

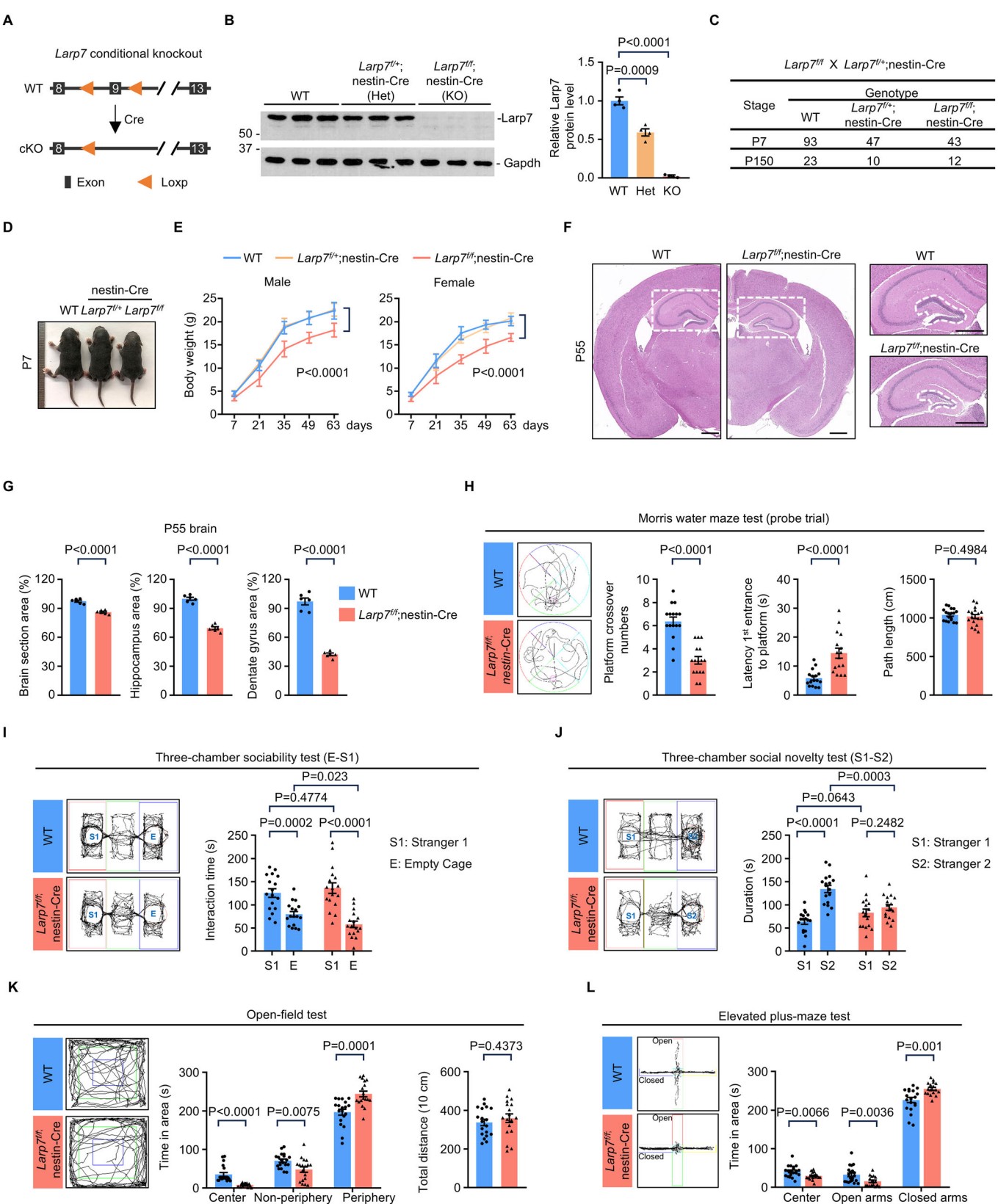

◄ **Figure 1. Conditional knockout of *Larp7* by nestin-Cre recapitulates salient features of patients with Alazami syndrome.**

(A) The strategy to generate conditional *Larp7* knockout mice. In knockout mice (cKO), the exon 9 of *Larp7* is deleted. WT, wild-type. (B) Protein blot analysis of Larp7 protein expression. Total protein lysates were generated from the forebrain of E18.5 wild-type (WT, $n = 3$), heterozygous (*Larp7$^{f/+}$*;nestin-Cre, $n = 3$), and homozygous (*Larp7$^{f/f}$*;nestin-Cre, $n = 3$) embryos, respectively. Error bars, mean ± SEM. Statistical significance was assessed using the two-sided unpaired Student's *t* test. (C) Genotype analyses of pups generated by crossing *Larp7$^{f/+}$*;nestin-Cre and *Larp7$^{f/f}$* mice. (D) Littermates of wild-type (WT), heterozygous (*Larp7$^{f/+}$*;nestin-Cre), and homozygous (*Larp7$^{f/f}$*;nestin-Cre) mice at postnatal day (P) 7. (E) Body weight growth curve of wild-type (WT, $n = 34$), heterozygotes (*Larp7$^{f/+}$*;nestin-Cre, $n = 19$), and homozygotes (*Larp7$^{f/f}$*;nestin-Cre, $n = 24$) from P7 to P63. Both male and female mice exhibit body weight reduction. Error bars, mean ± SEM. Statistical significance was assessed using the two-sided unpaired Student's *t* test. (F) Representative images showing hematoxylin and eosin (HE)-stained coronal sections of the brain from P55 wild-type (WT) and knockout (*Larp7$^{f/f}$*;nestin-Cre) mice ($n = 6$). The white box denotes the hippocampus. The boxed region is enlarged on the right to show the dentate gyrus, demarcated by white dashed lines. Scale bar: 500 μm. (G) The size of the whole brain (left panel), the hippocampus (middle panel) and the dentate gyrus (right panel) in P55 wild-type (WT; $n = 6$) and knockout (*Larp7$^{f/f}$*;nestin-Cre; $n = 6$) mice. Of note, the size of the dentate gyrus was disproportionally reduced in mutant mice. Error bars, mean ± SEM. Statistical significance was assessed using the two-sided unpaired Student's *t* test. (H) Morris water maze test. During the probe trial (platform removed), representative traces of wild-type (WT; $n = 16$) and knockout (*Larp7$^{f/f}$*;nestin-Cre; $n = 16$) mice in Morris water maze (left panel), the number of platform crossing, latency to find the location of removed platform, and the total path length (right panel) were documented by SMART video tracking system. Error bars, mean ± SEM. Statistical significance was assessed using the two-sided unpaired Student's *t* test. (I) Three-chamber sociability test. Representative traces of WT ($n = 17$) and knockout (*Larp7$^{f/f}$*;nestin-Cre; $n = 17$) mice in three-chamber (left panel), and total sniffing time spent towards empty chamber (E) or stranger 1 (S1) (right panel) were documented by SMART video tracking system. Error bars, mean ± SEM. Statistical significance was assessed using the two-sided unpaired Student's *t* test. (J) Three-chamber social novelty test. Representative traces of WT ($n = 17$) and knockout (*Larp7$^{f/f}$*;nestin-Cre; $n = 17$) mice in three-chamber (left panel), and total sniffing time spent toward stranger 1 (S1) or stranger 2 (S2) (right panel) were documented by SMART video tracking system. Error bars, mean ± SEM. Statistical significance was assessed using the two-sided unpaired Student's *t* test. (K) Open field test. Representative traces of WT ($n = 19$) and knockout (*Larp7$^{f/f}$*;nestin-Cre; $n = 18$) mice in the open field (left panel), the time spent in the indicated areas, including Center, Non-periphery and Periphery (middle panel), and total distance (right panel) were documented by SMART video tracking system. Error bars, mean ± SEM. Statistical significance was assessed using the two-sided unpaired Student's *t* test. (L) Elevated plus maze test. Representative traces of WT ($n = 20$) and knockout (*Larp7$^{f/f}$*;nestin-Cre; $n = 17$) mice in the elevated plus maze (left panel), the time spent in the indicated areas, including Center, Open arms, and Closed arms (right panel) were documented by SMART video tracking system. Error bars, mean ± SEM. Statistical significance was assessed using the two-sided unpaired Student's *t* test. Source data are available online for this figure.

mice (Fig. 1E). These observations are reminiscent of stunted growth in children with Alazami syndrome. Histological assessment of P55 brain revealed that the hippocampal dentate gyrus size was comparable in wild-type and heterozygous mice, but was markedly reduced in knockout mice (Fig. 1F). Consistent with reduced body length and weight, knockout mice had smaller brains at P55 (Fig. 1G). Notably, the degree of reduction was more severe in the dentate gyrus than the whole brain (Fig. 1G), demonstrating that *Larp7* deficiency preferentially affects the dentate gyrus. To analyze whether brain functions were impaired in knockout mice, several behavioral tests were carried out to assess functions associated with the dentate gyrus, such as spatial navigation and memory consolidation, exploration, and anxiety (Bannerman et al, 2014). In the Morris water maze test (Vorhees and Williams, 2006), the swimming distances were comparable between wild-type and knockout mice during the probe trial. However, mutant mice spent more time to find the location of removed platform, and crossed the location of removed platform less frequently than wild-type mice (Fig. 1H), indicating that spatial learning and memory is compromised in knockout mice. In three-chamber sociability test (Rein et al, 2020), both wild-type and knockout mice preferred the cage containing stranger 1 over the empty cage (Fig. 1I). However, in the following social novelty test, wild-type mice clearly preferred interacting with stranger 2, whereas mutant mice failed to distinguish stranger 1 from stranger 2 (Fig. 1J). In the open field test (Seibenhener and Wooten, 2015), wild-type and knockout mice exhibited similar ambulatory ability, but mutant mice exhibited significantly more anxiety-related behaviors (Fig. 1K). Similarly, anxiety-like behaviors were increased in the mutant mice (Fig. 1L), measured by the elevated plus maze test (Walf and Frye, 2007).

Taken together, we concluded that knockout of *Larp7* by nestin-Cre (*Larp7$^{f/f}$*;nestin-Cre) causes developmental defects, resembling core phenotypes observed in Alazami syndrome patients, that is, global growth delay and impaired cognitive functions. Whether the

dentate gyrus hypoplasia is associated with, or independent of, overall growth delay is addressed below in this study.

## Loss of *Larp7* causes enhanced self-renewal of neuronal intermediate progenitor cells in the dentate gyrus during the neonatal period

In this study, we focused on dissecting the mechanisms underlying shrunken dentate gyrus, due to its medical relevance in several human conditions such as intrauterine growth restriction, Alzheimer's disease, and aging (Babcock et al, 2021; Fotuhi et al, 2012; Gilchrist et al, 2018). Because the size of the dentate gyrus was reduced at P55 in *Larp7$^{f/f}$*;nestin-Cre mice (Fig. 1F,G), we examined timepoints before P55. During embryogenesis, neural stem and progenitor cells migrate out of the medial pallium region, and move along the dentate migratory stream to the future anatomic site of dentate gyrus (Amaral et al, 2007). At E18.5, the number of Tbr2$^+$ intermediate progenitor cells along the dentate migratory stream were indistinguishable between wild-type and *Larp7$^{f/f}$*;nestin-Cre brains (Appendix Fig. S1). After birth, the size of developing dentate gyrus was highly alike in wild-type and mutant mice at P0 and P7 (Fig. 2A). Thus, *Larp7* is dispensable for the formation of initial germinative matrices (Li and Pleasure, 2014). However, the dentate gyrus was smaller in mutant mice at P14 (Fig. 2A,B), and further shrunk at P55 (Fig. 1F). Thus, *Larp7* is required during the peak of neurogenesis in the dentate gyrus between P7 and P14. During postnatal development, granule cells in the dentate gyrus are produced by a typical hierarchy of stem cell differentiation (Hochgerner et al, 2018). Neural stem cells named radial glia-like cells (RGLs) give rise to transit-amplifying progenitors called neuronal intermediate progenitor cells (nIPCs). nIPCs further differentiate into neuroblasts, which in turn produce granule cells. Among these cell types, only nIPCs are highly proliferative. To further narrow down the timepoint when the defect happens

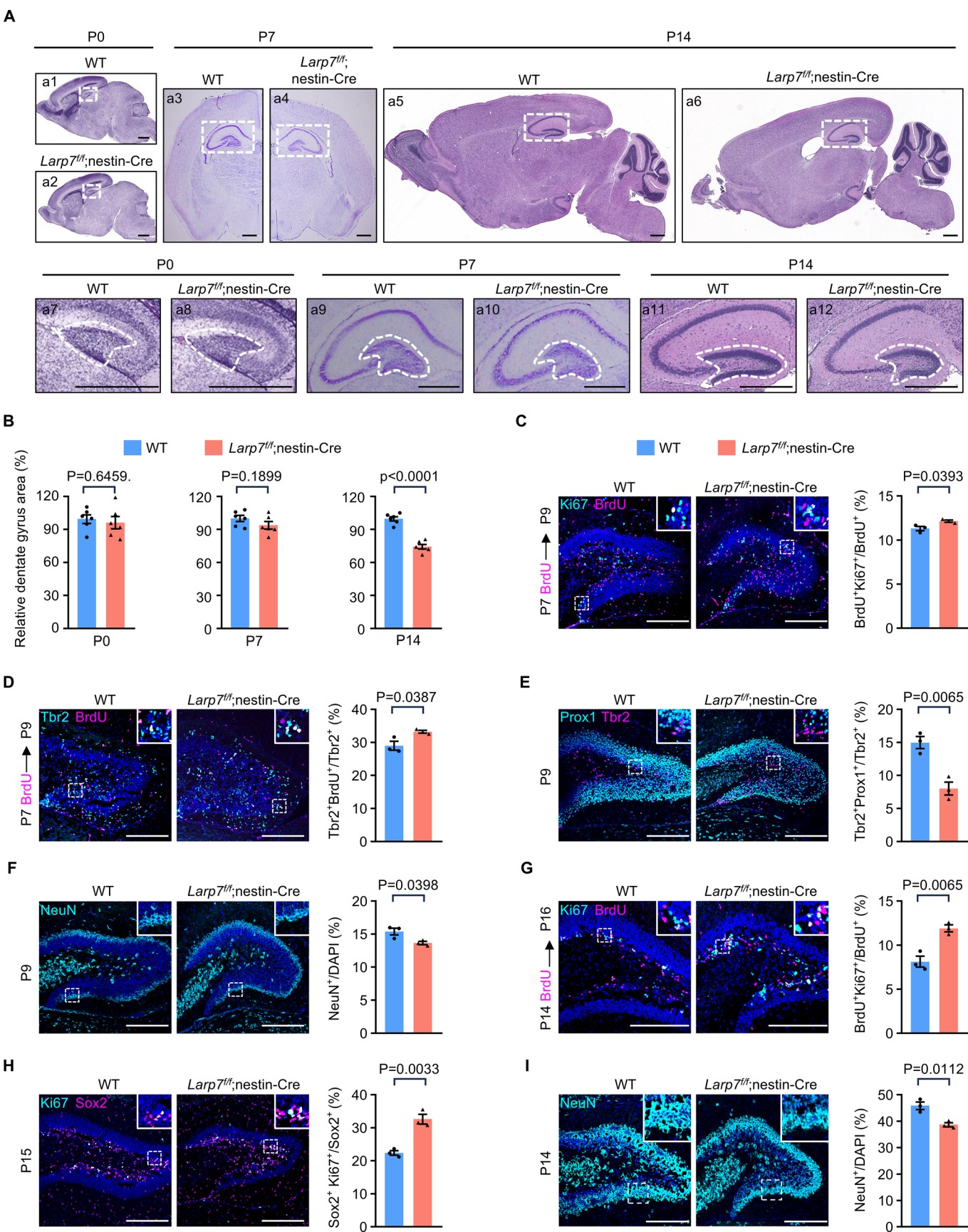

**Figure 2.  Loss of *Larp7* causes enhanced self-renewal of neuronal intermediate progenitor cells in the dentate gyrus during the neonatal period.**

(A) Representative images showing sections of the brain from wild-type (WT; *n* = 6) and knockout (*Larp7*^f/f^;nestin-Cre; *n* = 6) mice at P0, P7, and P14. P0 and P14 samples were sagittal sections stained by hematoxylin and eosin (HE). P7 samples were coronal sections stained by cresyl violet. The hippocampus was demarcated by the white box (a1–a6 in upper panel). The boxed areas were enlarged to show the dentate gyrus (demarcated by white-dashed lines; a7–a12 in lower panel). Scale bar: 500 μm. (B) Quantification of relative areas of the dentate gyrus of wild-type (WT) and mutant (*Larp7*^f/f^;nestin-Cre) mice at P0 (left), P7 (middle) and P14 (right). The mean ratio of dentate gyrus area versus the whole brain in WT mice was set as 100% (*n* = 6). This ratio was decreased in mutant mice at P14 (*n* = 6). Error bars, mean ± SEM. Statistical significance was assessed using the two-sided unpaired Student's *t* test. (C) Cell proliferation status in the dentate gyrus of wild-type (WT; *n* = 3) and mutant (*Larp7*^f/f^;nestin-Cre; *n* = 3) mice at P7-P9. BrdU was administered via peritoneal injection at P7, followed by the visualization of Ki67 and BrdU by immunofluorescence within the dentate gyrus at P9. White dashed boxed areas were enlarged on top-right corners. The percentage of Ki67^+^BrdU^+^ cells among BrdU^+^ cells, indicative of proliferating cells at P7-P9, was quantified (right panel). Error bars, mean ± SEM. Statistical significance was assessed using the two-sided unpaired Student's *t* test. Scale bar: 200 μm. (D) The proliferative status of nIPCs in the dentate gyrus of wild-type (WT; *n* = 3) and mutant (*Larp7*^f/f^;nestin-Cre; *n* = 3) mice at P7. BrdU was administered at P7, followed by the visualization of BrdU and Tbr2 by immunofluorescence within the dentate gyrus at P9. White dashed boxed areas were enlarged on top-right corners. The percentage of BrdU^+^Tbr2^+^ cells among Tbr2^+^ cells, indicative of proliferating nIPCs, was quantified (right panel). Error bars, mean ± SEM. Statistical significance was assessed using the two-sided unpaired Student's *t* test. Scale bar: 200 μm. (E) Comparison of differentiating nIPCs in the dentate gyrus of wild-type (WT; *n* = 3) and mutant (*Larp7*^f/f^; nestin-Cre; *n* = 3) mice at P9. Cells positive for both Tbr2 and Prox1 were regarded as differentiating nIPCs. White dashed boxed areas were enlarged on top-right corners. The percentage of Prox1^+^Tbr2^+^ cells among Tbr2^+^ cells, indicative of differentiating nIPCs, was quantified (right panel). Error bars, mean ± SEM. Statistical significance was assessed using the two-sided unpaired Student's *t* test. Scale bar: 200 μm. (F) The status of granule cells in the dentate gyrus of wild-type (WT; *n* = 3) and mutant (*Larp7*^f/f^; nestin-Cre; *n* = 3) mice at P9. Granule cells were indicated by positive staining for NeuN, encoded by *Rbfox3*. White dashed boxed areas were enlarged on top-right corners. The percentage of NeuN^+^ cells was quantified (right panel). Error bars, mean ± SEM. Statistical significance was assessed using the two-sided unpaired Student's *t* test. Scale bar: 200 μm. (G) Cell proliferation status in the dentate gyrus of wild-type (WT; *n* = 3) and mutant (*Larp7*^f/f^;nestin-Cre; *n* = 3) mice at P14-P16. BrdU was administered via peritoneal injection at P14, followed by the visualization of Ki67 and BrdU by immunofluorescence within the dentate gyrus at P16. White dashed boxed areas were enlarged on top-right corners. The percentage of Ki67^+^BrdU^+^ cells among BrdU^+^ cells, indicative of proliferating cells at P14-P16, was quantified (right panel). Error bars, mean ± SEM. Statistical significance was assessed using the two-sided unpaired Student's *t* test. Scale bar: 200 μm. (H) The proliferative status of neural stem and progenitor cells in the dentate gyrus of wild-type (WT; *n* = 3) and mutant (*Larp7*^f/f^;nestin-Cre; *n* = 3) mice at P15. Proliferating neural stem and progenitor cells were indicated by positive immunofluorescence for both Ki67 and Sox2. White dashed boxed areas were enlarged on top-right corners. The percentage of Ki67^+^Sox2^+^ cells among Sox2^+^ cells, indicative of proliferating neural stem and progenitor cells, was quantified (right panel). Error bars, mean ± SEM. Statistical significance was assessed using the two-sided unpaired Student's *t* test. Scale bar: 200 μm. (I) The status of granule cells in the dentate gyrus of wild-type (WT; *n* = 3) and mutant (*Larp7*^f/f^;nestin-Cre; *n* = 3) mice at P14. Granule cells were indicated by positive staining for NeuN. White dashed boxed areas were enlarged on top-right corners. The percentage of NeuN^+^ cells was quantified (right panel). Error bars, mean ± SEM. Statistical significance was assessed using the two-sided unpaired Student's *t* test. Scale bar: 200 μm. Source data are available online for this figure.

between P7 and P14, we carried out maker analyses. At P7–P9, bromodeoxyuridine (BrdU) and Ki-67 labeling demonstrated that there was a moderate increase in proliferative cells in the dentate gyrus of mutant mice (Fig. 2C). Consistently, the number of proliferative Tbr2^+^ nIPCs also moderately increased in mutant mice (Fig. 2D). The differentiation of Tbr2^+^ nIPCs towards granule cells requires transcription factor Prox1 (Lavado et al, 2010). Interestingly, the percentage of Tbr2^+^ IPCs expressing Prox1 dropped in mutant mice (Fig. 2E), concomitant with reduced numbers of NeuN^+^ mature granule cells (Fig. 2F). At P14-P16, an increase in proliferative nIPCs (Fig. 2G,H), and a decrease in NeuN^+^ mature granule cells became more prominent (Fig. 2I). These results indicate that the self-renewal of nIPCs is enhanced, at the expense of their differentiation towards granule cells.

## Single-cell sequencing reveals that *Larp7* loss enhances self-renewal transcriptional programs in neuronal intermediate progenitor cells

To gain mechanistic insight, we carried out single-cell RNA sequencing analyses on P7 dentate gyrus obtained by microdissection. This timepoint was chosen because the dentate gyrus started to shrink between P7 and P9 (Fig. 2). In total, 12,820 wild-type and 7153 *Larp7*^f/f^;nestin-Cre cells passed quality control and filtering steps, and used for downstream bioinformatic analysis. Ten major cell clusters were identified by unbiased clustering (Seurat), and their identities were assigned using previously defined markers (Appendix Fig. S2A,B) (Hochgerner et al, 2018). We focused on characterizing the neuronal lineage, including radial glia-like (RGL), neuronal intermediate progenitor cell (nIPC) and

neuroblast. At P7, the cellular states and the differentiation trajectory of the neuronal linage were similar in wild-type and *Larp7*^f/f^;nestin-Cre dentate gyrus (Fig. 3A; Appendix Fig. S2C). Loss of Larp7 protein is expected to enhance cellular P-TEFb activity to promote RNA polymerase II transcriptional output. Accordingly, the number of genes transcribed and the amount of mRNA per cell were increased in *Larp7*^f/f^;nestin-Cre cells, most prominently in nIPCs (Fig. 3B). The number of proliferating nIPCs were increased in mutant dentate gyrus by marker analyses (Fig. 2). Indeed, single-cell sequencing analysis confirmed that a higher percentage of mutant nIPCs were in an active cell cycle (Fig. 3C). Consistently, genes essential for cell cycle progression were expressed at higher levels in mutant nIPCs than in wild-type nIPCs (Fig. 3D). Furthermore, we analyzed differentially expressed genes in wild-type and mutant nIPCs. P-TEFb promotes RNA polymerase II transcription. Accordingly, 93% (257/277) of differentially expressed genes in mutant nIPCs were upregulated, compared to their wild-type counterparts (Fig. 3E). These genes included many known positive regulators of nIPC self-renewal (Fig. 3F), indicating that a higher percentage of mutant nIPCs are proliferating in an undifferentiated state. In favor of enhanced self-renewal of nIPCs, unbiased Gene Ontology (GO) enrichment analysis demonstrated that upregulated genes were enriched in "DNA replication", "positive regulation of cell cycle" and "neural precursor cell proliferation pathways" (Fig. 3G). ChEA3 (Keenan et al, 2019) was used to deduce transcriptional networks underlying differentially expressed genes (Fig. 3H). This analysis revealed that these genes were regulated by core self-renewal transcription factors such as Sox2 and Nanog, and key cell cycle regulators such as E2F family transcription factors in G1/S transition and Foxm1 in G2/M

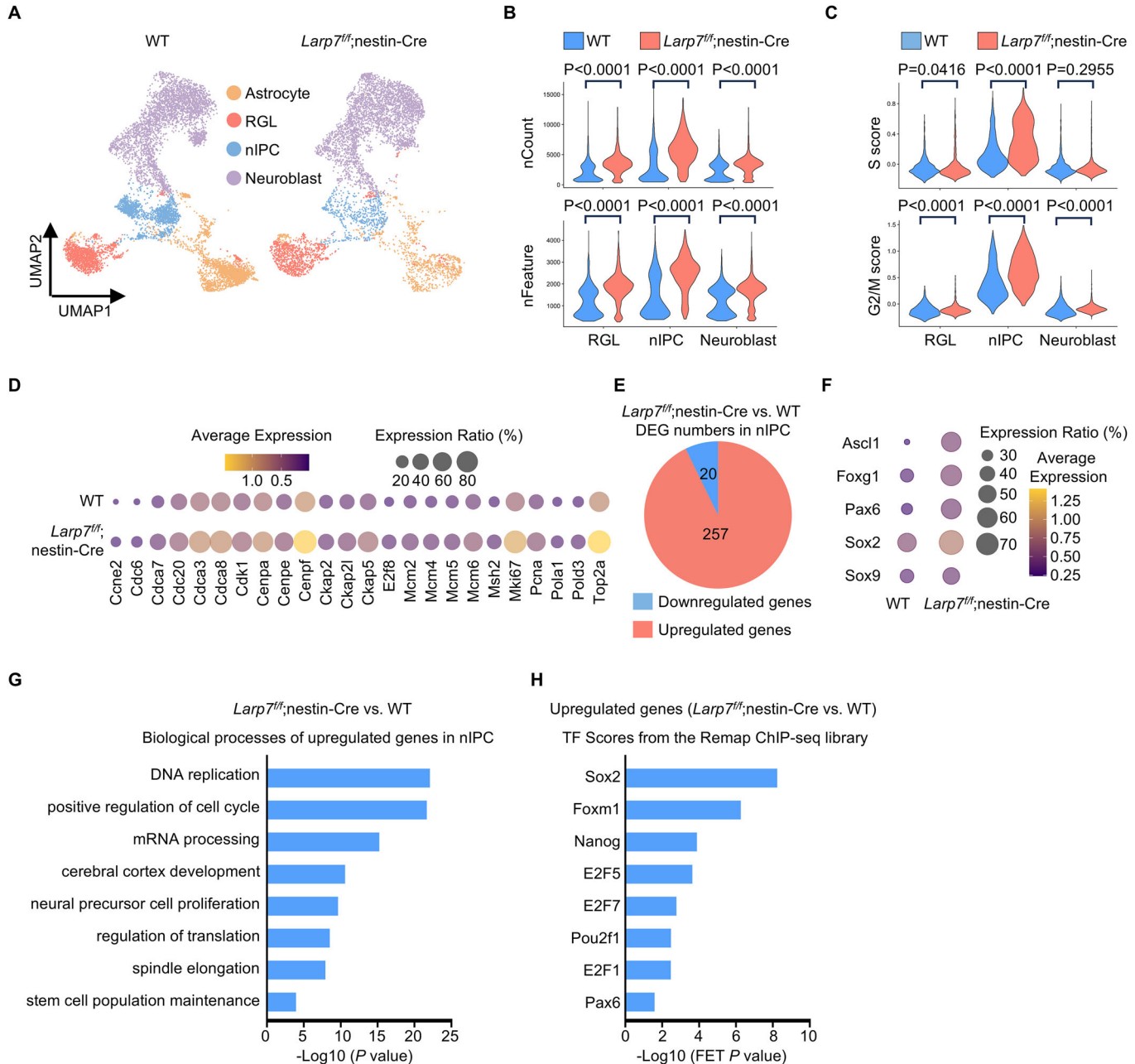

**Figure 3.** Single-cell sequencing reveals that *Larp7* loss enhances self-renewal transcriptional programs in neuronal intermediate progenitor cells.

(A) Single-cell sequencing analysis of the microdissected dentate gyrus from P7 wild-type (WT) and mutant (*Larp7f/f*;nestin-Cre) mice. Four cell types, astrocyte, RGL, nIPC and neuroblast, were visualized by UMAP. RGL, radial glia-like. nIPC, neuronal intermediate progenitor cell. (B) Violin plot presenting the number of genes transcribed (upper panel) as well as the amount of mRNA (lower panel) per cell in RGLs (WT; *n* = 1457 cells; *Larp7f/f*;nestin-Cre; *n* = 1013 cells), nIPCs (WT; *n* = 1600 cells; *Larp7f/f*; nestin-Cre; *n* = 580 cells) and neuroblasts (WT; *n* = 3561 cells; *Larp7f/f*;nestin-Cre; *n* = 3604 cells). Of note, mutant nIPCs (*Larp7f/f*;nestin-Cre) exhibit the most prominent increase in the number of genes transcribed and the amount of mRNA per cell. Statistical significance was assessed using the two-sided Wilcoxon rank-sum test. (C) Violin plot presenting cell cycle scoring analysis in RGLs (WT; *n* = 1457 cells; *Larp7f/f*;nestin-Cre; *n* = 1013 cells), nIPCs (WT; *n* = 1600 cells; *Larp7f/f*;nestin-Cre; *n* = 580 cells) and neuroblasts (WT; *n* = 3561 cells; *Larp7f/f*;nestin-Cre; *n* = 3604 cells). Of note, among these cell types, only nIPCs are proliferative cells as indicated by high S and G2/M scores. Statistical significance was assessed using the two-sided Wilcoxon rank-sum test. (D) Bubble plot showing the expression of key cell cycle genes in nIPCs from WT and knockout (*Larp7f/f*;nestin-Cre) mice. (E) Pie chart showing the number of differentially expressed genes in mutant (*Larp7f/f*;nestin-Cre) versus WT nIPCs. (F) Bubble plot presenting the expression of key self-renewal transcription factors in WT and mutant (*Larp7f/f*;nestin-Cre) nIPCs. (G) GO analysis of the biological process of upregulated genes in mutant (*Larp7f/f*;nestin-Cre) versus WT nIPCs. Statistical significance was assessed using the two-sided Wilcoxon rank-sum test. (H) ChEA3 transcription factor (TF) analysis of upregulated genes in mutant nIPCs. Statistical significance was assessed using the two-sided Wilcoxon rank-sum test. Source data are available online for this figure.

transition (Fischer et al, 2022). Taken together, both phenotypic analyses (Fig. 2) and single-cell transcriptomic profiling (Fig. 3) suggest that loss of Larp7 protein activates P-TEFb to enhance self-renewal programs in neuronal progenitor cells, resulting in compromised neuronal differentiation.

## Dentate gyrus hypoplasia is independent of global developmental delay

So far, we have demonstrated that knockout of *Larp7* by nestin-Cre (*Larp7^f/f*;nestin-Cre) results in impaired cognitive functions associated with dentate gyrus hypoplasia, but also causes developmental delay suggested by reduced body length and weight (Fig. 1D,E). The recombinant activity of nestin-Cre is active in multiple cell types spanning the whole brain, but also in other organs such as kidney (Wang et al, 2024), stomach and lung (Appendix Fig. S3). Therefore, it is unclear whether dentate gyrus hypoplasia is a result of global developmental delay, or an independent cell-intrinsic phenotype.

Because single cell analyses by immunofluorescence and sequencing have revealed that the most prominent defects occur in nIPCs (Figs. 2 and 3), we hypothesized that dentate gyrus hypoplasia might be caused by intrinsic defects in neural stem and progenitor cells. During embryogenesis, RGLs and nIPCs are derived from Emx1+ neural stem and progenitor cells. Therefore, we crossed *Larp7*-floxed mice with Emx1-Cre mice (Gorski et al, 2002) to delete *Larp7* in Emx1+ neural stem and progenitor cells (*Larp7^f/f*;Emx1-Cre). Identical to *Larp7^f/f*;nestin-Cre mice, the reduction of the dentate gyrus size occurred in *Larp7^f/f*;Emx1-Cre mice from P7 onwards (Fig. 4A). The severity was also highly similar to that of *Larp7^f/f*;nestin-Cre mice (Fig. 4B). However, in sharp contrast to *Larp7^f/f*;nestin-Cre mice, there was no reduction in brain size, body length or body weight in *Larp7^f/f*;Emx1-Cre mice at these time points (Appendix Fig. S4). Even after 8 months, the appearance of wild-type and *Larp7^f/f*;Emx1-Cre mice was indistinguishable (Fig. 4C). Thus, shrunken dentate gyrus caused by *Larp7* loss does not stem from global developmental delay, but rather caused by cell-intrinsic defects in neural stem and progenitor cells.

At P7, BrdU and Ki-67 labeling revealed that there were more proliferating cells in *Larp7^f/f*;Emx1-Cre dentate gyrus (Fig. 4D). Marker analyses by immunofluorescence confirmed that the number of self-renewing Sox2+ and Tbr2+ nIPCs was increased in *Larp7^f/f*;Emx1-Cre dentate gyrus (Fig. 4E,F). Concomitantly, there was a decrease in the number of differentiating Tbr1+Tbr2+ and Prox1+Tbr2+ nIPCs (Fig. 4G,H), and NeuN+ mature granule cells (Fig. 4I). At P15, the trend remained (Fig. 4J,K). In addition, *Larp7^f/f*;Emx1-Cre mice exhibited same cognitive deficits in behavioral tests (Fig. 4L–P), similar to *Larp7^f/f*;nestin-Cre mice (Fig. 1). Thus, regarding the dentate gyrus, the timing and severity of phenotypic defects were highly similar in *Larp7^f/f*;nestin-Cre and *Larp7^f/f*;Emx1-Cre mice. These observations favor the notion that *Larp7* loss enhances nIPCs self-renewal at the expense of their differentiation by a cell-intrinsic mechanism.

## Single-cell sequencing reveals shared cell-intrinsic defects in neuronal intermediate progenitor cells, regardless of global developmental delay

Both *Larp7^f/f*;nestin-Cre and *Larp7^f/f*;Emx1-Cre mice exhibit the same defect in the dentate gyrus. However, only *Larp7^f/f*;nestin-Cre

mice exhibit global developmental delay (compare Fig. 1D,E with Fig. 4C and Appendix Fig. S4). If dentate gyrus hypoplasia is truly independent of global developmental delay, we reasoned that there should be some shared cellular defects present in both *Larp7^f/f*;nestin-Cre and *Larp7^f/f*;Emx1-Cre dentate gyri.

To elucidate shared molecular and cellular defects, we carried out single-cell RNA sequencing analyses on P7 *Larp7^f/f*;Emx1-Cre dentate gyrus, and compared this dataset with that obtained from P7 *Larp7^f/f*;nestin-Cre dentate gyrus. At P7, the overall states of wild-type RGLs, nIPCs and neuroblasts, respectively, were similar to their counterparts in *Larp7^f/f*;nestin-Cre or *Larp7^f/f*;Emx1-Cre dentate gyrus (Fig. 5A). However, similar to *Larp7^f/f*;nestin-Cre cells (Fig. 3B), the number of genes transcribed and the amount of mRNA per cell were increased in *Larp7^f/f*;Emx1-Cre cells, also most prominently in nIPCs (Fig. 5B). Like *Larp7^f/f*;nestin-Cre nIPCs (Fig. 3C), more *Larp7^f/f*;Emx1-Cre nIPCs were in an active cell cycle (Fig. 5C). The majority of differentially expressed genes (88%; 221/251) in *Larp7^f/f*;Emx1-Cre nIPCs were upregulated (Fig. 5D). Importantly, 65% of upregulated genes in *Larp7^f/f*;Emx1-Cre nIPCs were also upregulated in *Larp7^f/f*;nestin-Cre nIPCs (Fig. 5E). These commonly upregulated genes include many genes essential for cell cycle progression (Fig. 5F), and known positive regulators of nIPC self-renewal (Fig. 5G). Unbiased GO analysis demonstrated that upregulated genes were enriched in "DNA replication", "positive regulation of cell cycle", and "neural precursor cell proliferation" pathways for both *Larp7^f/f*;nestin-Cre (Fig. 3G) and *Larp7^f/f*;Emx1-Cre nIPCs (Fig. 5H). ChEA3 analysis was used to infer transcriptional networks underlying differentially expressed genes. For both *Larp7^f/f*;nestin-Cre and *Larp7^f/f*;Emx1-Cre nIPCs, similar transcription factors were enriched, including core self-renewal transcription factors Sox2 and Nanog, and key cell cycle regulators such as E2F family transcription factors and Foxm1 (Fig. 5I).

Furthermore, we carried out single-cell RNA sequencing analyses on P14 *Larp7^f/f*;Emx1-Cre dentate gyrus. P14 timepoint was chosen because the peak of neurogenesis in the dentate gyrus occurs within 14 days after birth. Similarly, the self-renewal transcriptional programs were enhanced in *Larp7^f/f*;Emx1-Cre nIPCs (Appendix Fig. S5). Taken together, single-cell sequencing analyses demonstrate that the same cellular defects occur in nIPCs from both *Larp7^f/f*;nestin-Cre and *Larp7^f/f*;Emx1-Cre dentate gyri. These observations provide further support for the notion that *Larp7* loss enhances nIPCs self-renewal at the expense of their differentiation by a cell-intrinsic mechanism, independent of global developmental delay.

## Loss of *Larp7* enhances self-renewal and compromises differentiation of neural stem and progenitor cells in vitro

To further corroborate the cell-intrinsic nature of compromised differentiation observed in vivo, we carried out the neurosphere assay. The dentate gyrus was microdissected from P1 wild-type and *Larp7^f/f*;Emx1-Cre mice, respectively, and used to culture neural stem and progenitor cells in vitro. Under the self-renewal culture condition (Soares et al, 2020), the microscopic morphology of wild-type and *Larp7* knockout neurospheres was highly similar (Fig. 6A). In *Larp7* knockout neurospheres, as expected, the expression of Larp7 protein as well as 7sk snRNA were essentially eliminated (Fig. 6B). Consistent with a heightened P-TEFb activity, the Ser2

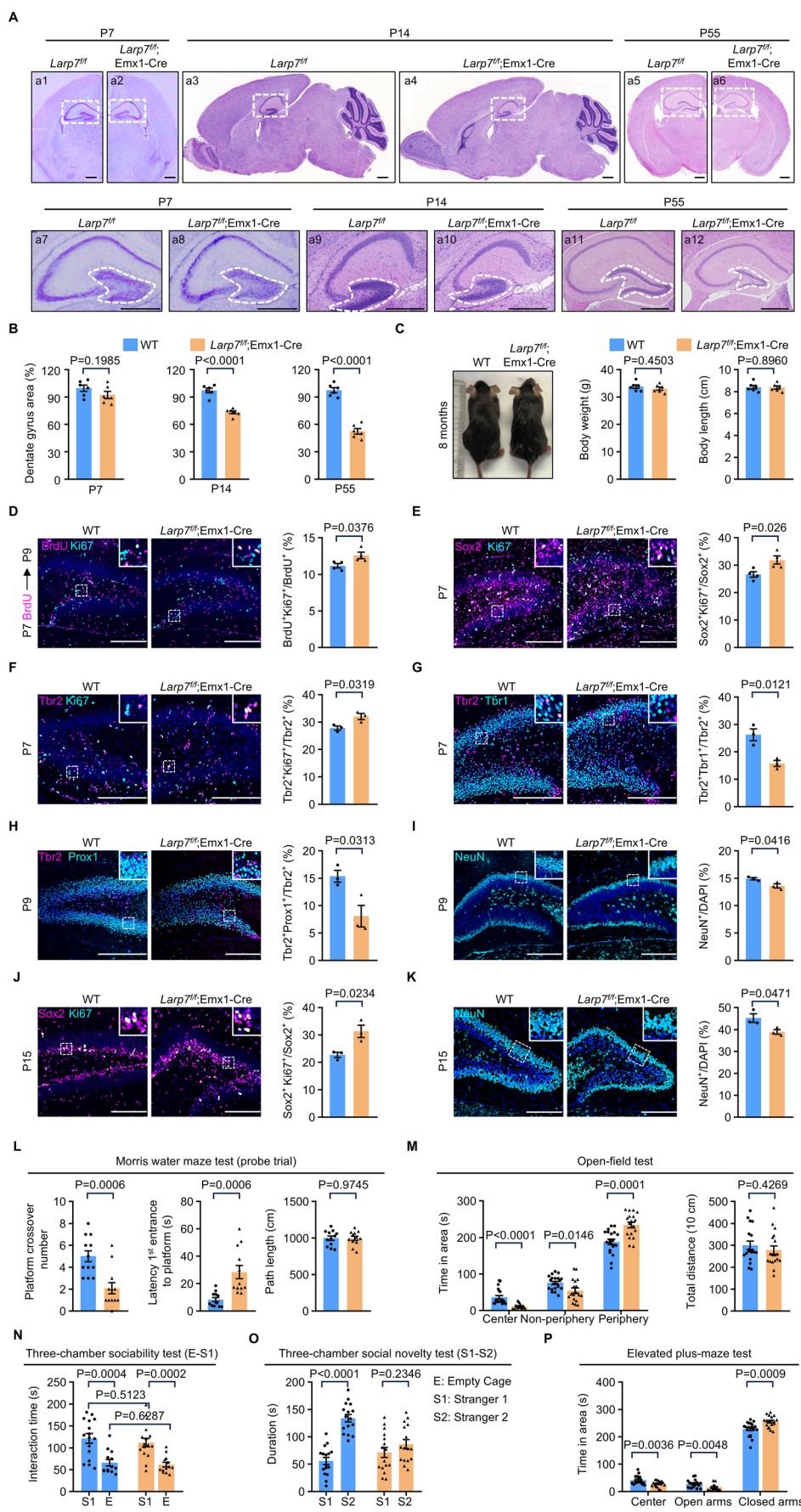

◄ **Figure 4. Dentate gyrus hypoplasia is independent of global developmental delay.**

(A) Representative images showing sections of the brain from wild-type (WT; $n = 6$) and knockout ($Larp7^{f/f}$;Emx1-Cre; $n = 6$) mice at P7, P14, and P55. P7 samples were coronal sections stained by cresyl violet. P14 samples were sagittal sections and P55 samples were coronal sections stained by hematoxylin and eosin (HE). The hippocampus was demarcated by the white box (a1–a6 in upper panel). The boxed areas were enlarged to show the dentate gyrus (demarcated by white-dashed lines; a7–a12 in lower panel). Scale bar: 500 μm. (B) Quantification of relative areas of the dentate gyrus of wild-type (WT) and mutant ($Larp7^{f/f}$;Emx1-Cre) mice at P7 (left), P14 (middle) and P55 (right). The mean ratio of dentate gyrus area versus the whole brain in WT mice was set as 100% ($n = 6$). This ratio was decreased in mutant mice at P14 and P55 ($n = 6$). Of note, unlike $Larp7^{f/f}$;nestin-Cre mice, the whole brain size of $Larp7^{f/f}$;Emx1-Cre mice was not reduced at P7 ($n = 6$), P14 ($n = 6$) or P55 ($n = 6$) (Appendix Fig. S4). Error bars, mean ± SEM. Statistical significance was assessed using the two-sided unpaired Student's *t* test. (C) Littermates of wild-type (WT) and homozygous ($Larp7^{f/f}$;Emx1-Cre) mice at 8 months. Of note, unlike $Larp7^{f/f}$;nestin-Cre mice, the body length and weight of $Larp7^{f/f}$;Emx1-Cre mice were indistinguishable from WT counterparts ($n = 6$). Error bars, mean ± SEM. Statistical significance was assessed using the two-sided unpaired Student's *t* test. (D) The status of cell proliferation in the dentate gyrus of wild-type (WT; $n = 3$) and mutant ($Larp7^{f/f}$;Emx1-Cre; $n = 3$) mice at P7-P9. BrdU was administered via peritoneal injection at P7, followed by the visualization of Ki67 and BrdU by immunofluorescence within the dentate gyrus at P9. White dashed boxed areas were enlarged on top-right corners. The percentage of Ki67⁺BrdU⁺ cells among BrdU⁺ cells, indicative of proliferating cells at P7-P9, was quantified (right panel). Error bars, mean ± SEM. Statistical significance was assessed using the two-sided unpaired Student's *t* test. Scale bar: 200 μm. (E) The proliferative status of neural stem and progenitor cells in the dentate gyrus of wild-type (WT; $n = 3$) and mutant ($Larp7^{f/f}$;Emx1-Cre; $n = 3$) mice at P7. White dashed boxed areas were enlarged on top-right corners. The percentage of Ki67⁺Sox2⁺ cells among Sox2⁺ cells, indicative of proliferating neural stem and progenitor cells, was quantified (right panel). Error bars, mean ± SEM. Statistical significance was assessed using the two-sided unpaired Student's *t* test. Scale bar: 200 μm. (F) The proliferative status of nIPCs in the dentate gyrus of wild-type (WT; $n = 3$) and mutant ($Larp7^{f/f}$; Emx1-Cre; $n = 3$) mice at P7. White dashed boxed areas were enlarged on top-right corners. The percentage of Ki67⁺Tbr2⁺ cells among Tbr2⁺ cells, indicative of proliferating nIPCs, was quantified (right panel). Error bars, mean ± SEM. Statistical significance was assessed using the two-sided unpaired Student's *t* test. Scale bar: 200 μm. (G) The differentiation status of nIPCs in the dentate gyrus of wild-type (WT; $n = 3$) and mutant ($Larp7^{f/f}$;Emx1-Cre; $n = 3$) mice at P7. White dashed boxed areas were enlarged on top-right corners. The percentage of Tbr1⁺Tbr2⁺ cells among Tbr2⁺ cells, indicative of differentiating nIPCs, was quantified (right panel). Error bars, mean ± SEM. Statistical significance was assessed using the two-sided unpaired Student's *t* test. Scale bar: 200 μm. (H) The differentiation status of nIPCs in the dentate gyrus of wild-type (WT; $n = 3$) and mutant ($Larp7^{f/f}$;Emx1-Cre; $n = 3$) mice at P9. White dashed boxed areas were enlarged on top-right corners. The percentage of Prox1⁺Tbr2⁺ cells among Tbr2⁺ cells, indicative of differentiating nIPCs, was quantified (right panel). Error bars, mean ± SEM. Statistical significance was assessed using the two-sided unpaired Student's *t* test. Scale bar: 200 μm. (I) The differentiation status of nIPCs in the dentate gyrus of wild-type (WT; $n = 3$) and mutant ($Larp7^{f/f}$;Emx1-Cre; $n = 3$) mice at P9. White dashed boxed areas were enlarged on top-right corners. The percentage of NeuN⁺ cells among dentate gyrus cells, indicative of differentiated nIPCs, was quantified (right panel). Error bars, mean ± SEM. Statistical significance was assessed using the two-sided unpaired Student's *t* test. Scale bar: 200 μm. (J) The proliferative status of neural stem cells in the dentate gyrus of wild-type (WT; $n = 3$) and mutant ($Larp7^{f/f}$;Emx1-Cre; $n = 3$) mice at P15. White dashed boxed areas were enlarged on top-right corners. The percentage of Ki67⁺Sox2⁺ cells among Sox2⁺ cells, indicative of proliferating neural stem cells, was quantified (right panel). Error bars, mean ± SEM. Statistical significance was assessed using the two-sided unpaired Student's *t* test. Scale bar: 200 μm. (K) The differentiation status of nIPCs in the dentate gyrus of wild-type (WT; $n = 3$) and mutant ($Larp7^{f/f}$;Emx1-Cre; $n = 3$) mice at P15. White dashed boxed areas were enlarged on top-right corners. The percentage of NeuN⁺ cells among all cells in the dentate gyrus, indicative of differentiated nIPCs, was quantified (right panel). Error bars, mean ± SEM. Statistical significance was assessed using the two-sided unpaired Student's *t* test. Scale bar: 200 μm. (L) Morris water maze test. During the probe trial (platform removed), wild-type (WT; $n = 12$) and knockout ($Larp7^{f/f}$;Emx1-Cre; $n = 12$) mice were tested. The number of platform crossing, latency to find the location of removed platform, and the total path length were documented by SMART video tracking system. Error bars, mean ± SEM. Statistical significance was assessed using the two-sided unpaired Student's *t* test. (M) Open field test. Wild-type (WT; $n = 20$) and knockout ($Larp7^{f/f}$;Emx1-Cre; $n = 18$) mice were tested. The time spent in the indicated areas, including Center, Non-periphery and Periphery, and total distance were documented by SMART video tracking system. Error bars, mean ± SEM. Statistical significance was assessed using the two-sided unpaired Student's *t* test. (N) Three-chamber sociability test. Wild-type (WT; $n = 16$) and knockout ($Larp7^{f/f}$;Emx1-Cre; $n = 13$) mice were tested. Total sniffing time spent towards empty chamber (E) or stranger 1 (S1) were documented by SMART video tracking system. Error bars, mean ± SEM. Statistical significance was assessed using the two-sided unpaired Student's *t* test. (O) Three-chamber social novelty test. Wild-type (WT, $n = 16$) and knockout ($Larp7^{f/f}$;Emx1-Cre; $n = 16$) mice were tested. Total sniffing time spent toward stranger 1 (S1) or stranger 2 (S2) were documented by SMART video tracking system. Error bars, mean ± SEM. Statistical significance was assessed using the two-sided unpaired Student's *t* test. (P) Elevated plus maze test. Wild-type (WT; $n = 20$) and knockout ($Larp7^{f/f}$;Emx1-Cre; $n = 18$) mice were tested. The time spent in the indicated areas, including Center, Open arms, and Closed arms, was documented by SMART video tracking system. Error bars, mean ± SEM. Statistical significance was assessed using the two-sided unpaired Student's *t* test.

phosphorylation was increased in *Larp7* knockout neurospheres (Fig. 6B). To address whether loss of *Larp7* has functional consequences on RNA polymerase II transcription, we analyzed the chromatin accessibility by the assay for transposase-accessible chromatin using sequencing (ATAC-seq) (Grandi et al, 2022). Higher transcription activities are correlated with more accessible chromatin. Compared to wild-type cells, *Larp7* knockout cells had significantly more ATAC-seq peaks in the -2,000 bp to +2,000 bp region surrounding the annotated transcription start site (TSS) (Fig. 6C), indicating a higher transcriptional activity in *Larp7* knockout cells. Furthermore, using Cleavage Under Targets and Tagmentation (CUT&Tag) assay (Kaya-Okur et al, 2019), we measured the density of total as well as active RNA polymerase II on annotated genes. Consistent with elevated P-TEFb activity, the pausing index of RNA polymerase II was decreased (Fig. 6D), and more active RNA polymerase II peaks were detected in the gene body regions in *Larp7* knockout cells than wild-type cells (Fig. 6E). Differentially expressed genes in *Larp7*-null nIPCs, in vivo, are predicted by bioinformatics analyses to be target genes of the Sox2

transcriptional factor (Figs. 3H and 5I). To validate this prediction in neurospheres, we carried out CUT&Tag assay to map the genomic distribution of Sox2 protein. This analysis revealed that the Sox2 transcriptional program was indeed enhanced in *Larp7*-null neurospheres in vitro (Appendix Fig. S6). We conclude that *Larp7* loss increases the transcriptional output of RNA polymerase II in neural stem and progenitor cells in vitro, similar to their counterparts in vivo.

We assessed whether *Larp7* loss enhances self-renewal to compromise differentiation in vitro. The self-renewal of neural stem and progenitor cells in vitro is maintained by using EGF and FGF factors. Under optimal culture condition, the neurospheres contain densely packed cells and display a round structure with a smooth perimeter. When the concentration of EGF and FGF was lowered to 10% of that in the standard culture condition, wild-type neurospheres started to lose the round morphology and exhibited multiple branch-like structures extending from the core. In contrast, the morphology of *Larp7* knockout neurospheres was minimally affected (Fig. 6F), indicating that the self-renewal is

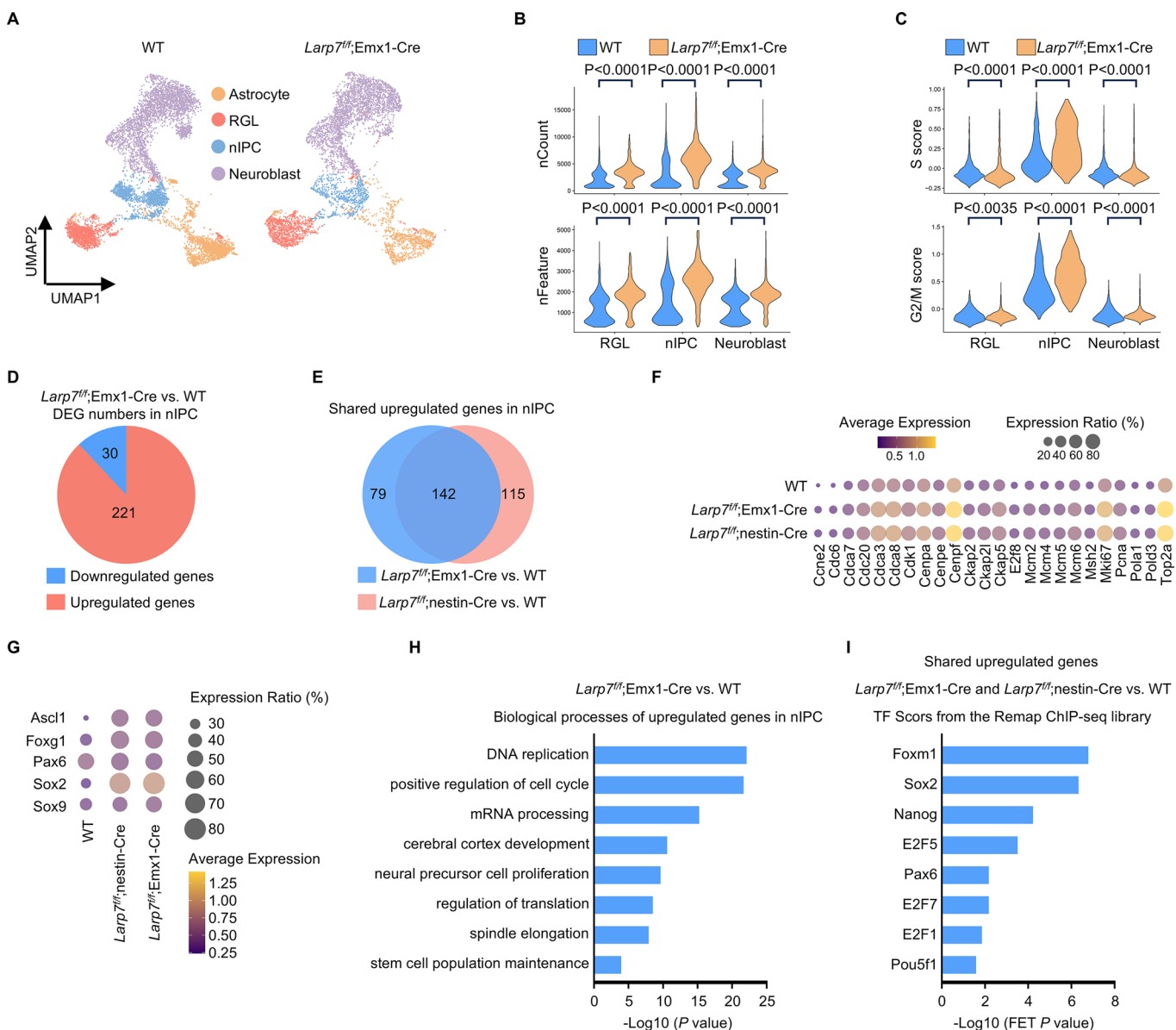

**Figure 5. Single-cell sequencing reveals shared cell-intrinsic defects in neuronal intermediate progenitor cells, regardless of global developmental delay.**

(A) Single-cell sequencing analysis of the microdissected dentate gyrus from wild-type (WT) and mutant (*Larp7^f/f*;Emx1-Cre) mice at P7. Four cell types, astrocyte, RGL, nIPC and neuroblast, were visualized by UMAP. RGL, radial glia-like. nIPC, neuronal intermediate progenitor cell. (B) Violin plot presenting the number of genes transcribed (upper panel) as well as the amount of mRNA (lower panel) per cell in RGLs (WT; n = 1457 cells; *Larp7^f/f*;Emx1-Cre; n = 1073 cells), nIPCs (WT; n = 1600 cells; *Larp7^f/f*; Emx1-Cre; n = 723 cells) and neuroblasts (WT; n = 3561 cells; *Larp7^f/f*; Emx1-Cre; n = 3082 cells). Of note, mutant nIPCs (*Larp7^f/f*;Emx1-Cre) exhibit the most prominent increase in the number of genes transcribed and the amount of mRNA per cell. Statistical significance was assessed using the two-sided Wilcoxon rank-sum test. (C) Violin plot presenting cell cycle scoring analysis in RGLs (WT; n = 1457 cells; *Larp7^f/f*;Emx1-Cre; n = 1073 cells), nIPCs (WT; n = 1600 cells; *Larp7^f/f*;Emx1-Cre; n = 723 cells) and neuroblasts (WT; n = 3561 cells; *Larp7^f/f*; Emx1-Cre; n = 3082 cells). Of note, among these cell types, only nIPCs are proliferative cells as indicated by high S and G2/M scores. Statistical significance was assessed using the two-sided Wilcoxon rank-sum test. (D) Pie chart showing the number of differentially expressed gene (DEG) in mutant (*Larp7^f/f*;Emx1-Cre) versus WT nIPCs. (E) Venn diagram showing upregulated genes in *Larp7^f/f*;Emx1-Cre and *Larp7^f/f*;nestin-Cre nIPCs, compared to WT nIPCs. There are 142 genes upregulated in both *Larp7^f/f*;Emx1-Cre and *Larp7^f/f*;nestin-Cre nIPCs. (F) Bubble plot showing the expression of key cell cycle genes in WT and mutant (*Larp7^f/f*;nestin-Cre and *Larp7^f/f*;Emx1-Cre) nIPCs. (G) Bubble plot presenting the expression of key self-renewal transcription factors in WT and mutant (*Larp7^f/f*;nestin-Cre and *Larp7^f/f*;Emx1-Cre) nIPCs. (H) GO analysis of the biological process of upregulated genes in mutant nIPC (*Larp7^f/f*;Emx1-Cre). Statistical significance was assessed using the two-sided Wilcoxon rank-sum test. (I) ChEA3 transcription factor (TF) analysis of genes upregulated in both *Larp7^f/f*;nestin-Cre and *Larp7^f/f*;Emx1-Cre nIPCs. Statistical significance was assessed using the two-sided Wilcoxon rank-sum test.

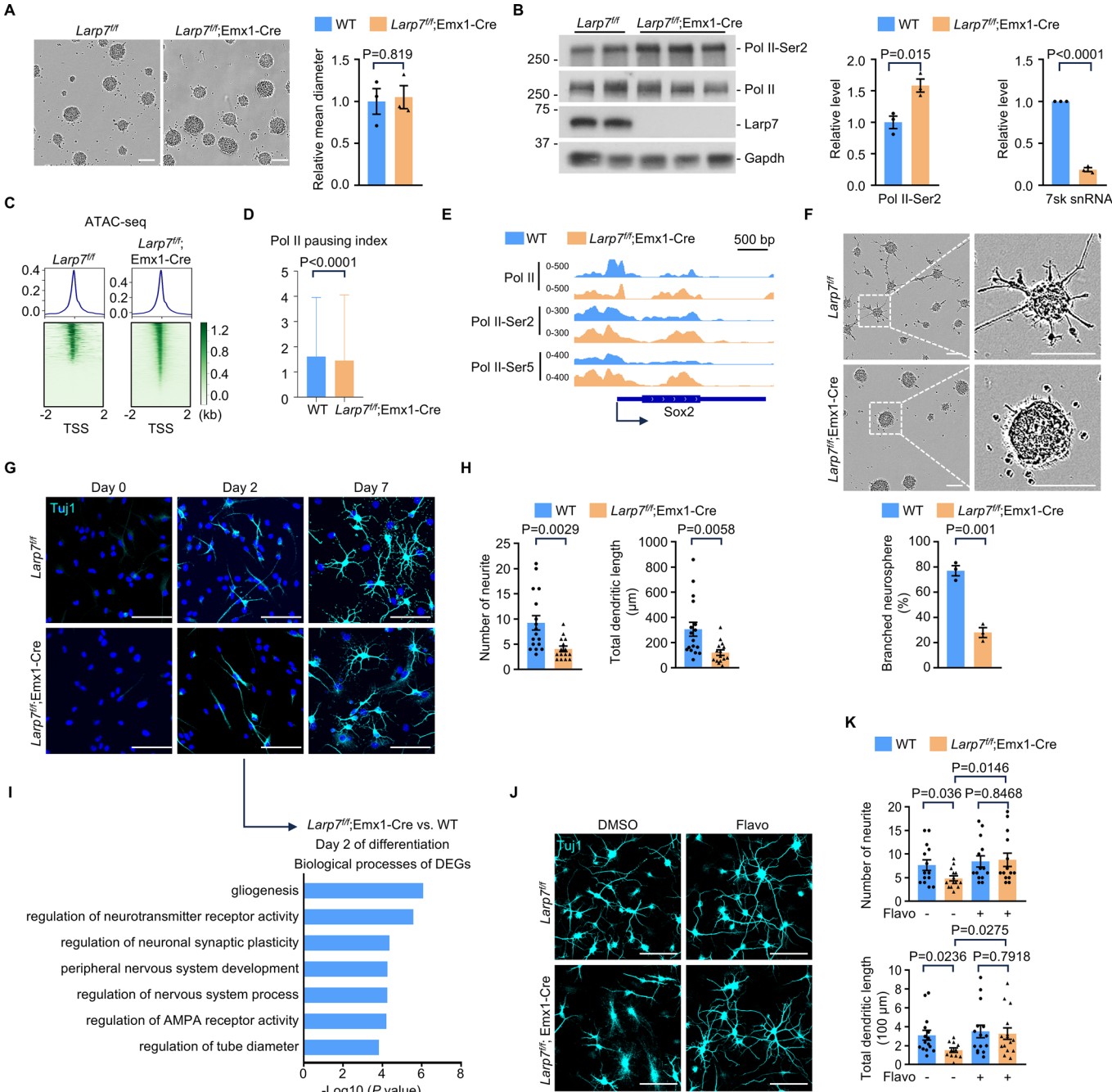

enhanced in these cells. To initiate neural differentiation, EGF and FGF were removed and serum was added in the basal culture condition (Day 0) (Hutton and Pevny, 2008). Immunofluorescence analysis of Tuj1, the marker of differentiated neurons, was carried out at Day 0 as well as after differentiation for 2 days (Day 2) and 7 days (Day 7) (Fig. 6G). At Day 2, Tuj1-positive, immature cells with underdeveloped neurites started to appear, indicating that neural differentiation is initiated. At this timepoint, both wild-type and *Larp7* knockout cells exhibited similar morphological features. At Day 7, Tuj1-positive wild-type cells developed highly branched structures. In contrast, Tuj1-positive knockout cells were significantly less mature, indicated by reduced number of neurites and

total dendritic length (Fig. 6H). In addition, transcriptomic profiling of early differentiation (Day 2) revealed that neurogenic pathways were downregulated in *Larp7* knockout cells (Fig. 6I). Importantly, adding flavopiridol, the small-molecule inhibitor of P-TEFb (Chao and Price, 2001), at a low concentration (3 nM) promoted the maturation of *Larp7* knockout cells, to a similar degree as wild-type cells (Fig. 6J,K).

To corroborate results observed in *Larp7^{f/f}*;Emx1-Cre neural and progenitor cells, the dentate gyrus was microdissected from P1 *Larp7^{f/f}*;nestin-Cre mice, and used to culture neural stem and progenitor cells in vitro. Similarly, these cells also exhibited enhanced self-renewal and compromised differentiation in vitro

◄

**Figure 6.  Loss of *Larp7* enhances self-renewal and compromises differentiation of neural stem and progenitor cells in vitro.**

(A) Phase contrast images of self-renewing neurospheres, derived from the dentate gyrus of P1 wild-type (*Larp7^{f/f}*) and mutant (*Larp7^{f/f}*;Emx1-Cre) mice, respectively. Representative results were shown (n = 4). Error bars, mean ± SEM. Statistical significance was assessed using the two-sided unpaired Student's *t* test. Scale bars: 100 μm. (B) Protein blot analysis (left panel) of indicated proteins and qPCR analysis of 7sk snRNA level (right panel) in wild-type (*Larp7^{f/f}*) and mutant (*Larp7^{f/f}*;Emx1-Cre) neurospheres. Representative results were shown (n = 3). Pol II, the largest subunit of RNA polymerase II, Rpb1. Pol II-Ser2, phosphorylation of Ser2 in the carboxyl-terminal domain of Rpb1. Error bars, mean ± SEM. Statistical significance was assessed using the two-sided unpaired Student's *t* test. (C) ATAC-seq heat maps and profiles of wild-type and mutant (*Larp7^{f/f}*;Emx1-Cre) neurospheres. Representative results were shown (n = 4). The *y* axis represents the average normalized signal density per genomic position, ranging from −2000 bp to +2000 bp from the transcription start site (TSS). (D) The pausing index (PI) (n = 18,318 genes) in wild-type (WT) and mutant (*Larp7^{f/f}*;Emx1-Cre) cells. Statistical significance was assessed using the two-sided Wilcoxon rank-sum test. (E) Browser shot depicting Pol II peaks within *Sox2* gene in wild-type (WT) and mutant (*Larp7^{f/f}*;Emx1-Cre) cells. (F) The self-renewal capacity of wild-type (*Larp7^{f/f}*) and mutant (*Larp7^{f/f}*;Emx1-Cre) neurospheres under sub-optimal self-renewal culture condition. Representative results were shown (n = 4). The percentage of neurospheres with a branched morphology, indicative of differentiation, was quantified. Error bars, mean ± SEM. Statistical significance was assessed using the two-sided unpaired Student's *t* test. Scale bars: 100 μm. (G) The differentiation capacity of wild-type (*Larp7^{f/f}*) and mutant (*Larp7^{f/f}*;Emx1-Cre) neurospheres. Representative images of differentiation for 0, 2, and 7 days were shown (n = 3). Tuj1 immunofluorescence was used to identify differentiated neurons, and to quantitate the number as well as the total length of neurites. Scale bars: 100 μm. (H) Quantification of the number and the total length of neurites at day 7 post-differentiation (Day 7 from (G)) (n = 3). Error bars, mean ± SEM. Statistical significance was assessed using the two-sided unpaired Student's *t* test. (I) GO analysis of the biological process of differentially expressed genes in mutant (*Larp7^{f/f}*;Emx1-Cre) versus wild-type (*Larp7^{f/f}*) after differentiation for 2 days (Day 2 from (G)). Statistical significance was assessed using the two-sided Wilcoxon rank-sum test. (J) Differentiation of neurospheres for 7 days with or without P-TEFb inhibitor (flavopiridol). DMSO is the solvent of flavopiridol (Flavo). Tuj1 immunofluorescence was used to identify differentiated neurons, and to quantitate the number as well as the total length of neurites. Representative results from three independent experiments were shown. Scale bars: 100 μm. (K) Quantification of the number and the total length of neurites from (J) (n = 3). Error bars, mean ± SEM. Statistical significance was assessed using the two-sided unpaired Student's *t* test. Source data are available online for this figure.

(Appendix Fig. S7). Taken together, we conclude that neural stem and progenitor cells in vitro, like their counterparts in vivo, exhibit enhanced self-renewal and compromised differentiation, when *Larp7* is genetically ablated. These results support the notion that *Larp7* loss causes dysregulated stem cell dynamics in a cell-autonomous, P-TEFb-dependent manner.

## Disruption of the inhibitory 7sk snRNP complex causes dentate gyrus hypoplasia

To further corroborate the notion that enhanced P-TEFb activity indeed underlies dentate gyrus hypoplasia, we investigated whether knockout of other components of 7SK snRNP would phenocopy LARP7 loss. The majority of cellular P-TEFb is sequestered and inactivated by 7SK snRNP, containing 7SK snRNA, LARP7, and HEXIM1 or HEXIM2 (HEXIM1/2). Loss of LARP7 protein disrupts 7SK snRNP, thus releases and activates P-TEFb. So far, we have shown that loss of LARP7 enhances mRNA production at the single-cell level, consistent with elevated P-TEFb activity per cell. Of note, LARP7 knockout largely eliminates 7SK snRNA in cells. One may argue that *Larp7* loss-induced dentate gyrus hypoplasia might be caused by 7SK snRNA-dependent, but P-TEFb-independent mechanisms. Like LARP7 knockout, knockout of HEXIM1/2 also releases P-TEFb from 7SK snRNP. However, HEXIM1/2 knockout does not affect cellular 7SK snRNA level because LARP7 protein is unaltered. Therefore, knockout of HEXIM1/2 provides an opportunity to address the possibility that *Larp7* loss-induced dentate gyrus hypoplasia might be caused by some unknown 7SK snRNA-dependent, but P-TEFb-independent mechanisms (Fig. 7A).

We examined whether *Hexim1* or *Hexim2* is expressed in the dentate gyrus. Microdissected hippocampi were subjected to transcriptomic profiling by RNA-seq. The expression of *Hexim1*, but not *Hexim2*, could be easily detected (Fig. 7B). In our single-cell sequencing datasets, *Hexim1* was expressed at a higher level than *Hexim2* in nIPCs in vivo (Fig. 7C), as well as their counterparts in vitro (Fig. 7D). A previous study has shown that in HeLa cells the activation of P-TEFb increases HEXIM1 mRNA level (Liu et al,

2014). We found that *Hexim1* mRNA expression is also moderately increased in *Larp7* knockout cells (Fig. 7B,D), consistent with enhanced P-TEFb activity in these cells. Furthermore, we queried published single-cell sequencing datasets of the dentate gyrus, spanning from E16.5 to P132 (Hochgerner et al, 2018). Again, *Hexim1* is expressed at higher levels than *Hexim2* in nIPCs at E16.5, P0 and P5 (Fig. 7E). Thus, we generated *Hexim1*-floxed mice (*Hexim1^{f/f}*; Fig. 7F), and crossed *Hexim1^{f/f}* mice with Emx1-Cre mice to delete *Hexim1* in Emx1+ neural stem and progenitor cells (*Hexim1^{f/f}*;Emx1-Cre). Remarkably, *Hexim1^{f/f}*;Emx1-Cre mice also exhibited dentate gyrus hypoplasia (Fig. 7G,H). In addition, these mice exhibited cognitive deficits in behavioral tests (Fig. 7I–M), similar to *Larp7^{f/f}*;nestin-Cre and *Larp7^{f/f}*;Emx1-Cre mice. Neural stem and progenitor cells were derived from microdissected P1 *Hexim1^{f/f}*;Emx1-Cre dentate gyrus. Under optimal self-renewal culture condition, both wild-type and *Hexim1* knockout neurospheres contain densely packed cells with a smooth perimeter (Fig. 7N). We confirmed that the expression levels of 7sk snRNA, Cdk9, and Larp7 were comparable in wild-type and *Hexim1* knockout neural stem and progenitor cells. In contrast, in *Larp7* knockout cells, 7sk snRNA expression was markedly decreased, whereas the expression levels of Cdk9 and Hexim1 were unaltered (Fig. 7O,P). Consistent with elevated P-TEFb activity, CUT&Tag assay revealed that the pausing index of RNA polymerase II was decreased (Fig. 7Q), and more active RNA polymerase II peaks were detected in the gene body regions in *Hexim1* knockout cells than wild-type cells (Fig. 7R). Under sub-optimal culture condition, the morphology of *Hexim1* knockout neurospheres was minimally affected, whereas wild-type neurospheres started to exhibit multiple branch-like structures extending from the core (Fig. 7S). Compared to wild-type cells, differentiation of *Hexim1* knockout cells was compromised, and this defect could be alleviated by P-TEFb inhibitor (Fig. 7T). Thus, *Hexim1* knockout and *Larp7* knockout share many phenotypic similarities in mice as well as in neural stem and progenitor cells. We conclude that the disruption of the 7SK snRNP complex, thus the activation of P-TEFb, causes hypoplastic dentate gyrus.

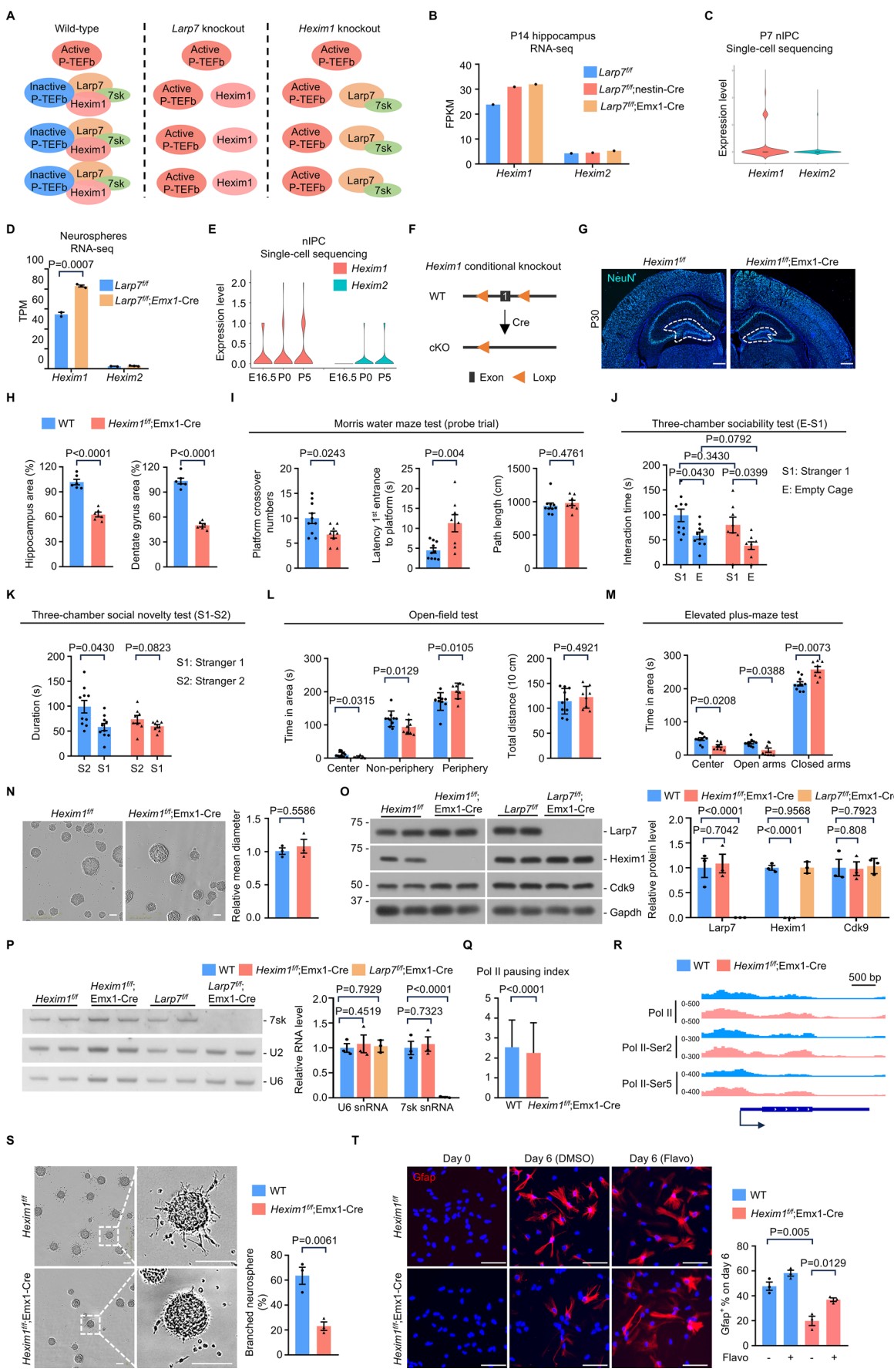

**Figure 7. Disruption of the inhibitory 7sk snRNP complex causes dentate gyrus hypoplasia.**

(A) Schematic presentation of inactive and active P-TEFb in wild-type and knockout cells (see text). (B) *Hexim1* and *Hexim2* expression in the hippocampus at P14, measured by RNA-seq. (C) Violin plot presenting the expression of *Hexim1* and *Hexim2* in P7 nIPCs ($n = 1600$ cells), measured by single-cell RNA sequencing. (D) *Hexim1* and *Hexim2* expression in neurospheres derived from P1 dentate gyrus of wild-type (WT; $n = 2$) and knockout (*Larp7^{f/f}*;Emx1-Cre; $n = 3$) mice, measured by RNA-seq. Statistical significance was assessed using the two-sided unpaired Student's $t$ test. (E) Violin plot presenting the expression of *Hexim1* and *Hexim2* in nIPCs at E16.5 ($n = 51$ cells), P0 ($n = 60$ cells) and P5 ($n = 120$ cells). Expression data were extracted from the re-analysis of single-cell RNA-seq datasets (GSE95753). (F) The strategy to generate conditional *Hexim1* knockout mice. In conditional knockout mice (cKO), the exon 1 of *Hexim1* is deleted. WT, wild-type. (G) Representative images showing coronal sections of the brain from wild-type (*Hexim1^{f/f}*; $n = 6$) and knockout (*Hexim1^{f/f}*;Emx1-Cre; $n = 6$) mice at P30. The dentate gyrus was demarcated by white-dashed lines. Scale bar: 500 µm. (H) The size of the hippocampus (left panel) and the dentate gyrus (right panel) in P30 wild-type (WT; $n = 6$) and knockout (*Hexim1^{f/f}*;Emx1-Cre; $n = 6$) mice. Error bars, mean ± SEM. Statistical significance was assessed using the two-sided unpaired Student's $t$ test. (I) Morris water maze test. During the probe trial (platform removed), wild-type (WT; $n = 10$) and knockout (*Hexim1^{f/f}*;Emx1-Cre; $n = 8$) mice were tested. The number of platform crossing, latency to find the location of removed platform, and the total path length were documented by SMART video tracking system. Error bars, mean ± SEM. Statistical significance was assessed using the two-sided unpaired Student's $t$ test. (J) Three-chamber sociability test. Wild-type (WT; $n = 10$) and knockout (*Hexim1^{f/f}*;Emx1-Cre; $n = 8$) mice were tested. Total sniffing time spent towards empty chamber (E) or stranger 1 (S1) was documented by SMART video tracking system. Error bars, mean ± SEM. Statistical significance was assessed using the two-sided unpaired Student's $t$ test. (K) Three-chamber social novelty test. Wild-type (WT, $n = 10$) and knockout (*Hexim1^{f/f}*;Emx1-Cre; $n = 8$) mice were tested. Total sniffing time spent toward stranger 1 (S1) or stranger 2 (S2) were documented by SMART video tracking system. Error bars, mean ± SEM. Statistical significance was assessed using the two-sided unpaired Student's $t$ test. (L) Open field test. Wild-type (WT; $n = 10$) and knockout (*Hexim1^{f/f}*;Emx1-Cre; $n = 8$) mice were tested. The time spent in the indicated areas, including Center, Non-periphery and Periphery, and total distance were documented by SMART video tracking system. Error bars, mean ± SEM. Statistical significance was assessed using the two-sided unpaired Student's $t$ test. (M) Elevated plus maze test. Wild-type (WT; $n = 10$) and knockout (*Hexim1^{f/f}*;Emx1-Cre; $n = 8$) mice were tested. The time spent in the indicated areas, including Center, Open arms, and Closed arms, was documented by SMART video tracking system. Error bars, mean ± SEM. Statistical significance was assessed using two-sided unpaired Student's $t$ test. (N) Phase contrast images of self-renewing neurospheres, derived from the dentate gyrus of P1 wild-type (*Hexim1^{f/f}*) and knockout (*Hexim1^{f/f}*;Emx1-Cre) mice, respectively. Representative results were shown ($n = 3$). Error bars, mean ± SEM. Statistical significance was assessed using the two-sided unpaired Student's $t$ test. Scale bars: 100 µm. (O) Protein blot analyses of indicated proteins from wild-type and mutant neurospheres. Representative results were shown ($n = 3$). Error bars, mean ± SEM. Statistical significance was assessed using the two-sided unpaired Student's $t$ test. (P) RNA blot analyses of indicated RNA from wild-type and mutant neurospheres. Representative results were shown ($n = 3$). Error bars, mean ± SEM. Statistical significance was assessed using the two-sided unpaired Student's $t$ test. (Q) The pausing index (PI) ($n = 15,873$ genes) in wild-type and knockout (*Hexim1^{f/f}*;Emx1-Cre) neural stem and progenitor cells. Statistical significance was assessed using the two-sided Wilcoxon rank-sum test. (R) Browser shot depicting Pol II peaks within *Sox2* gene in wild-type and knockout (*Hexim1^{f/f}*;Emx1-Cre) neural stem and progenitor cells. (S) The self-renewal capacity of wild-type and knockout (*Hexim1^{f/f}*; Emx1-Cre) neurospheres under sub-optimal self-renewal culture condition. Representative results were shown ($n = 4$). The percentage of neurospheres with a branched morphology, indicative of differentiation, was quantified. Error bars, mean ± SEM. Statistical significance was assessed using the two-sided unpaired Student's $t$ test. Scale bars: 100 µm. (T) Differentiation of neurospheres for 6 days with or without P-TEFb inhibitor (flavopiridol). DMSO is the solvent of flavopiridol (Flavo). Representative images were shown (left panel) ($n = 3$). Gfap immunofluorescence was used to quantify differentiated cells (right panel). Error bars, mean ± SEM. Statistical significance was assessed using the two-sided unpaired Student's $t$ test. Scale bars: 100 µm. Source data are available online for this figure.

## Discussion

The present study provides an explanation for a long-standing conundrum. Higher P-TEFb activity is thought to promote cellular growth and proliferation by increasing RNA polymerase II transcriptional out. Paradoxically, human genetics studies have repeatedly linked inactivating variants of *LARP7*, a ubiquitous negative regulator of P-TEFb, to Alazami syndrome, a human neurodevelopmental disorder characterized by severe growth restriction and impaired cognitive functions. The majority of cellular P-TEFb is sequestered and inhibited by 7SK snRNP, containing 7SK snRNA, LARP7 and HEXIM1 protein. In this study, we use three genetically engineered mouse models to show that knockout of either *Larp7* or *Hexim1* results in hippocampal dentate gyrus hypoplasia accompanied by cognitive deficits. Interestingly, knockout of *Larp7* by nestin-Cre (*Larp7^{f/f}*;nestin-Cre) exhibits not only dentate gyrus hypoplasia, but also global developmental delay. These phenotypes closely mimic the salient features of patients afflicted with Alazami syndrome (Appendix Fig. S8). It is worth noting that the reduction of the body size of *Larp7^{f/f}*;nestin-Cre mice seems less severe than that in Alazami syndrome patients. One potential explanation is that nestin-Cre only functions in selected organs in mice. In contrast, human patients carry germline inactivating variants of *LARP7*, that is, the function of LARP7 protein is lost in all cells. Importantly, we show that knockout of either *Larp7* or *Hexim1* by Emx1-Cre (*Larp7^{f/f}*;Emx1-Cre and *Hexim1^{f/f}*;Emx1-Cre) phenocopies dentate gyrus hypoplasia in *Larp7^{f/f}*;nestin-Cre mice, but bears no signs of global developmental delay. These observations demonstrate that dentate gyrus hypoplasia is phenotypically separated from global developmental delay. To our

knowledge, these genetically engineered mouse models are the first to provide experimental support that *LAPR7* inactivation is indeed the underlying etiology for Alazami syndrome. Of note, one prominent feature of Alazami syndrome patients is microcephaly. Damaging variants in primary microcephaly genes, such as those encoding centrosome proteins, DNA replication and repair factors, can cause the imbalance between symmetric and asymmetric division of neural progenitor cells. This defect is proposed to underlie the etiology of primary microcephaly (Phan and Holland, 2021). In a broad sense, our results are consistent with this notion in that *Larp7* deficiency also causes unbalanced differentiation of neural progenitor cells.

How does enhanced P-TEFb activity, supposedly increasing cellular proliferation and growth, results in tissue hypoplasias? Our single-cell analysis reveals that loss of *Larp7* indeed leads to higher P-TEFb activity and greater RNA polymerase II transcriptional output per cell, especially in nIPCs, the progenitor cell forming the dentate gyrus. Importantly, many genes are upregulated in both *Larp7^{f/f}*;nestin-Cre and *Larp7^{f/f}*;Emx1-Cre nIPCs, demonstrating the cell-intrinsic nature of the transcriptional defects regardless of the global developmental delay. These genes are enriched in two categories. The first group contains genes essential for cell-cycle progression regulated by key transcription factors such as E2F and FOXM1 (Fischer et al, 2022). The second group contains genes regulated by core transcription factors, such as SOX2, that maintain the multipotency of neural stem and progenitor cells (Graham et al, 2003). These transcriptomic profiles suggest that the self-renewal of nIPCs is enhanced. Results from both marker analyses by immunofluorescence and neurosphere assays further corroborate enhanced self-renewal, and reveal a

concomitant reduction in nIPC differentiation to generate mature granule cells. Compromised differentiation is likely caused by the ability of self-renewal transcription factors to suppress neuronal differentiation (Graham et al, 2003). Because the number of granule cells determines the final size of the dentate gyrus, these observations reconcile increased cellular proliferation with dentate gyrus hypoplasia. We suspect that increased P-TEFb activity may in general cause organ hypoplasias. This hypothesis is supported by the fact that *Larp7^{f/f}*;nestin-Cre mice exhibit global development delay, likely because the recombination activity of nestin-Cre is present in stem and progenitor cells of several organs. One important unanswered question is whether human neural stem and progenitor cells with *LARP7* deficiency would exhibit similar phenotypic and transcriptomic defects as their murine counterparts. To address this issue, primary cells derived from Alazami syndrome patients could be used to generate iPS cells, and then differentiated into neural stem cells or brain organoids.

Our results suggest potential therapeutic interventions for several medical conditions associated with shrunken dentate gyri, such as aging. Intriguingly, the transcription elongation speed of RNA polymerase II is increased during aging, and this phenomenon is evolutionarily conserved from worms to humans to influence organismal longevity (Debes et al, 2023). Considering the facts that nIPCs during perinatal period and adult neurogenesis are highly similar (Hochgerner et al, 2018) and P-TEFb is the major promoter of transcription elongation speed (Fujinaga et al, 2023), it will be of great interest to dissect whether aging-related size reduction of the dentate gyrus is due to elevated P-TEFb activity. The safety of several P-TEFb inhibitors has been proven in clinical trials for cancer (Fujinaga, 2020). These clinical-grade reagents may provide translational opportunities to intervene with aging-related cognitive decline associated with shrunken dentate gyrus.

Our studies suggest that P-TEFb is a reinforcer of master transcription factors to determine the cell fate in a context-dependent manner, rather than a factor to simply promote cell proliferation as previously thought. Recently, we have shown that chemotherapy-induced P-TEFb activation is required for cell death in embryonic stem cells (Fang et al, 2022). Because cell death is entirely dependent on the transcriptional activity of p53 protein in this cell type (He et al, 2016), the inhibition of P-TEFb blocks the induction of p53 transcriptional targets and thus block chemotherapy-induced cell death (Fang et al, 2022). The present study highlights that the subtissular stem cell dynamics determines the final phenotypic presentation of dysregulated transcription elongation. It is well-established that transcriptional programs of differentiation are repressed by self-renewal transcription factors in stem and progenitor cells (Takahashi and Yamanaka, 2016; Young, 2011). Thus, enhanced P-TEFb activity in these cell types reinforce the transcriptional network of self-renewal, that is, proliferation in an undifferentiated cellular state. On the other hand, insufficient P-TEFb activity may also cause developmental defects in humans (Maddirevula et al, 2019; Shaheen et al, 2016). Precise regulation of gene expression is required for proper development, and dysregulated transcription underlies many human diseases. The control of RNA polymerase II elongation, by P-TEFb and other transcription elongation factors, is the main mechanism to regulate eukaryotic gene expression. Yet, the physiological and pathological roles of positive and negative transcription elongation factors remain poorly characterized. Even less known is how cells limit enhanced

P-TEFb activity in vivo. We find that *Hexim1* is transcriptionally upregulated in *Larp7* knockout neural progenitor cells, potentially reflecting a cellular compensational mechanism. Further in vivo studies are needed to delineate the context-dependent functions of P-TEFb, and to fulfill the translational potential of emerging therapeutic P-TEFb inhibitors.

# Methods

### Reagents and tools table

| Reagent/resource | Reference or source | Identifier or catalog number |
|---|---|---|
| **Experimental models** | | |
| *Larp7^{f/f}* | This paper | |
| *Hexim1^{f/f}* | This paper | |
| nestin-Cre (B6.Cg-Tg (Nes-cre) 1Kln/J) | Jackson Lab, #003771 | |
| Emx1-Cre (B6.129S2-Emx1^{tm1(cre)Krj}/J) | Jackson Lab, #005628 | |
| **Recombinant DNA** | | |
| **Antibodies** | | |
| Anti-Sox2 | | R&D, AF2018 |
| Anti-Gapdh | | KangChen, KC-5G5 |
| Anti-Dcx | | Millipore, SAB2501666 |
| Anti-Tbr2 | | Abcam, ab23345 |
| Anti-Tbr1 | | Abcam, ab183032 |
| Anti-Pax6 | | Proteintech, 12323-1-AP |
| Anti-Ki67 | | Invitrogen, MA5-14520 |
| Anti-Larp7 | | Abcam, ab134757 |
| Anti-NeuN | | CST, 24307S |
| Anti-Tuj1 | | Abcam, ab18207 |
| Anti-Prox1 | | Invitrogen, P21936 |
| Anti-BrdU | | Invitrogen, B35128 |
| Anti-Pol II | | Abcam, ab817 |
| Anti-Pol II-Ser2 | | Abcam, ab5095 |
| Anti-Pol II-Ser5 | | Abcam, ab5131 |
| Anti-mouse IgG-HRP | | CST, 7076S |
| Anti-rabbit IgG-HRP | | CST, 7074S |
| Alexa Fluor 488 AffiniPure Donkey Anti-Rabbit IgG | | Jackson Immunoresearch, 711-545-152 |
| CyTM3 AfriniPure Donkey Anti-Mouse IgG | | Jackson Immunoresearch, 715-165-150 |
| Alexa Fluor 488 AffiniPure Donkey Anti-Mouse IgG | | Jackson Immunoresearch, 715-545-150 |
| CyTM3 AffiniPure Donkey Anti-Goat IgG | | Jackson Immunoresearch, 705-165-147 |

| Reagent/resource | Reference or source | Identifier or catalog number |
|---|---|---|
| **Oligonucleotides and other sequence-based reagents** | | |
| U6 probe: GGAACGCTTCACGAA TTTGCGTGTCATCCTTGCGCA GGGGCCA/iBiodT/gc/iBiodT/A | | |
| Sn7sk probe: CGGGGAAGGTCGTCCTC/ iBiodT/iBiodT/C | | |
| U2 probe: GAGCAAGCTCCTATT CCAACTCCTACTTCCAAAAA iBiodT/iBiodT/ | | |
| **Chemicals, enzymes and other reagents** | | |
| Flavopiridol | | MCE, HY-10005 |
| DMEM/F12 | | Gibico, 11320-033 |
| N2 | | 17502048 |
| B27 | | 17504044 |
| EGF | | Sino Biological, 10605-HNAE |
| bFGF | | Sino Biological, 10014-HNAE |
| 5-BrdU | | Sigma, B5002 |
| Accutase | | Gibico, A1110501 |
| qPCRmix (ssofast evagreen@ supermix) | | Bio-Rad, 1725201 |
| Fluoromount-G® | | SouthernBiotech, 0100-01 |
| Trizol | | Invitrogen, 15596018 |
| Matrigel | | Corning, 356237 |
| **Software** | | |
| CellRanger (v6.1.2) | PMID: 28091601 | |
| Seurat (v4.3) | PMID: 25867923 | |
| Harmony (v0.1.0) | PMID: 31740819 | |
| Monocle 3 | PMID: 30787437 | |
| ChEA3 | PMID: 31066453 | |
| clusterProfiler (v4.10) | PMID: 22455463 | |
| Minimap2 (v2.24) | PMID: 29750242 | |
| FastQC (v0.11.9) | Andrews, 2010 | |
| Trimmomatic (v0.39) | PMID: 24695404 | |
| Bowtie2 (v2.4.5) | PMID: 22388286 | |
| SAMtools (v1.15) | PMID: 19505943 | |
| BEDTools (v2.30.0) | PMID: 20110278 | |
| MACS2 (v2.2.7.1) | PMID: 18798982 | |
| ChIPseeker (v1.34.1) | PMID: 25765347 | |
| deepTools (v3.5.1) | PMID: 24799436 | |
| R (v4.2.2) | Ihaka R and Gentleman R, 1996 | |
| Bioconductor (v3.16) | PMID: 15461798 | |

| Reagent/resource | Reference or source | Identifier or catalog number |
|---|---|---|
| **Other** | | |
| High-Sensitivity Open Chromatin Profile Kit 2.0 (for Illumina) | | Novoprotein, N248 |
| CUT&Tag 4.0 High-Sensitivity Kit (for Illumina) | | Novoprotein, N259 |

## Generation of conditional knockout mouse models

To generate floxed mice, two *loxp* sequences were inserted into the introns flanking exon 9 of *Larp7* (*Larp7*$^{f/f}$) and exon 1 of *Hexim1* (*Hexim1*$^{f/f}$), respectively. To obtain conditional knockout mice, floxed mice were crossed with nestin-Cre (B6.Cg-Tg (Nes-cre)1Kln/J; Jackson Lab, #003771) or Emx1-Cre (B6.129S2-Emx1$^{tm1(cre)Krj}$/J; Jackson Lab, #005628) mice, respectively. Noon on the day of the vaginal plug was defined as E0.5. Embryos and offsprings were genotyped by genomic PCR. Knockout efficiencies were further validated by protein blotting. Mice were housed under a 12-h light-dark cycle, and had *ad libitum* access to food and water in a controlled animal facility. Animal studies were approved by the Institutional Animal Care and Use Committee at West China Second University Hospital of Sichuan University (2022-054).

## Tissue analysis

For brain tissues, samples were fixed with 4% paraformaldehyde overnight at 4 °C, cryopreserved in 25% sucrose for 1 day, and then embedded in Tissue-Tek O.C.T. Compound (Sakura Finetek, USA). Tissue blocks were sectioned on a cryostat at 8–10 μm, and then stained with indicated antibodies. Images were obtained using a fluorescent confocal microscope (FV3000; Olympus, Japan). For the analysis of tissue architecture, Hematoxylin and eosin (HE) and Nissl staining were carried out. HE staining procedures were performed as previously described (Cardiff et al, 2014). Images were obtained using Pannoramic MIDI digital slide scanner (3DHISTECH). Nissl staining was performed following the manufacturer's instructions (Beyotime, C0117), and the image was documented by microscope (DMI8; Leica, Germany). To measure hippocampal and dentate gyrus areas, wild-type and mutant brains were sectioned consecutively from the Bregma at 10 μm intervals. Sections were compared to the Allen Mouse Brain Atlas to select comparable positions in wild-type and mutant brains. For quantification, three sections each from 6 mice of indicated genotypes were selected, and added up to represent the size of the hippocampal or dentate gyrus area, including *Larp7*-floxed ($n = 6$), *Larp7*$^{f/f}$;nestin-Cre ($n = 6$), *Larp7*$^{f/f}$;Emx1-Cre ($n = 6$), *Hexim1*-floxed ($n = 6$) and *Hexim1*$^{f/f}$;Emx1-Cre ($n = 6$), respectively.

## Behavioral testing

Behavioral analyses were carried out as previously described (Rein et al, 2020; Vorhees and Williams, 2006; Walf and Frye, 2007). Briefly, 8- to 12-week-old wild-type (WT), *Larp7*$^{f/f}$;nestin-Cre and *Larp7*$^{f/f}$;Emx1-Cre mice with equal sex ratio were placed in the test

room one hour prior to each test. Wild-type and *Larp7^{f/f}*;nestin-Cre as well as wild-type and *Larp7^{f/f}*;Emx1-Cre mice were evaluated on the same day, and equipment was cleaned with 70% ethanol after each animal. Tests were performed starting with the least aversive to most aversive task, scored automatically and analyzed by SMART 3.0 (Panlab). For the open field test, the time spent in three areas were determined, defined as the "Periphery (5 cm from the walls)", "Center (a 14 cm×14 cm square in the center)" and "Non-periphery" (areas excluding Periphery and Center). For three-chamber sociability and social novelty test, active interactions between the mice were scored with 10 min of total test time. For elevated plus maze test, mice were placed in the center of a black elevated plus maze (each arm is 30 cm long and 6 cm wide with two arms enclosed by 15 cm-high walls), and allowed to explore for 5 min in a dimly lit room. Then the time spans spent in the open, closed arms and in the middle were determined. For Morris water maze test, during a 5-day training phase, mice were trained to locate the escape platform (1 cm below the water surface) in a tank filled with water colored with powdered milk (Appendix Fig. S9). For the probe trial, the escape platform was removed on day 7. Each mouse was allowed to swim for 1 min to determine whether the animal remembered the location of the platform. "Latency 1st entrance to platform" is defined by the time taken by the mouse to reach the location of removed platform for the first time. "Platform crossover numbers" is defined by how many times the animal crosses the location of removed platform. These two parameters were regarded as indicators of how well the animal has learned the spatial location of the platform. "Path length" is defined by the overall swimming distance of each mouse. Wild-type and mutant mice exhibited comparable speed as they swam similar overall distance within 1 min. For the probe trial, the time spent in the target quadrant was also documented (Appendix Fig. S9).

### Neurosphere assay

Neurospheres containing neural stem and progenitor cells (NPCs) were derived using microdissected dentate gyrus from P1 wild-type, *Larp7^{f/f}*;nestin-Cre, *Larp7^{f/f}*;Emx1-Cre and *Hexim1^{f/f}*;Emx1-Cre mice, respectively. Briefly, P1 dentate gyrus was dissociated by accutase into single cell, and cultured in N2B27 medium containing bFGF (20 ng/ml) and EGF (20 ng/ml) to generate neurospheres as previously described (Soares et al, 2020). The status of the neurosphere was routinely monitored by Incucyte S3 live-cell analysis system (Sartorius). To examine the self-renewal capacity of wild-type, *Larp7* or *Hexim1* knockout cells, neurospheres were cultured in N2B27 medium containing bFGF (2 ng/ml) and EGF (2 ng/ml) for 4 days, and documented by Incucyte S3 live-cell analysis system. Neurosphere differentiation was carried out as previously described (Hutton and Pevny, 2008). Briefly, neurospheres were dissociated into single cells, replated onto Matrigel-treated plates, cultured in N2B27 medium containing bFGF (20 ng/ml) for 2 days, and followed by culturing in N2B27 medium containing 2% FBS for 4–5 days. To measure the degree of differentiation, the percentage of Gfap-positive cells was quantified. Alternatively, the number of Tuj1-positive neurites of individual neuron were quantified, and the total dendritic length of neurites per neuron was calculated using ImageJ's Simple Neurite Tracer plugin (Longair et al, 2011).

### Single-cell sequencing, RNA-seq and bioinformatic analyses

Single-cell RNA-seq was used to analyze cell compositions and functional states of the dentate gyrus, obtained from microdissected wild-type (P7 and P14), *Larp7^{f/f}*;Emx1-Cre (P7 and P14), and *Larp7^{f/f}*;nestin-Cre mutant mice (P7), respectively. To obtain enough materials, 3–4 dentate gyri from littermates were combined for each genotype. The dentate gyrus was dissociated into single-cell suspension, loaded into Chromium microfluidic chips with 3′ reagent kit (v3.1 chemistry), and barcoded with a 10X Chromium Controller (10X Genomics). Sequencing was performed using Illumina NovaSeq platform following manufacturer's instructions (Annoroad Gene Technology (Beijing) Co., Ltd.). The bioinformatics pipeline was previously described (Li et al, 2023). After quality control, 12,820 wild-type (P7), 6,775 *Larp7^{f/f}*;Emx1-Cre (P7), 7,153 *Larp7^{f/f}*;nestin-Cre (P7), 9,617 wild-type (P14), and 10,930 *Larp7^{f/f}*;Emx1-Cre (P14) cells were used for downstream bioinformatic analyses. Briefly, the sequencing data were aligned to the mouse transcriptome to generate the expression matrix using the CellRanger (v6.1.2). Cells with detected gene numbers above 500, and a percentage of mitochondrial genes below 15% were used for unbiased cell clustering by Seurat (v4.3). Batch effects were corrected using Harmony package v0.1.0 in UMAP visualization. The putative identities of each cluster were assigned using pre-defined markers (Hochgerner et al, 2018). Cell differentiation trajectories were inferred by Monocle 3 (Cao et al, 2019). Transcription factor enrichment analysis using ChEA3 was performed as previously described (Keenan et al, 2019). Differentially expressed genes were further subjected to Gene Ontology (GO) enrichment analysis using clusterProfiler (v4.10) in the R package. Enriched GO terms were further grouped based on similarity using Simplify in clusterProfiler (v4.10) (Wu et al, 2021). To identify differentially regulated pathways during differentiation, RNA-seq experiments were carried out. Minimap2 (v2.24) (Li, 2018) was used to align reads with the ENSEMBL (v102) mouse reference transcriptome.

### ATAC-seq, CUT&Tag, and data analyses

Chromatin accessibility profiles were analyzed by transposase-accessible chromatin using sequencing (ATAC-seq) (Grandi et al, 2022). RNA polymerase II binding sites were determined by Cleavage Under Targets and Tagmentation (CUT&Tag) (Kaya-Okur et al, 2019). ATAC-seq libraries were constructed by High-Sensitivity Open Chromatin Profile Kit 2.0 (for Illumina) (Novoprotein, N248), following the manufacturer's instructions. Briefly, neurospheres were dissociated by accutase, resuspended in ATAC lysis buffer supplemented with 0.1% NP40, 0.1% Tween 20 and 0.01% digitonin, incubated for 5 min on ice, and then centrifuged for 5 min at 500 g at 4 °C. Nuclei were incubated in tagmentation buffer for 30 min at 37 °C. Tagmented DNA was collected using Tagment DNA extract beads and amplified by PCR. The DNA libraries were purified using DNA clean beads and sequenced by Illumina Nova-seq 6000 platform with pair-end reads of 150 bp. CUT&Tag-seq libraries were constructed by CUT&Tag 4.0 High-Sensitivity Kit (for Illumina) (Novoprotein, N259). Briefly, neurospheres were dissociated by Accutase. Cells were enriched by ConA beads, and incubated with primary antibody

buffer supplemented with anti-Pol II antibody (Abcam, ab817), anti-Pol II-Ser2 antibody (Abcam, ab5095), and anti-Pol II-Ser5 antibody (Abcam, ab5131), respectively. ConA beads were then incubated with the secondary antibody buffer containing anti-mouse IgG antibody (Novoprotein, N270) for anti-Pol II-Ser5 antibody, or anti-rabbit IgG antibody (Novoprotein, N269) for anti-Pol II antibody and anti-Pol II-Ser2 antibody, respectively. After wash, ConA beads were incubated with protein A/G-Tn5 transposome. The DNA fragments were purified, amplified, and sequenced by Illumina Nova-seq 6000 platform with pair-end reads of 150 bp. Quality control of raw FASTQ files was carried out by FastQC (v0.11.9) and adapter trimming with Trimmomatic (v0.39), using parameters: TRAILING:20, SLIDINGWINDOW:4:20, MIN-LEN:36. Trimmed reads were then aligned to the mm10 reference genome using Bowtie2 (v2.4.5) with default parameters, and SAM files were converted to sorted BAM files by SAMtools (v1.15). Data quality was ensured by FRiP scores calculated by BEDTools (v2.30.0) (FriP>30%). Accessible chromatin regions and polymerase II binding regions were identified by MACS2 (v2.2.7.1), using parameters (--shift -75 --extsize -150 --nomodel -B --SPMR -g mm). Called peaks were annotated using ChIPseeker (v1.34.1) with TxDb.Mmusculus.UCSC.mm10.knownGene database. Genomic features were classified into promoter ($-2000$ bp to $+2000$ bp from the TSS), gene body, intergenic, and flanking regions. BAM files were converted to BigWig format using BamCoverage from deepTools (version 3.5.1), using parameters (--binSize 10 --normal-izeUsing RPKM --minMappingQuality 30). Signal densities around the transcription start site (TSS) were quantified using deepTools (v3.5.1). Signal aggregation was performed using computeMatrix reference-point centered on TSS coordinates ($-2000$ bp to $+2000$ bp) from the mm10 reference genome annotation. Mito-chondrial and blacklisted regions were excluded during matrix generation (--skipZeros). Three complementary analyses were performed to visualize the genomic localization and enrichment patterns of accessible chromatin regions and polymerase II binding regions, including annotation distribution (plotAnnoBar), TSS proximity profiles (plotDistToTSS), and average signal profiles at promoters (plotAvgProf) using tag matrices ($-2000$ bp to $+2000$ bp from the TSS). Pausing index (PI) was calculated using CUT&Tag profiling of RNA polymerase II, that is, the ratio of the Pol II signal within the promoter-proximal region ($-50$ bp to $+300$ bp from the TSS) to the Pol II signal within the gene body region (from the TSS $+ 300$ bp to the transcription termination site $+3000$ bp). $P$ values were calculated using the Wilcoxon test from the library (rstatix) (version 0.7.2) from the R language. To visualize ATAC-seq and CUT&Tag datasets, the plotProfile and plotHeatmap functions of deepTools were used. All analyses were conducted in R (v4.2.2) and Bioconductor (v3.16). Visualization outputs were exported as vector PDFs for publication-quality figures. Key packages: ChIPseeker, GenomicRanges, DiffBind, ggplot2, Gviz, TxDb.Mmusculus.UCSC.mm10.knownGene, org.Mm.eg.db.

## Statistics

The sample size $n$ refers to the number of independent experiments or the number of mice in each experiment. Representative results from at least three independent experiments were shown. Data are represented as means ± SEM. Statistical comparisons were made by using Student's $t$ test with GraphPad Prism 8. Statistical comparisons for single-cell analysis, Gene Ontology analysis, and pausing index were performed using Wilcoxon tests.

## Data availability

The sequencing data produced in this study have been deposited in Gene Expression Omnibus (GEO) repository under the accession number GSE300207. The code used to perform analyses in this paper is available on GitHub at: https://github.com/ketone08/Larp7_KO_mouse.

The source data of this paper are collected in the following database record: biostudies:S-SCDT-10_1038-S44318-026-00752-w.

## Peer review information

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

## Acknowledgements

We thank Mr. Shucheng Zhao for his dedication to this research project. This work was supported by the National Natural Science Foundation of China (32271348), the Science and Technology Department of Sichuan Province (2024NSFSC0731), and the West Women and Children Medical Research Development Center (XBFY-YJXM2025004).

## Author contributions

**Yin Fang**: Data curation; Formal analysis; Investigation; Methodology; Writing—review and editing. **Tong Qiu**: Data curation; Formal analysis; Validation; Investigation; Visualization; Methodology. **Ping Wang**: Data curation; Formal analysis; Investigation; Methodology. **Shujun Bai**: Data curation; Formal analysis; Investigation; Methodology. **Min Wang**: Investigation. **Chao Yang**: Data curation. **Yan Wang**: Investigation. **Peixuan Zhang**: Investigation. **He Wang**: Resources. **Shanling Liu**: Resources; Supervision. **Xue Xiao**: Resources; Supervision. **Qintong Li**: Conceptualization; Resources; Data curation; Formal analysis; Supervision; Funding acquisition; Investigation; Visualization; Methodology; Writing—original draft; Writing—review and editing.

Source data underlying figure panels in this paper may have individual authorship assigned. Where available, figure panel/source data authorship is listed in the following database record: biostudies:S-SCDT-10_1038-S44318-026-00752-w.

## Disclosure and competing interests statement

The authors declare no competing interests.

