## [Peer Review File · The EMBO Journal]

Enhanced P-TEFb activity compromises dentate gyrus neurogenesis in mice

Yin Fang, Tong Qiu, Ping Wang, Shujun Bai, Min Wang, Chao Yang, Yan Wang, Peixuan Zhang, He Wang, Shanling Liu, Xue Xiao, and Qintong Li

Corresponding author(s): Qintong Li (liqintong@scu.edu.cn) , Xue Xiao (xiaoxuela@scu.edu.cn), Shanling Liu (liuxl@scu.edu.cn)

Review Timeline:

Submission Date:	1st Aug 25
Editorial Decision:	30th Sep 25
Revision Received:	29th Dec 25
Editorial Decision:	13th Feb 26
Revision Received:	15th Feb 26
Accepted:	25th Feb 26

Editor: Ioannis Papaioannou

Transaction Report:

Dear Prof. Li,

Thank you again for the submission of your manuscript EMBOJ-2025-122043 for consideration by The EMBO Journal, and for your patience during peer review. Your manuscript has now been seen by three experts in the field, and we have received the full set of their reports, which are included below.

I am very pleased to say that the referees recognize that this manuscript addresses an important question, indicate interest in the findings, find the work of good quality for the most part, and point out that the conclusions are well-supported by the presented data. However, they also identify certain parts of the manuscript as rather preliminary and make several reasonable and constructive suggestions for further strengthening of the manuscript and increase of its impact on the field.

In light of the largely positive referees' comments and recommendations, I would like to invite you to submit a revised version of your manuscript taking the referees' suggestions on board, along with a detailed point-by-point response addressing all referees' comments. Please note that it is The EMBO Journal policy to allow only a single round of major revision, and acceptance of your manuscript will therefore depend on the completeness of your responses in this revised version.

Please let me know if you have any questions or comments that you would like to discuss with me; it would be useful to discuss your revision plan already during the initial phase of your revisions. You are very welcome to share with me a draft point-by-point response letter/revision plan explaining if there are any points you do not agree with or cannot address, or alternatively we could arrange a video call, if you prefer.

We generally allow three months as standard revision time (December 29, 2025). As a matter of policy, competing manuscripts published during this period will not negatively impact our assessment of the conceptual advance presented by your study. However, we request that you contact us as soon as possible upon publication of any related work, to discuss how to proceed. Should you foresee a problem in meeting this three-month deadline, please let us know in advance and we will be able to grant an extension.

Thank you for the opportunity to consider your work for publication in The EMBO Journal. I look forward to your revision.

Best regards,

Ioannis

Instructions for preparing your revised manuscript

1. When you are ready to submit the revision, please upload:

- A Word file of the manuscript text (including legends of main Figures, EV Figures and Tables). Please make sure that changes are highlighted (or "tracked") to be clearly visible.

- Individual production-quality figure files (one file per figure). When assembling your figures, please refer to our figure preparation guidelines in order to ensure proper formatting and readability in print as well as on screen:

If the data shown in a figure are obtained from n {less than or equal to} 2, please use scatter plots showing the individual data points.

- i. the name of the statistical test used to generate error bars and P values
- ii. the number (n) of independent experiments (please specify technical or biological replicates) underlying each data point (discussion of statistical methodology can be reported in the Materials and Methods section, but figure legends should contain a basic description of n , P, and the test applied)
- iii. the nature of the bars and error bars (s.d., s.e.m.).

- A point-by-point response to the referees' comments, with a detailed description of the changes made (as a word file). All referees' concerns must be fully addressed and their suggestions taken on board. When preparing your letter of response to the referees' comments, please bear in mind that this will form part of the Review Process File and will therefore be available online to the community. Please note that you have the possibility to opt out of the transparent process at any stage prior to publication by letting the editorial office know (contact@embojournal.org); if you do opt out, the Review Process File link will point to the following statement: "No Review Process File is available with this article, as the authors have chosen not to make the review process public in this case.". For more details on our Transparent Editorial Process, please visit our website: <https://www.embopress.org/page/journal/14602075/authorguide#transparentprocess>

- Expanded View (EV) files (replacing Supplementary Information) that are collapsible/expandable online. A maximum of 5 EV Figures can be typeset. EV Figures should be cited as "Figure EV1, Figure EV2" etc. in the text, and their respective legends should be included in the manuscript file after the legends of regular figures. See detailed instructions regarding Expanded View files here: <https://www.embopress.org/page/journal/14602075/authorguide#expandedview>

- For the figures that you do NOT wish to display as Expanded View figures, they should be bundled together with their legends in a single PDF file called "Appendix", which should start with a short Table of Contents (including page numbers). Appendix figures should be referred to in the main text as: "Appendix Figure S1, Appendix Figure S2" etc. Please see detailed instructions here: <https://www.embopress.org/page/journal/14602075/authorguide#expandedview>

- A complete author checklist, which you can download from our author guidelines (<https://www.embopress.org/page/journal/14602075/authorguide>). Please note that the checklist will also be part of the Review Process File.

2. Please note that no statistics should be calculated and shown in Figures if $n=2$. Please also note that each p value should be reported as an exact value.

3. Before submitting your revision, primary datasets (and computer code, where appropriate) produced in this study need to be deposited in appropriate public databases (see <https://www.embopress.org/page/journal/14602075/authorguide#dataavailability>). In particular, all single-cell, RNA, and ATAC sequencing data generated in your study must be deposited in an appropriate repository. The accession numbers, database, and the specific URLs (links) should be listed in a formal "Data availability" section (placed after Methods), following the example below:

"The RNA-seq datasets produced in this study are available in the following database:
Gene Expression Omnibus GSE46843 (<https://www.ncbi.nlm.nih.gov/geo/query/acc.cgi?acc=GSE46843>)"

*** All links should resolve to a page where the data can be accessed. ***

*** Please remember to provide in the Data availability section of your revised manuscript reviewer passwords if the datasets are not yet public. ***

*** The Data Availability Section is restricted to new primary data that are part of this study. In case you have no data that require deposition in a public database, please state so instead of referring to the database: "Our study includes no data deposited in public repositories." under the heading "Data availability". ***

4. The materials and methods need to be described in the manuscript using our structured methods format, which is now required for all research articles. According to this format, the Methods section includes a single "Reagents and Tools Table" - listing key reagents, experimental models, software and relevant equipment including their sources and relevant identifiers - followed by a "Methods and Protocols" section describing the methods. Please download and fill our Reagents and Tools Table template (.docx), which you can find in our author guide:

<https://www.embopress.org/page/journal/14602075/authorguide#structuredmethods>. When submitting your revised manuscript, please do not include the Reagents and Tools Table in the Methods section of the manuscript but instead upload it as a separate file choosing the file type "Reagent Table".

5. Please check that the title and the abstract of the manuscript are brief, yet explicit, even to non-specialists. The length of the title should not exceed 100 characters, and the abstract should be a single paragraph not exceeding 175 words.

6. Please also note our reference format: <https://www.embopress.org/page/journal/14602075/authorguide#referencesformat>.

8. Please remember: digital image enhancement is acceptable practice, as long as it accurately represents the original data and

conforms to community standards. If a figure has been subjected to significant electronic manipulation, this must be noted in the figure legend or in the "Materials and Methods" section. The editors reserve the right to request original versions of figures and the original images that were used to assemble the figure.

9. Our journal encourages inclusion of data citations in the reference list to directly cite datasets that were obtained from public databases. Data citations in the article text are distinct from normal bibliographical citations and should directly link to the database records from which the data can be accessed. In the main text, data citations are formatted as follows: "Data ref: Smith et al, 2001" or "Data ref: NCBI Sequence Read Archive PRJNA342805, 2017". In the Reference list, data citations must be labeled with "[DATASET]". A data reference must provide the database name, accession number/identifiers, and a resolvable link to the landing page from which the data can be accessed at the end of the reference. Further instructions are available at: <https://www.embopress.org/page/journal/14602075/authorguide#referencesformat>.

10. We request authors to consider both actual and perceived competing interests. Please review our policy (<https://www.embopress.org/page/journal/14602075/authorguide#conflictsofinterest>) and update your competing interests statement if necessary. Please name this section 'Disclosure and competing interests statement' and place it after the Acknowledgements section.

11. Please note that all corresponding authors are required to provide an ORCID ID upon submission of a revised manuscript (<https://orcid.org/>). Please find instructions on how to link your ORCID ID to your account in our manuscript tracking system in our Author guidelines (<https://www.embopress.org/page/journal/14602075/authorguide#authorshipguidelines>).

12. We use CRediT to specify the contributions of each author in the journal submission system. CRediT replaces the author contribution section, which should be removed from the manuscript. Please use the free text box to provide more detailed descriptions. See also guide to authors: <https://www.embopress.org/page/journal/14602075/authorguide#authorshipguidelines>.

14. We would also welcome the submission of cover suggestions or motifs to be used by our Graphics Illustrator in designing a cover.

Referee #1:

Fang and colleagues provide compelling data on the role of LARP7 in mediating growth through its role in P-TEFb. They specifically probe the consequences of its deficiency in the dentate gyrus and show that while it does lead to increased P-TEFb and consequently Pol II activity as expected, it leads to hypoplasia rather than hyperplasia because it keeps progenitor cells in a perpetual self renewal state. The work is of high quality and has important clinical implications not only in terms of explaining the pathogenicity of Alazami syndrome but also potentially age-related degeneration of the dentate gyrus. I have several comments to improve the quality of the manuscript:

1- The proposal by the authors is as intriguing as it is radical. As the authors know, primary microcephaly is widely believed to be caused by the exact opposite mechanism to what they're proposing. Specifically, it's been clearly shown that neuroprogenitors in primary microcephaly models exit the self renewal state and enter prematurely into differentiation resulting in an overall reduced bulk of cells that make the future cortex. Is it possible that the timing in development accounts for this difference? In other words, Larp7 deficient mice are born with normal size but develop growth deceleration postnatally so the mechanism proposed by the authors only operate postnatally whereas prenatally timed premature exit of the self-renewal state causes primary microcephaly?

2- The authors are to be commended for the careful characterization of the growth deceleration pattern, which is consistent with the clinical updates on Alazami syndrome. It is now widely established that the microcephalic dwarfism in this syndrome is postnatal and not prenatal onset as erroneously thought in the original report.

3- The paper falls short of explaining the whole body growth deceleration. First, we are not shown if LARP7 is absent in the non-brain tissue as speculated by the authors. Second, even if it is, how does their proposal apply to the rest of the body when it is specific to neuronal progenitors?

4- Interestingly, there is an autosomal recessive condition linked to CDK9 and the authors may want to refer to it as an example of how unique LARP7's role is compared to CDK9 (see PMID: 26633546 and PMID: 30237576).

Minor:

- 1- What is exactly meant by "ordination" and "subtissular"?
- 2- The sentence "to establish or disprove the causal link between LARP7 deficiency and Alazami syndrome" is a bit hyperbolic. Animal models do not establish or disprove a human Mendelian disease if the human genetics is not consistent. In the case of Alazami syndrome, the human genetics evidence is definitive at this point and cannot be "refuted" by an animal model.
- 3- "variants" instead of "mutations" to conform to the modern nomenclature.
- 4- "started to shrink" instead of "started to shrank".
- 5- "phenocopies dentate gyrus" instead of "phenocopy dentate gyrus".

Referee #2:

LARP7 is involved in the inhibition of the P-TEFb activity through the formation of the 7SKsnRNP complex and considered to be a negative factor against transcription and cell proliferation. However, genetic inactivation of LARP7 is associated with growth restriction and cognitive impairment such as the Alzami syndrome. To resolve the seeming paradox, the authors generated and analysed LARP7 conditional knockout mice, and report that the size of the brain, particularly the hippocampal dentate gyrus, is reduced during the neonatal period in these mice. Various behavioural tests indicate that functions associated with the dentate gyrus, such as spatial navigation, memory consolidation, exploration, and tolerating anxiety, are compromised in these mice. Increased P-TEFb-dependent transcription and enhanced self-renewal of transit-amplifying neuronal progenitor cells that limit neurogenesis in the dentate gyri were observed in LARP7 knockout cells by various functional and multi-omics analyses. A similar gyrus hypoplasia was also observed in mice with a genetic deletion of HEXIM1/2, another 7SKsnRNP component. The authors conclude that the lack of LARP7 results in dysregulation of sub-tissular stem cell dynamics via enhancement of P-TEFb-dependent transcription and propose re-evaluation of P-TEFb inhibitors' therapeutic potential for the treatment of medical conditions associated with dentate gyrus size and function. The observations are very interesting and the data support the authors' conclusion very well. Data with HEXIM1/2 floxed mice are somewhat preliminary, however. Some additional experiments/data would ensure the robustness of the study.

Specific points:

1. The reduction of the body size of LARP7 f/f mice seems less severe than that in Alzami syndrome. Any further comments about this difference would be helpful. Any other factors/pathways dysregulated by the deletion of LARP7 need to be considered?
2. Although circumstantial data support the enhancement of P-TEFb activity in LARP7 knockout cells, data directly showing that P-TEFb activity is elevated would also be very helpful. Particularly, possibilities of cellular compensational mechanisms to limit the P-TEFb activity in the absence of LARP7/7SKsnRNA could be taken into consideration.
3. Similarly, it is helpful to show that the protein levels of P-TEFb (CDK9, CycT1, CycT2 etc) and 7SKsnRNP components (HEXIM1, for example) did not change with or without LARP7 knockout.
4. As I mentioned above, the data with HEXIM1/2 f/f mice seem somewhat preliminary. In addition, at least two groups have reported HEXIM1 knockout mice (<https://doi.org/10.1161/CIRCRESAHA.107.157859>, <https://doi.org/10.1016/j.mod.2004.04.012>), which could be referred to.
5. In LARP7 f/f mice, the mRNA expression of HEXIM1 is modestly increased, which authors did not mention. Is this increase statistically significant? Since HEXIM1 itself is one of P-TEFb's target genes transcribed immediately after P-TEFb activation, this could be a sign of P-TEFb activation although one could expect the increase be greater. Additional comments would be helpful.

Referee #3:

In this study, the authors investigate the role of LARP7 and HEXIM1 in regulating neuronal development through modulation of P-TEFb activity. Enhanced P-TEFb activity is generally thought to promote cell proliferation by increasing the transcriptional output of RNA polymerase II, and under normal conditions most P-TEFb is inhibited by the 7SK snRNP complex containing LARP7 and HEXIM1. Paradoxically, biallelic inactivation of LARP7 in humans' results in Alazami syndrome, a neurodevelopmental disorder that is characterized by growth restriction and cognitive impairment. The authors report that conditional ablation of either *Larp7* or *Hexim1* in the murine brain reduces the size of the hippocampal dentate gyrus during the neonatal period. By combining transcriptomic, epigenomic, and functional analyses, they further propose that heightened P-TEFb activity enhances self-renewal transcriptional programs in transit-amplifying neuronal progenitors, thereby limiting neurogenesis in the developing dentate gyrus. These findings suggest a potential mechanistic explanation linking dysregulated progenitor dynamics with reduced brain size and highlight possible translational applications of clinical-grade P-TEFb inhibitors. The topic is interesting and the authors address an important question at the interface of transcriptional regulation and neurodevelopment. Many of the experiments are carefully conducted and the conclusions are of potential significance to the field. However, I have some doubts on the presentation of the manuscript and on the conclusions reached by the authors based on these experiments. Additional data should be provided to sustain the discussion. Overall, this study reveals the impact of P-TEFb activity on dentate gyrus neurogenesis in Alazami syndrome mouse models, but the quality of data in this paper is not high and improvements are needed.

Major concerns:

1. The introduction could be enhanced by including recent advances on P-TEFb regulation in neural stem cell fate determination. Furthermore, updating the clinical spectrum of Alzami syndrome with recent findings, particularly regarding its genotype-phenotype heterogeneity, would improve the timeliness and clinical relevance of the discussion.
2. In the manuscript, there are inherent differences between the mouse model and patient samples. It is recommended that the authors include an analysis of Alzami syndrome patient samples carrying LARP7 mutations, and compare the findings with those obtained from the mouse model to strengthen the clinical relevance.
3. Although the mouse models recapitulate key features of Alzami syndrome, the authors should discuss whether similar transcriptomic changes (e.g., upregulated self-renewal genes) have been observed in human patient-derived iPSCs or organoids, if such data are available.
4. For Figure 1G, how many brain sections were analyzed per mouse, and how was consistency in sectioning positions ensured? At present, the staining and quantification shown in Figures 1F and 1G are not sufficient to support the conclusion that the dentate gyrus is reduced in size. Additional types of experimental evidence are required.
5. The study mentions that Alzami syndrome patients exhibit cognitive deficits but does not clarify the correspondence between dentate gyrus hypoplasia and specific cognitive functions (e.g., spatial memory, social ability). It is suggested to supplement direct evidence for the association between dentate gyrus neuronal number/functional abnormalities and behavioral deficits (e.g., through neuron-specific rescue experiments).
6. Behavioral quantification: mismatch between Methods and probe-trial claims. In the Behavioral testing / Morris water maze subsection, the Methods state that mice were trained for 5 days and that "time in the target quadrant" and "number of crossings" during the 5-day training period were used as learning indices. By field standards, training-phase learning is quantified by escape latency/path length across days, whereas time in target quadrant and platform-location crossings are metrics of the probe trial (platform removed). In contrast, the figure legends (e.g., Fig. 1H, Fig. 2I) describe probe-trial measures. This creates a Methods-to-Results inconsistency. Please clarify and, if needed, re-analyze to conform to standard MWM practice: (i) report training curves (escape latency and/or path length) over days with the appropriate repeated-measures analysis; (ii) for the probe trial, report time in target quadrant and platform-location crossings (with chance-level comparison), and avoid training metrics here; (iii) provide control measures (swim speed, visible-platform performance) to exclude motor/visual confounds; (iv) make Methods and figure legends internally consistent (clearly separate training vs probe metrics and specify which are plotted in each panel).
7. The single-cell transcriptomic analyses presented in Figure 3 are not sufficient to directly support the conclusion that "loss of Larp7 activates P-TEFb and promotes RNA Pol II transcription." More direct experimental evidence is required.
8. The study reports that LARP7 deficiency increases the self-renewal of nIPCs while reducing their differentiation. However, it remains unclear whether this is due to depletion of the progenitor pool or delayed neuronal differentiation. It is recommended that the authors perform longitudinal analyses at multiple developmental stages, including single-cell RNA sequencing with pseudotime analysis, to clarify the underlying cause.
9. To better assess the effects of Larp7 deficiency (Larp7^{f/f}; nestin-Cre) on neurogenesis, the inclusion of in vitro assays and pharmacological inhibition experiments is recommended.
10. Phenotypic analysis of the Hexim1 knockout model in Figure 7 is relatively brief. It is recommended to supplement behavioral data (e.g., spatial memory, anxiety levels) and quantitative results of nIPC proliferation/differentiation in Hexim1 knockout mice to form a complete comparison with the LARP7 knockout model.
11. The manuscript lacks sufficient mechanistic exploration, including predictions and validation of downstream target genes, as well as rescue experiments.

Minor comments:

1. The author should provide the statistical data of Figure 1B, Figure 6A, Figure 6B, Figure 6F. In manuscript, in figure 2, the behavioral experiments related to mice, such as learning and memory, anxiety, at least two methods of each of behavioral evidence are required to be provided.
2. Figure 1B Western blot lacks quantitative analysis of protein expression.
3. The reference formatting contains errors: there should be a space between the preceding text and the parentheses, and multiple references within the same parentheses should be separated by semicolons.
4. In Figure S2, the single-cell cluster identities are not clearly defined. Do these clusters include cell types beyond astrocytes, RGLs, nIPCs, and neuroblasts?
5. Using multiple markers to independently identify NSCs, nIPCs, and neurons would strengthen the reliability of the results.
6. The ATAC-seq peak plot in Figure 6C lacks a vertical axis scale, which should be added to facilitate interpretation.
7. Figure 7G lacks corresponding statistical analyses.
8. The conclusion "We conclude that the disruption of the 7SK snRNP complex, thus the activation of P-TEFb, causes hypoplastic dentate gyrus" is not fully supported. In the Hexim1^{f/f};Emx1-Cre mouse model, only dentate gyrus size was assessed, other functional or structural readouts were not examined, and the potential effects of Hexim2 deletion were not investigated.
9. The statement in the abstract "Multi-omics and functional analyses reveal ..." is misleading. Although both transcriptomic and epigenomic analyses were performed, the datasets are presented separately without integration, so describing the study as a "multi-omics analysis" is not appropriate.
10. The statement in the abstract "Here, we report that conditional ablation of either Larp7 or Hexim1 in the murine brain reduces the size and function of the hippocampal dentate gyrus during the neonatal period" is imprecise. The manuscript presents evidence only for Hexim1 deletion affecting DG size, without evaluating neuronal function.

Response to Reviewers:

We sincerely thank all three Reviewers for their extensive and constructive comments. Responding to these has improved the manuscript. Our point-by-point responses to their comments are appended below in blue.

Referee #1:

Fang and colleagues provide compelling data on the role of LARP7 in mediating growth through its role in P-TEFb. They specifically probe the consequences of its deficiency in the dentate gyrus and show that while it does lead to increased P-TEFb and consequently Pol II activity as expected, it leads to hypoplasia rather than hyperplasia because it keeps progenitor cells in a perpetual self renewal state. The work is of high quality and has important clinical implications not only in terms of explaining the pathogenicity of Alzami syndrome but also potentially age-related degeneration of the dentate gyrus. I have several comments to improve the quality of the manuscript:

Thank you for your overall positive view of the manuscript. We are grateful for your suggestions, which we believe have improved the revised manuscript.

1- The proposal by the authors is as intriguing as it is radical. As the authors know, primary microcephaly is widely believed to be caused by the exact opposite mechanism to what they're proposing. Specifically, it's been clearly shown that neuroprogenitors in primary microcephaly models exit the self renewal state and enter prematurely into differentiation resulting in an overall reduced bulk of cells that make the future cortex. Is it possible that the timing in development accounts for this difference? In other words, Larp7 deficient mice are born with normal size but develop growth deceleration postnatally so the mechanism proposed by the authors only operate postnatally whereas prenatally timed premature exit of the self-renewal state causes primary microcephaly?

We thank the Reviewer for this insightful comment. As you mentioned, the size of the developing cortex depends on the balance between self-renewal and differentiation of neural progenitor cells. We suspect that the timing of the phenotypic manifestation depends, at least partly, on the biological properties of primary microcephaly genes. On the one hand, many primary microcephaly mutations occur

in genes encoding core machineries of DNA replication, such as those encoding centrosome proteins, DNA replication and repair factors. Genetic ablation of these genes, such as *ATR*, in mouse models usually elicits very strong DNA damage responses, which in turn promote neural progenitor cell death and differentiation. In these cases, a significant decline in neural progenitor cell number can be observed during embryogenesis. On the other hand, among known microcephaly genes, *LARP7* is an outlier because it functions as a regulator of RNA polymerase II transcription rather than the core machineries of DNA replication. Its genetic ablation in murine neural progenitor cells does not elicit strong DNA damage responses. Thus, the phenotypic manifestation of *Larp7* ablation may be delayed. It will be of interest in future studies to determine whether molecular changes in *Larp7*-null neural progenitor cells already occur during embryogenesis, or only happen postnatally.

2- The authors are to be commended for the careful characterization of the growth deceleration pattern, which is consistent with the clinical updates on Alazami syndrome. It is now widely established that the microcephalic dwarfism in this syndrome is postnatal and not prenatal onset as erroneously thought in the original report.

Thank you for pointing this out. In the revised manuscript, we have added a summary of core features of Alazami syndrome patients based on recent literature, and compared them with phenotypes in our mouse models (Appendix Figure S8, new data in the revised manuscript). The new figure is reproduced here for your reference:

Phenotypic Category	Human LARP7 Mutation Patients	Variant 1	Variant 2	Larp7^{off} ;nestin-Cre Mice	Larp7^{off} ;Emx1-Cre Mice Hexim1 ;Emx1-Cre Mice
Inheritance	Autosomal recessive ¹	/	/	Autosomal recessive	Autosomal recessive
Height/Body length	Short stature	c.1173T>A (p.Y391*) ²	c.1173T>A (p.Y391*)	Reduced body length (fully penetrant)	No change
		c.1653_1654del (p.G553fs) ²	c.1653_1654del (p.G553fs)		
		c.646+3_646+6del ²	c.646+3_646+6del		
		c.1173T>A (p.Y391*) ²	c.1653_1654del (p.G553fs)		
		c.213_214dup (p.S72fs) ³	c.651_655del (p.(K219fs))		
		c.1070_1073del (p.R357fs) ⁴	c.1070_1073del (p.R357fs)		
		c.1091_1094del (p.K364fs) ⁵	c.1091_1094del (p.K364fs)		
		c.1669-1_1671del ⁶	c.834dup (p.R279fs)		

		c.503_504dup (p.A169fs) ⁷	c.503_504dup (p.A169fs)		
		c.892_895dup (p.S299fs) ⁸	c.1087_1091del (p.H363fs)		
		c.1024_1030dup (p.T344fs) ⁴	c.1024_1030dup (p.T344fs)		
Weight	Low body weight	c.1173T>A (p.Y391*) ²	c.1173T>A (p.Y391*)	Reduced body weight (both sexes)	No change
		c.1653_1654del (p.G553fs) ²	c.1653_1654del (p.G553fs)		
		c.646+3_646+6del ²	c.646+3_646+6del		
		c.1173T>A (p.Y391*) ²	c.1653_1654del (p.G553fs)		
		c.213_214dup (p.S72fs) ³	c.651_655del (p.K219fs)		
		c.1091_1094del (p.K364fs) ⁵	c.1091_1094del (p.K364fs)		
		c.1669-1_1671del ⁶	c.834dup (p.R279fs)		
		c.503_504dup (p.A169fs) ⁷	c.503_504dup (p.A169fs)		
		c.892_895dup (p.S299fs) ⁸	c.1087_1091del (p.H363fs)		
		c.1024_1030dup (p.T344fs) ⁴	c.1024_1030dup (p.T344fs)		
Head	Microcephaly	c.1173T>A (p.Y391*) ²	c.1173T>A (p.Y391*)	Disproportionately reduced dentate gyrus size during neonatal period	No overall brain size reduction, but reduced dentate gyrus
		c.1653_1654del (p.G553fs) ²	c.1653_1654del (p.G553fs)		
		c.646+3_646+6del ²	c.646+3_646+6del		
		c.1173T>A (p.Y391*) ²	c.1653_1654del (p.G553fs)		
		c.213_214dup (p.S72fs) ³	c.651_655del (p.K219fs)		
		c.1070_1073del (p.R357fs) ⁴	c.1070_1073del (p.R357fs)		
		c.1091_1094del (p.K364fs) ⁵	c.1091_1094del (p.K364fs)		
		c.1669-1_1671del ⁶	c.834dup (p.R279fs)		
		c.503_504dup (p.A169fs) ⁷	c.503_504dup (p.A169fs)		
		c.892_895dup (p.S299fs) ⁸	c.1087_1091del (p.H363fs)		
c.1024_1030dup (p.T344fs) ⁴	c.1024_1030dup (p.T344fs)				
Central Nervous System	Intellectual disability	c.832A>T (p.K278*) ⁹	c.832A>T (p.K278*)	Impaired spatial learning and memory	Impaired spatial learning and memory
		c.1173T>A (p.Y391*) ²	c.1173T>A (p.Y391*)		
		c.646+5G>C ¹⁰	c.834dup (p.R279fs)		
		c.349del (p.E117fs) ¹¹	c.620_646+25del (p.P208_E216del)		
		c.1653_1654del (p.G553fs) ²	c.1653_1654del (p.G553fs)		
		c.646+3_646+6del ²	c.646+3_646+6del		
		c.1173T>A (p.Y391*) ²	c.1653_1654del (p.G553fs)		
		c.827dup (p.K277fs) ¹⁰	c.827dup (p.K277fs)		
		c.1024_1030dup (p.T344fs) ^{4,10}	c.1024_1030dup (p.T344fs)		
		c.1091_1094del (p.K364fs) ¹⁰	c.1091_1094del (p.K364fs)		
		c.756_757del (p.R253fs) ¹⁰	c.756_757del (p.R253fs)		
		c.503_504dup (p.A169fs) ^{7,10}	c.503_504dup (p.A169fs)		
		c.213_214dup (p.S72fs) ³	c.651_655del (p.K219fs)		
		c.1070_1073del (p.R357fs) ⁴	c.1070_1073del (p.R357fs)		

		c.1091_1094del (p.K364fs) ⁵	c.1091_1094del (p.K364fs)		
		c.1669-1_1671del ⁶	c.834dup (p.R279fs)		
		c.892_895dup (p.S299fs) ⁸	c.1087_1091del (p.H363fs)		
Behavioral Psychiatric Manifestations	Severe anxiety	c.1173T>A (p.Y391*) ²	c.1653_1654del (p.G553fs)	Significantly increased anxiety-related behaviors	Significantly increased anxiety-related behaviors
		c.213_214dup (p.S72fs) ^{3,10}	c.651_655del (p.K219fs)		
		c.1091_1094del (p.K364fs) ¹⁰	c.1091_1094del (p.K364fs)		
Other Features	/	/	/	Normal lifespan	Normal lifespan

3- The paper falls short of explaining the whole body growth deceleration. First, we are not shown if LARP7 is absent in the non-brain tissue as speculated by the authors. Second, even if it is, how does their proposal apply to the rest of the body when it is specific to neuronal progenitors?

Thank you for raising this point. In the present study, we focus on dissecting the defects in neural progenitor cells for two reasons. Firstly, one of the core phenotypes among Alazami syndrome patients is impaired cognitive functions. It was previously unknown whether this phenotype is linked to any defects of neural progenitor cells. Secondly, ablation of *Larp7* by nestin-cre or Emx1-cre causes the same defect in the hippocampal dentate gyrus. However, *Larp7* ablation by nestin-cre additionally causes whole-body growth deceleration, whereas its ablation by Emx1-cre does not. This difference provided an opportunity to dissect defective neurogenesis in dentate gyrus without the complications of the whole-body growth deceleration.

We fully agree with the Reviewer that our proposed mechanism for neural progenitor cells may not be applicable in other organs. It is very interesting that *Larp7* ablation by nestin-cre results in whole-body growth defects, reminiscent of core features of Alazami syndrome patients. However, it is difficult to pinpoint how this defect occurs, due to the expression of nestin-cre in several organs. It is an ongoing effort in the lab to use other tissue-specific cre mouse strains to knock out *Larp7*, in order to determine the exact cause of whole-body growth defects.

Following your suggestion, we have added new information on the organs, in which nestin-cre activity can be detected (Appendix Figure S3, new data in the revised manuscript). The new data is also reproduced here for your reference:

We have added additional text in the revised manuscript as follows:

Line 237-240:

“The recombinant activity of nestin-Cre is active in multiple cell types spanning the whole brain, but also in other organs such as kidney (Wang, Liu et al., 2024), stomach and lung (Appendix Fig. S3).”

4- Interestingly, there is an autosomal recessive condition linked to CDK9 and the authors may want to refer to it as an example of how unique LARP7's role is compared to CDK9 (see PMID: 26633546 and PMID: 30237576).

Thank you for the information. We have added additional text in Discussion in the revised manuscript as follows:

Line 475-476:

“On the other hand, insufficient P-TEFb activity may also cause developmental defects in humans (Maddirevula, Alzahrani et al., 2019, Shaheen, Patel et al., 2016).”

Minor:

1- What is exactly meant by "ordination" and "subtissular"?

Thank you for pointing this out. “ordination” is a typo, and has been corrected to “coordination” in the revised manuscript (Line 42). “subtissular” refers to the fact that the dentate gyrus occupies a very small portion of the cortical region.

2- The sentence "to establish or disprove the causal link between LARP7 deficiency and Alzami syndrome" is a bit hyperbolic. Animal models do not establish or disprove a human Mendelian disease if the human genetics is not consistent. In the

case of Alazami syndrome, the human genetics evidence is definitive at this point and cannot be "refuted" by an animal model.

Thank you for pointing this out. In the revised manuscript (Line 102), this statement is deleted, and changed to "to investigate the cellular and molecular mechanisms underlying Alazami syndrome".

3- "variants" instead of "mutations" to conform to the modern nomenclature.

Following your advice, we have used "variants" in the revised manuscript.

4- "started to shrink" instead of "started to shrank".

The text has been corrected in the revised manuscript (Line 204). Thank you.

5- "phenocopies dentate gyrus" instead of "phenocopy dentate gyrus".

The text has been corrected in the revised manuscript (Line 422). Thank you.

Referee #2:

LARP7 is involved in the inhibition of the P-TEFb activity through the formation of the 7SKsnRNP complex and considered to be a negative factor against transcription and cell proliferation. However, genetic inactivation of LARP7 is associated with growth restriction and cognitive impairment such as the Alazami syndrome. To resolve the seeming paradox, the authors generated and analysed LARP7 conditional knockout mice, and report that the size of the brain, particularly the hippocampal dentate gyrus, is reduced during the neonatal period in these mice. Various behavioural tests indicate that functions associated with the dentate gyrus, such as spatial navigation, memory consolidation, exploration, and tolerating anxiety, are compromised in these mice. Increased P-TEFb-dependent transcription and enhanced self-renewal of transit-amplifying neuronal progenitor cells that limit neurogenesis in the dentate gyri were observed in LARP7 knockout cells by various functional and multi-omics analyses. A similar gyrus hypoplasia was also observed in mice with a genetic deletion of HEXIM1/2, another 7SK snRNP component. The authors conclude that the lack of LARP7 results in dysregulation of sub-tissular stem cell dynamics via enhancement of P-TEFb-dependent transcription and propose re-evaluation of P-TEFb inhibitors' therapeutic potential for the treatment of medical conditions

associated with dentate gyrus size and function. The observations are very interesting and the data support the authors' conclusion very well. Data with HEXIM1/2 floxed mice are somewhat preliminary, however. Some additional experiments/data would ensure the robustness of the study.

Thank you for highlighting the novelty, impact, and rigor of the work. We are gratified that we were able to convey some of the excitement of the results.

Specific points:

1. The reduction of the body size of LARP7 f/f mice seems less severe than that in Alazami syndrome. Any further comments about this difference would be helpful. Any other factors/pathways dysregulated by the deletion of LARP7 need to be considered?

We agree with the Reviewer that the reduction of the body size in *Larp7* knockout mice is less severe than what has been observed in the Alazami syndrome patients. One likely explanation is that in patients, *LARP7* is inactivated in the zygote stage, that is, *LARP7* function is lost in all cells in patients. This would be equivalent to germline knockout of *Larp7* in mice. In contrast, nestin-cre functions in limited tissues (Appendix Figure S3, new data in the revised manuscript). It is an ongoing effort in the lab to use other tissue-specific cre mouse strains to knock out *Larp7* to determine the exact cause of whole-body growth defects. Unfortunately, germline knockout of *Larp7* is not feasible as mice die shortly after birth for unknown reasons. We mentioned this fact in the Results (Line 131), and added additional text in Discussion in the revised manuscript as follows:

Line 417-421:

“It is worth noting that the reduction of the body size of *Larp7^{ff};nestin-Cre* mice seems less severe than that in Alazami syndrome patients. One potential explanation is that nestin-Cre only functions in selected organs in mice. In contrast, human patients carry germline inactivating variants of *LARP7*, that is, the function of LARP7 protein is lost in all cells.”

The new data (Appendix Figure S3) is reproduced here for your reference:

2. Although circumstantial data support the enhancement of P-TEFb activity in LARP7 knockout cells, data directly showing that P-TEFb activity is elevated would also be very helpful. Particularly, possibilities of cellular compensational mechanisms to limit the P-TEFb activity in the absence of LARP7/7SKsnRNA could be taken into consideration.

Previous studies have established that without 7SK snRNP, P-TEFb kinase activity is increased. In the present study, we corroborate this conclusion in the context of primary neural stem and progenitor cells from several aspects. Firstly, the best characterized substrate of P-TEFb is the carboxy-terminal domain (CTD) of the largest subunit of RNA polymerase II. CTD contains multiple repeats of the heptapeptide sequence YSPTSPS, and P-TEFb phosphorylates Ser2 within this sequence. We have shown that Ser2 phosphorylation level is increased in *Larp7*-null cells (Figure 6B). Secondly, the outcome of increased P-TEFb activity is expected to increase RNA polymerase II transcription. We have shown that the mRNA content is increased in *Larp7*-null cells (Figures 3B and 5B). Although mRNA content can be influenced by multiple mechanisms, increased mRNA content per cell is in line with higher transcription activity. Thirdly, it is generally accepted that higher transcription activities are correlated with more accessible chromatin. We analyzed the chromatin accessibility by ATAC-seq, and showed that the chromatin is more accessible in *Larp7*-null cells than wild-type cells (Figure 6C). Fourthly, we used CUT&Tag assay to measure the density of total as well as elongating RNA polymerase II on annotated genes. The pausing index of RNA polymerase II was decreased (Figure 6D), and more peaks of elongating RNA polymerase II were detected in the gene body regions in *Larp7*-null cells than wild-type cells (Sox2 gene body was used as one of many examples in Figure 6E). Decreased pausing index and more elongating RNA polymerase II in the gene body regions are consistent with higher transcription activity.

The Reviewer raised an interesting point on potential cellular compensational

mechanisms to limit the P-TEFb activity in the absence of 7SK snRNP. As far as we know, this topic is understudied especially *in vivo*. At this stage, we can only speculate that one potential candidate might be one or several CTD phosphatases. As aforementioned, an ongoing effort in the lab is to use other tissue-specific cre mouse strains to knock out *Larp7* in different organs. These mice will provide tools to address compensational mechanisms *in vivo* in future studies.

3. Similarly, it is helpful to show that the protein levels of P-TEFb (CDK9, CycT1, CycT2 etc) and 7SKsnRNP components (HEXIM1, for example) did not change with or without LARP7 knockout.

Following your suggestion, we carried out additional experiments. We find that 7sk snRNA level is minimal in *Larp7*-null neural stem and progenitor cells, whereas Cdk9, and Hexim1 protein levels are unaltered (Figure 7O and 7P, new data in the revised manuscript). In contrast, 7sk snRNA, Cdk9 and *Larp7* levels are unaltered in *Hexim1*-null cells (Figure 7O and 7P, new data in the revised manuscript). The new data is reproduced here for your reference:

We added additional text in the revised manuscript as follows:

Line: 1188-1193

In Figure 7 legends:

“(O) Protein blot analyses of indicated proteins from wild-type and mutant neurospheres. Representative results were shown (n=3). Error bars, mean ± SEM. P-values, two-sided unpaired student’s t test.

(P) RNA blot analyses of indicated RNA from wild-type and mutant neurospheres. Representative results were shown (n=3). Error bars, mean ± SEM. P-values, two-sided unpaired student's t test.”

4. As I mentioned above, the data with HEXIM1/2 f/f mice seem somewhat preliminary. In addition, at least two groups have reported HEXIM1 knockout mice (<https://doi.org/10.1161/CIRCRESAHA.107.157859>, <https://doi.org/10.1016/j.mod.2004.04.012>), which could be referred to.

Following your suggestion, we carried out additional experiments with *Hexim1* knockout mice. In the revised manuscript (new data, Figure 7H-7T), we show that *Hexim1* knockout shares similar features with *Larp7* knockout: (1) both *Hexim1* and *Larp7* knockout mice have hypoplastic dentate gyrus, and exhibit similar deficits in behavioral tests (new data, Figure 7H-7M); (2) both *Hexim1* and *Larp7* knockout neurospheres exhibit enhanced self-renewal and compromised differentiation (new data, Figure 7N-7T). We added additional text in the revised manuscript as follows:

In Results, line 387-404:

“In addition, these mice exhibited cognitive deficits in behavioral tests (Fig. 7I-7M), similar to *Larp7^{ff};nestin-Cre* and *Larp7^{ff};Emx1-Cre* mice. Neural stem and progenitor cells were derived from microdissected P1 *Hexim1^{ff};Emx1-Cre* dentate gyrus. Under optimal self-renewal culture condition, both wild-type and *Hexim1* knockout neurospheres contain densely packed cells with a smooth perimeter (Fig. 7N). We confirmed that the expression levels of 7sk snRNA, Cdk9, and *Larp7* were comparable in wild-type and *Hexim1* knockout neural stem and progenitor cells. In contrast, in *Larp7* knockout cells, 7sk snRNA expression was markedly decreased, whereas the expression levels of Cdk9 and *Hexim1* were unaltered (Fig. 7O and 7P). Consistent with elevated P-TEFb activity, CUT&Tag assay revealed that the pausing index of RNA polymerase II was decreased (Fig. 7Q), and more active RNA polymerase II peaks were detected in the gene body regions in *Hexim1* knockout cells than wild-type cells (Fig. 7R). Under suboptimal culture condition, the morphology of *Hexim1* knockout neurospheres was minimally affected, whereas wild-type neurospheres started to exhibit multiple branch-like structures extending from the core (Fig. 7S). Compared to wild-type cells, differentiation of *Hexim1* knockout cells was compromised, and this defect could be alleviated by P-TEFb inhibitor (Fig. 7T). Thus, *Hexim1* knockout and *Larp7* knockout share many phenotypic similarities

in mice as well as in neural stem and progenitor cells. We conclude that the disruption of the 7SK snRNP complex, thus the activation of P-TEFb, causes hypoplastic dentate gyrus.”

The new data is also reproduced here for your reference:

Line: 1164-1207

In Figure 7 legends:

“(I) Morris water maze test. During the probe trial (platform removed), wild-type (WT; n=10) and knockout (*Hexim1^{fl/fl};Emx1-Cre*; n=8) mice were tested. The number of platform crossing, latency to find the location of removed platform, and the total path

length were documented by SMART video tracking system. Error bars, mean \pm SEM. P-values, two-sided unpaired student's t test.

(J) Three-chamber sociability test. Wild-type (WT; n=10) and knockout (*Hexim1^{ff}*;Emx1-Cre; n=8) mice were tested. Total sniffing time spent towards empty chamber (E) or stranger 1 (S1) was documented by SMART video tracking system. Error bars, mean \pm SEM. P-values, two-sided unpaired student's t test.

(K) Three-chamber social novelty test. Wild-type (WT, n=10) and knockout (*Hexim1^{ff}*;Emx1-Cre; n=8) mice were tested. Total sniffing time spent toward stranger 1 (S1) or stranger 2 (S2) were documented by SMART video tracking system. Error bars, mean \pm SEM. P-values, two-sided unpaired student's t test.

(L) Open field test. Wild-type (WT; n=10) and knockout (*Hexim1^{ff}*;Emx1-Cre; n=8) mice were tested. The time spent in the indicated areas, including Center, Non-periphery and Periphery, and total distance were documented by SMART video tracking system. Error bars, mean \pm SEM. P-values, two-sided unpaired student's t test.

(M) Elevated plus maze test. Wild-type (WT; n=10) and knockout (*Hexim1^{ff}*;Emx1-Cre; n=8) mice were tested. The time spent in the indicated areas, including Center, Open arms, and Closed arms, was documented by SMART video tracking system. Error bars, mean \pm SEM. P-values, two-sided unpaired student's t test.

(N) Phase contrast images of self-renewing neurospheres, derived from the dentate gyrus of P1 wild-type (*Hexim1^{ff}*) and knockout (*Hexim1^{ff}*;Emx1-Cre) mice, respectively. Representative results were shown (n=3). Error bars, mean \pm SEM. P-values, two-sided unpaired student's t test. Scale bars: 100 μ m.

(O) Protein blot analyses of indicated proteins from wild-type and mutant neurospheres. Representative results were shown (n=3). Error bars, mean \pm SEM. P-values, two-sided unpaired student's t test.

(P) RNA blot analyses of indicated RNA from wild-type and mutant neurospheres. Representative results were shown (n=3). Error bars, mean \pm SEM. P-values, two-sided unpaired student's t test.

(Q) The pausing index (PI) in wild-type and knockout (*Hexim1^{ff}*;Emx1-Cre) neural stem and progenitor cells. Statistical significance was assessed using the two-sided Wilcoxon rank-sum test.

(R) Browser shot depicting Pol II peaks within Sox2 gene in wild-type and knockout (*Hexim1^{ff}*;Emx1-Cre) neural stem and progenitor cells.

(S) The self-renewal capacity of wild-type and knockout (*Hexim1^{ff}*;Emx1-Cre) neurospheres under sub-optimal self-renewal culture condition. Representative results were shown (n=4). The percentage of neurospheres with a branched morphology, indicative of differentiation, was quantified. Error bars, mean \pm SEM. P-values, two-sided unpaired student's t test. Scale bars: 100 μ m.

(T) Differentiation of neurospheres for 6 days with or without P-TEFb inhibitor (flavopiridol). DMSO is the solvent of flavopiridol (Flavo). Representative images were shown (left panel) (n=3). GFAP immunofluorescence was used to quantify differentiated cells (right panel). Error bars, mean \pm SEM. P-values, two-sided unpaired student's t test. Scale bars: 100 μ m.”

5. In LARP7 *ff* mice, the mRNA expression of HEXIM1 is modestly increased, which authors did not mention. Is this increase statistically significant? Since HEXIM1 itself is one of P-TEFb's target genes transcribed immediately after P-TEFb activation, this could be a sign of P-TEFb activation although one could expect the increase be greater. Additional comments would be helpful.

Thank you for pointing this out. The modest increase in *Hexim1* mRNA level is reproducible, when *Larp7* is knocked out. It has been observed both *in vivo* (Figure 7B) and *in vitro* (Figure 7D). This phenomenon was also previously observed in HeLa cells.

We added additional text in the revised manuscript as follows:

Line 378-381:

“A previous study has shown that in HeLa cells the activation of P-TEFb increases HEXIM1 mRNA level (Liu, Xiang et al., 2014). We found that *Hexim1* mRNA expression is also moderately increased in *Larp7* knockout cells (Fig. 7B and 7D), consistent with enhanced P-TEFb activity in these cells.”

Referee #3:

In this study, the authors investigate the role of LARP7 and HEXIM1 in regulating neuronal development through modulation of P-TEFb activity. Enhanced P-TEFb

activity is generally thought to promote cell proliferation by increasing the transcriptional output of RNA polymerase II, and under normal conditions most P-TEFb is inhibited by the 7SK snRNP complex containing LARP7 and HEXIM1. Paradoxically, biallelic inactivation of LARP7 in humans' results in Alazami syndrome, a neurodevelopmental disorder that is characterized by growth restriction and cognitive impairment. The authors report that conditional ablation of either *Larp7* or *Hexim1* in the murine brain reduces the size of the hippocampal dentate gyrus during the neonatal period. By combining transcriptomic, epigenomic, and functional analyses, they further propose that heightened P-TEFb activity enhances self-renewal transcriptional programs in transit-amplifying neuronal progenitors, thereby limiting neurogenesis in the developing dentate gyrus. These findings suggest a potential mechanistic explanation linking dysregulated progenitor dynamics with reduced brain size and highlight possible translational applications of clinical-grade P-TEFb inhibitors.

The topic is interesting and the authors address an important question at the interface of transcriptional regulation and neurodevelopment. Many of the experiments are carefully conducted and the conclusions are of potential significance to the field. However, I have some doubts on the presentation of the manuscript and on the conclusions reached by the authors based on these experiments. Additional data should be provided to sustain the discussion. Overall, this study reveals the impact of P-TEFb activity on dentate gyrus neurogenesis in Alazami syndrome mouse models, but the quality of data in this paper is not high and improvements are needed.

Thank you for your positive assessment of our study. We are grateful for your constructive feedback, which has helped us to improve the quality of the manuscript.

Major concerns:

1. The introduction could be enhanced by including recent advances on P-TEFb regulation in neural stem cell fate determination, Furthermore, updating the clinical spectrum of Alazami syndrome with recent findings, particularly regarding its genotype-phenotype heterogeneity, would improve the timeliness and clinical relevance of the discussion.

Thank you for pointing this out. As far as we know, there is no information in the published literature regarding P-TEFb regulation in neural stem cell fate

determination. The present study is the first to use mouse models to investigate the impact of dysregulated P-TEFb in mammalian neurogenesis.

Following your suggestion, in the revised manuscript, we have added a summary of core features of Alazami syndrome patients based on recent literature, compared them with phenotypes in our mouse models (new data, Appendix Figure S8), and added additional text as follows:

Phenotypic Category	Human LARP7 Mutation Patients	Variant 1	Variant 2	Larp7^{fl/fl} ;nestin-Cre Mice	Larp7^{fl/fl} ;Emx1-Cre Mice Hexim1 ;Emx1-Cre Mice
Inheritance	Autosomal recessive ¹	/	/	Autosomal recessive	Autosomal recessive
Height/Body length	Short stature	c.1173T>A (p.Y391*) ²	c.1173T>A (p.Y391*)	Reduced body length (fully penetrant)	No change
		c.1653_1654del (p.G553fs) ²	c.1653_1654del (p.G553fs)		
		c.646+3_646+6del ²	c.646+3_646+6del		
		c.1173T>A (p.Y391*) ²	c.1653_1654del (p.G553fs)		
		c.213_214dup (p.S72fs) ³	c.651_655del (p.(K219fs))		
		c.1070_1073del (p.R357fs) ⁴	c.1070_1073del (p.R357fs)		
		c.1091_1094del (p.K364fs) ⁵	c.1091_1094del (p.K364fs)		
		c.1669-1_1671del ⁶	c.834dup (p.R279fs)		
		c.503_504dup (p.A169fs) ⁷	c.503_504dup (p.A169fs)		
		c.892_895dup (p.S299fs) ⁸	c.1087_1091del (p.H363fs)		
		c.1024_1030dup (p.T344fs) ⁴	c.1024_1030dup (p.T344fs)		
Weight	Low body weight	c.1173T>A (p.Y391*) ²	c.1173T>A (p.Y391*)	Reduced body weight (both sexes)	No change
		c.1653_1654del (p.G553fs) ²	c.1653_1654del (p.G553fs)		
		c.646+3_646+6del ²	c.646+3_646+6del		
		c.1173T>A (p.Y391*) ²	c.1653_1654del (p.G553fs)		
		c.213_214dup (p.S72fs) ³	c.651_655del (p.(K219fs))		
		c.1091_1094del (p.K364fs) ⁵	c.1091_1094del (p.K364fs)		
		c.1669-1_1671del ⁶	c.834dup (p.R279fs)		
		c.503_504dup (p.A169fs) ⁷	c.503_504dup (p.A169fs)		
		c.892_895dup (p.S299fs) ⁸	c.1087_1091del (p.H363fs)		
		c.1024_1030dup (p.T344fs) ⁴	c.1024_1030dup (p.T344fs)		
		Head	Microcephaly		
c.1653_1654del (p.G553fs) ²	c.1653_1654del (p.G553fs)				
c.646+3_646+6del ²	c.646+3_646+6del				
c.1173T>A (p.Y391*) ²	c.1653_1654del (p.G553fs)				
c.213_214dup (p.S72fs) ³	c.651_655del (p.(K219fs))				
c.1070_1073del (p.R357fs) ⁴	c.1070_1073del (p.R357fs)				
c.1091_1094del (p.K364fs) ⁵	c.1091_1094del (p.K364fs)				
c.1669-1_1671del ⁶	c.834dup (p.R279fs)				

		c.503_504dup (p.A169fs) ⁷	c.503_504dup (p.A169fs)		
		c.892_895dup (p.S299fs) ⁸	c.1087_1091del (p.H363fs)		
		c.1024_1030dup (p.T344fs) ⁴	c.1024_1030dup (p.T344fs)		
Central Nervous System	Intellectual disability	c.832A>T (p.K278*) ⁹	c.832A>T (p.K278*)	Impaired spatial learning and memory	Impaired spatial learning and memory
		c.1173T>A (p.Y391*) ²	c.1173T>A (p.Y391*)		
		c.646+5G>C ¹⁰	c.834dup (p.R279fs)		
		c.349del (p.E117fs) ¹¹	c.620_646+25del (p.P208_E216del)		
		c.1653_1654del (p.G553fs) ²	c.1653_1654del (p.G553fs)		
		c.646+3_646+6del ²	c.646+3_646+6del		
		c.1173T>A (p.Y391*) ²	c.1653_1654del (p.G553fs)		
		c.827dup (p.K277fs) ¹⁰	c.827dup (p.K277fs)		
		c.1024_1030dup (p.T344fs) ^{4,10}	c.1024_1030dup (p.T344fs)		
		c.1091_1094del (p.K364fs) ¹⁰	c.1091_1094del (p.K364fs)		
		c.756_757del (p.R253fs) ¹⁰	c.756_757del (p.R253fs)		
		c.503_504dup (p.A169fs) ^{7,10}	c.503_504dup (p.A169fs)		
		c.213_214dup (p.S72fs) ³	c.651_655del (p.(K219fs)		
		c.1070_1073del (p.R357fs) ⁴	c.1070_1073del (p.R357fs)		
		c.1091_1094del (p.K364fs) ⁵	c.1091_1094del (p.K364fs)		
		c.1669-1_1671del ⁶	c.834dup (p.R279fs)		
c.892_895dup (p.S299fs) ⁸	c.1087_1091del (p.H363fs)				
Behavioral Psychiatric Manifestations	Severe anxiety	c.1173T>A (p.Y391*) ²	c.1653_1654del (p.G553fs)	Significantly increased anxiety-related behaviors	Significantly increased anxiety-related behaviors
		c.213_214dup (p.S72fs) ^{3,10}	c.651_655del (p.(K219fs)		
		c.1091_1094del (p.K364fs) ¹⁰	c.1091_1094del (p.K364fs)		
Other Features	/	/	/	Normal lifespan	Normal lifespan

Line 416-421:

“These phenotypes closely mimic the salient features of patients afflicted with Alazami syndrome (Appendix Fig. S8). It is worth noting that the reduction of the body size of *Larp7^{ff};nestin-Cre* mice seems less severe than that in Alazami syndrome patients. One potential explanation is that nestin-Cre only functions in selected organs in mice. In contrast, human patients carry germline inactivating variants of *LARP7*, that is, the function of LARP7 protein is lost in all cells.”

2. In the manuscript, there are inherent differences between the mouse model and patient samples. It is recommended that the authors include an analysis of Alazami syndrome patient samples carrying LARP7 mutations, and compare the findings with those obtained from the mouse model to strengthen the clinical relevance.

We agree with the Reviewer that this would be a very interesting experiment. Regrettably, we do not have access to primary cells derived from Alazami syndrome patients. In addition, this experiment would require reprogramming patient cells into iPS cells, and then using iPS cells to generate neural stem cells or brain organoids. At this stage, we do not have expertise in these complicated techniques.

3. Although the mouse models recapitulate key features of Alazami syndrome, the authors should discuss whether similar transcriptomic changes (e.g., upregulated self-renewal genes) have been observed in human patient-derived iPSCs or organoids, if such data are available.

We agree with the Reviewer that this is an important area to investigate. Unfortunately, as far as we know, there is no such data in the literature. In the revised manuscript, to acknowledge the limitation of the present study, we added additional text in Discussion as follows:

Line 448-451:

“One important unanswered question is whether human neural stem and progenitor cells with *LARP7* deficiency would exhibit similar phenotypic and transcriptomic defects as their murine counterparts. To address this issue, primary cells derived from Alazami syndrome patients could be used to generate iPS cells, and then differentiated into neural stem cells or brain organoids.”

4. For Figure 1G, how many brain sections were analyzed per mouse, and how was consistency in sectioning positions ensured? At present, the staining and quantification shown in Figures 1F and 1G are not sufficient to support the conclusion that the dentate gyrus is reduced in size. Additional types of experimental evidence are required.

In the revised manuscript, experimental details have been added as follows:

Line 505-511:

“To measure hippocampal and dentate gyrus areas, wild-type and mutant brains were sectioned consecutively from the Bregma at 10 μ m intervals. Sections were compared to the Allen Mouse Brain Atlas to select comparable positions in wild-type and mutant brains. For quantification, three sections each from 6 mice of indicated genotypes were selected, and added up to represent the size of the hippocampal or

dentate gyrus area, including *Larp7*-floxed (n=6), *Larp7^{ff}*;nestin-Cre (n=6), *Larp7^{ff}*;Emx1-Cre (n=6), *Hexim1*-floxed (n=6) and *Hexim1^{ff}*;Emx1-Cre (n=6), respectively.”

5. The study mentions that Alazami syndrome patients exhibit cognitive deficits but does not clarify the correspondence between dentate gyrus hypoplasia and specific cognitive functions (e.g., spatial memory, social ability). It is suggested to supplement direct evidence for the association between dentate gyrus neuronal number/functional abnormalities and behavioral deficits (e.g., through neuron-specific rescue experiments).

We thank the Reviewer for pushing us to answer these questions. We used four behavioral tests to measure cognitive deficits in three mouse models (knockout of *Larp7* by nestin-cre and Emx1-cre; knockout of *Hexim1* by Emx1-cre). These mice exhibited similar defects. Because the four behavioral tests reflect different aspects of cognitive functions, we speculate that the overall functionality of the dentate gyrus might be declined.

For genetic rescue experiments, we have generated *Cdk9*-floxed mice carrying two knock-in mutations, D167N and T186A. These mutations would significantly lower the kinase activity of P-TEFb as we previously showed *in vitro*. We crossed these mice with Emx1-Cre mice. However, so far, we haven't been able to obtain homozygotes containing D167N/T186A *Cdk9* (n=12), suggesting that mice might not tolerate this mutant. But we will keep trying. The scheme to generate this knock-in mutant and genotyping results are presented here for your reference:

6. Behavioral quantification: mismatch between Methods and probe-trial claims. in the Behavioral testing / Morris water maze subsection, the Methods state that mice were trained for 5 days and that "time in the target quadrant" and "number of

crossings" during the 5-day training period were used as learning indices. By field standards, training-phase learning is quantified by escape latency/path length across days, whereas time in target quadrant and platform-location crossings are metrics of the probe trial (platform removed). In contrast, the figure legends (e.g., Fig. 1H, Fig. 2I) describe probe-trial measures. This creates a Methods-to-Results inconsistency. Please clarify and, if needed, re-analyze to conform to standard MWM practice:(i) report training curves (escape latency and/or path length) over days with the appropriate repeated-measures analysis; (ii) for the probe trial, report time in target quadrant and platform-location crossings (with chance-level comparison), and avoid training metrics here; (iii) provide control measures (swim speed, visible-platform performance) to exclude motor/visual confounds; (iv) make Methods and figure legends internally consistent (clearly separate training vs probe metrics and specify which are plotted in each panel).

Thank you for pointing this out. The experiment was indeed carried out as the Reviewer described. We apologize for the lack of clarity in the previous manuscript. In the revised manuscript, the description is as follows:

Line 526-537:

“For Morris water maze test, during a 5-day training phase, mice were trained to locate the escape platform (1 cm below the water surface) in a tank filled with water colored with powdered milk (Appendix Fig. S9). For the probe trial, the escape platform was removed on day 7. Each mouse was allowed to swim for 1 min to determine whether the animal remembered the location of the platform. “Latency 1st entrance to platform” is defined by the time taken by the mouse to reach the location of removed platform for the first time. “Platform crossover numbers” is defined by how many times the animal crosses the location of removed platform. These two parameters were regarded as indicators of how well the animal has learned the spatial location of the platform. “Path length” is defined by the overall swimming distance of each mouse. Wild-type and mutant mice exhibited comparable speed as they swam similar overall distance within 1 min. For the probe trial, the time spent in the target quadrant was also documented (Appendix Fig. S9).”

7. The single-cell transcriptomic analyses presented in Figure 3 are not sufficient to directly support the conclusion that "loss of *Larp7* activates P-TEFb and promotes RNA Pol II transcription." More direct experimental evidence is required.

We agree with the Reviewer that the single-cell transcriptomic analysis alone does not provide direct evidence of upregulated transcription. This conclusion is based on the following four lines of evidence, all of which are consistent with higher transcription activity in *Larp7*-null neural stem and progenitor cells. Firstly, the outcome of increased P-TEFb activity is expected to increase RNA polymerase II transcription. We have shown that the mRNA content is increased in *Larp7*-null neural progenitor cells (Figure 3B and 5B). Although mRNA content can be influenced by multiple mechanisms, increased mRNA content per cell is in line with higher transcription activity. Secondly, it is generally accepted that higher transcription activities are correlated with more accessible chromatin. We analyzed the chromatin accessibility by ATAC-seq, and showed that the chromatin is more accessible in *Larp7*-null neural progenitor cells than wild-type cells (Figure 6C). Thirdly, we used CUT&Tag assay measured the density of total as well as elongating RNA polymerase II on annotated genes. The pausing index of RNA polymerase II was decreased (Figure 6D), and more peaks of elongating RNA polymerase II were detected in the gene body regions in *Larp7*-null cells than wild-type cells (Sox2 gene body was used as one of many examples in Figure 6E). Decreased pausing index and more elongating RNA polymerase II in the gene body regions are consistent with higher transcription activity. Fourthly, in the revised manuscript, we also added new data showing that this is also the case in *Hexim1*-null neural progenitor cells (Figure 7Q and 7R). Thus, results from different assays are all in line with higher transcription activity in *Larp7*- and *Hexim1*-null neural progenitor cells.

8. The study reports that LARP7 deficiency increases the self-renewal of nIPCs while reducing their differentiation. However, it remains unclear whether this is due to depletion of the progenitor pool or delayed neuronal differentiation. It is recommended that the authors perform longitudinal analyses at multiple developmental stages, including single-cell RNA sequencing with pseudotime analysis, to clarify the underlying cause.

Thank you for raising this point. Following your suggestion, in the revised manuscript, we carried out additional single-cell RNA sequencing analysis using P14 dentate gyrus. P14 timepoint was chosen because the peak of neurogenesis in the dentate gyrus occurs within 14 days after birth. In the previous manuscript, we showed that many cell cycle and self-renewal genes are upregulated in neural progenitor cells of P7 *Larp7*-null dentate gyrus (Figure 5). The same trend is

observed in P14 *Larp7*-null dentate gyrus (new data, Appendix Figure S5). Marker analyses indicate that progenitor pools are increased, rather than depleted, from P7 to P16 *in vivo* (Figure 2C, 2D, 2G, 2H, 4D, 4E, 4F, 4J). These results support the notion that *Larp7*-null cells are defective in differentiation *in vivo*. Neurosphere assays, *in vitro*, also confirmed the cell-intrinsic nature of defective differentiation (Figure 6F-6I). Importantly, treatment with P-TEFb inhibitor could prevent this defect (Figure 6J and 6K). In the revised manuscript, we carried out additional experiments using *Hexim1*-null neural progenitor cells. We found that *Hexim1*-null cells exhibit similar behavior as *Larp7*-null cells (new data, Figure 7N-7T; please see below for the response to your Major point #10). Taken together, these results support defective differentiation as the explanation, at least partly, for reduced dentate gyrus size.

The data on pseudotime analysis of P7 dentate gyri (new data, Appendix Figure S2) is reproduced here for your reference:

The data on pseudotime analysis of P14 dentate gyri (new data, Appendix Figure S5) is reproduced here for your reference:

These results suggest that the differentiation trajectories in wild-type and mutant mice were similar.

9. To better assess the effects of *Larp7* deficiency (*Larp7^{f/f}; nestin-Cre*) on neurogenesis, the inclusion of *in vitro* assays and pharmacological inhibition experiments is recommended.

Following your suggestion, we carried out additional experiments using neural

stem and progenitor cells derived from *Larp7^{ff};nestin-Cre* dentate gyrus (new data, Appendix Fig. S7). In the revised manuscript, we added new figures and additional text as follows:

In Results, Line 351-354:

“To corroborate results observed in *Larp7^{ff};Emx1-Cre* neural and progenitor cells, the dentate gyrus was microdissected from P1 *Larp7^{ff};nestin-Cre* mice, and used to culture neural stem and progenitor cells *in vitro*. Similarly, these cells also exhibited enhanced self-renewal and compromised differentiation *in vitro* (Appendix Fig. S7).”

The new data (Appendix Fig. S7) is reproduced here for your reference:

In Appendix Figure S7:

“(A) Phase contrast images of self-renewing neurospheres, derived from the dentate gyrus of P1 wild-type (*Larp7^{ff}*) and mutant (*Larp7^{ff};nestin-Cre*) mice, respectively. Representative results were shown (n=3). Error bars, mean ± SEM. P-values, two-sided unpaired student’s t test. Scale bars: 100 μm.

(B) The self-renewal capacity of wild-type (*Larp7^{ff}*) and mutant (*Larp7^{ff};nestin-Cre*) neurospheres under sub-optimal self-renewal culture condition. Representative results were shown (n=3). The percentage of neurospheres with a branched morphology, indicative of differentiation, was quantified. Error bars, mean ± SEM. P-values, two-sided unpaired student’s t test. Scale bars: 100 μm.

(C) Differentiation of neurospheres for 6 days with or without P-TEFb inhibitor (flavopiridol). DMSO is the solvent of flavopiridol (Flavo). Representative images were shown (left panel) (n=3). GFAP immunofluorescence was used to quantify

differentiated cells (right panel). Error bars, mean \pm SEM. P-values, two-sided unpaired student's t test. Scale bars: 100 μ m.”

10. Phenotypic analysis of the *Hexim1* knockout model in Figure 7 is relatively brief. It is recommended to supplement behavioral data (e.g., spatial memory, anxiety levels) and quantitative results of nIPC proliferation/differentiation in *Hexim1* knockout mice to form a complete comparison with the *LARP7* knockout model.

Following your suggestion, we carried out additional experiments with *Hexim1* knockout mice. In the revised manuscript (new data, Figure 7H-7T), we show that *Hexim1* knockout shares similar features with *Larp7* knockout: (1) both *Hexim1* and *Larp7* knockout mice have hypoplastic dentate gyrus, and exhibit similar deficits in behavioral tests (new data, Figure 7H-7M); (2) both *Hexim1* and *Larp7* knockout neural spheres exhibit enhanced self-renewal ability and compromised differentiation (new data, Figure 7N-7T). We added additional text in the revised manuscript as follows:

In Results, Line 387-404:

“In addition, these mice exhibited cognitive deficits in behavioral tests (Fig. 7I-7M), similar to *Larp7*^{ff};nestin-Cre and *Larp7*^{ff};Emx1-Cre mice. Neural stem and progenitor cells were derived from microdissected P1 *Hexim1*^{ff};Emx1-Cre dentate gyrus. Under optimal self-renewal culture condition, both wild-type and *Hexim1* knockout neurospheres contain densely packed cells with a smooth perimeter (Fig. 7N). We confirmed that the expression levels of 7sk snRNA, Cdk9, and *Larp7* were comparable in wild-type and *Hexim1* knockout neural stem and progenitor cells. In contrast, in *Larp7* knockout cells, 7sk snRNA expression was markedly decreased, whereas the expression levels of Cdk9 and *Hexim1* were unaltered (Fig. 7O and 7P). Consistent with elevated P-TEFb activity, CUT&Tag assay revealed that the pausing index of RNA polymerase II was decreased (Fig. 7Q), and more active RNA polymerase II peaks were detected in the gene body regions in *Hexim1* knockout cells than wild-type cells (Fig. 7R). Under suboptimal culture condition, the morphology of *Hexim1* knockout neurospheres was minimally affected, whereas wild-type neurospheres started to exhibit multiple branch-like structures extending from the core (Fig. 7S). Compared to wild-type cells, differentiation of *Hexim1* knockout cells was compromised, and this defect could be alleviated by P-TEFb inhibitor (Fig. 7T). Thus, *Hexim1* knockout and *Larp7* knockout share many phenotypic similarities

in mice as well as in neural stem and progenitor cells. We conclude that the disruption of the 7SK snRNP complex, thus the activation of P-TEFb, causes hypoplastic dentate gyrus.”

The new data (Figure 7H-7T) is also reproduced here for your reference:

Line: 1164-1207

In Figure 7 legends:

“(I) Morris water maze test. During the probe trial (platform removed), wild-type (WT; n=10) and knockout (*Hexim1^{ff};Emx1-Cre*; n=8) mice were tested. The number of platform crossing, latency to find the location of removed platform, and the total path

length were documented by SMART video tracking system. Error bars, mean \pm SEM. P-values, two-sided unpaired student's t test.

(J) Three-chamber sociability test. Wild-type (WT; n=10) and knockout (*Hexim1^{ff}*;Emx1-Cre; n=8) mice were tested. Total sniffing time spent towards empty chamber (E) or stranger 1 (S1) was documented by SMART video tracking system. Error bars, mean \pm SEM. P-values, two-sided unpaired student's t test.

(K) Three-chamber social novelty test. Wild-type (WT, n=10) and knockout (*Hexim1^{ff}*;Emx1-Cre; n=8) mice were tested. Total sniffing time spent toward stranger 1 (S1) or stranger 2 (S2) were documented by SMART video tracking system. Error bars, mean \pm SEM. P-values, two-sided unpaired student's t test.

(L) Open field test. Wild-type (WT; n=10) and knockout (*Hexim1^{ff}*;Emx1-Cre; n=8) mice were tested. The time spent in the indicated areas, including Center, Non-periphery and Periphery, and total distance were documented by SMART video tracking system. Error bars, mean \pm SEM. P-values, two-sided unpaired student's t test.

(M) Elevated plus maze test. Wild-type (WT; n=10) and knockout (*Hexim1^{ff}*;Emx1-Cre; n=8) mice were tested. The time spent in the indicated areas, including Center, Open arms, and Closed arms, was documented by SMART video tracking system. Error bars, mean \pm SEM. P-values, two-sided unpaired student's t test.

(N) Phase contrast images of self-renewing neurospheres, derived from the dentate gyrus of P1 wild-type (*Hexim1^{ff}*) and knockout (*Hexim1^{ff}*;Emx1-Cre) mice, respectively. Representative results were shown (n=3). Error bars, mean \pm SEM. P-values, two-sided unpaired student's t test. Scale bars: 100 μ m.

(O) Protein blot analyses of indicated proteins from wild-type and mutant neurospheres. Representative results were shown (n=3). Error bars, mean \pm SEM. P-values, two-sided unpaired student's t test.

(P) RNA blot analyses of indicated RNA from wild-type and mutant neurospheres. Representative results were shown (n=3). Error bars, mean \pm SEM. P-values, two-sided unpaired student's t test.

(Q) The pausing index (PI) in wild-type and knockout (*Hexim1^{ff}*;Emx1-Cre) neural stem and progenitor cells. Statistical significance was assessed using the two-sided Wilcoxon rank-sum test.

(R) Browser shot depicting Pol II peaks within Sox2 gene in wild-type and knockout (*Hexim1^{fl/fl};Emx1-Cre*) neural stem and progenitor cells.

(S) The self-renewal capacity of wild-type and knockout (*Hexim1^{fl/fl};Emx1-Cre*) neurospheres under sub-optimal self-renewal culture condition. Representative results were shown (n=4). The percentage of neurospheres with a branched morphology, indicative of differentiation, was quantified. Error bars, mean \pm SEM. P-values, two-sided unpaired student's t test. Scale bars: 100 μ m.

(T) Differentiation of neurospheres for 6 days with or without P-TEFb inhibitor (flavopiridol). DMSO is the solvent of flavopiridol (Flavo). Representative images were shown (left panel) (n=3). GFAP immunofluorescence was used to quantify differentiated cells (right panel). Error bars, mean \pm SEM. P-values, two-sided unpaired student's t test. Scale bars: 100 μ m.”

11. The manuscript lacks sufficient mechanistic exploration, including predictions and validation of downstream target genes, as well as rescue experiments.

In the revised manuscript, we carried out additional experiments to determine the genomic distribution of Sox2 transcription factor in wild-type and *Larp7*-null cells by CUT&Tag assay (new data, Appendix Figure S6). Sox2 is chosen because it is a core self-renewal transcription factor for neural progenitor cells, and also because differentially expressed genes in *Larp7*-null cells are predicted by bioinformatics analysis to be target genes of the Sox2 transcriptional program (Figure 3H and 5I). The new data is also reproduced here for your reference:

In Appendix Figure S6 legend:

“Heat-map representation of Sox2 CUT&Tag signals in wild-type and *Larp7^{fl/fl};Emx1-*

Cre neurospheres. TSS, transcription start site.”

Minor comments:

1. The author should provide the statistical data of Figure 1B, Figure 6A, Figure 6B, Figure 6F. In manuscript, in figure 2, the behavioral experiments related to mice, such as learning and memory, anxiety, at least two methods of each of behavioral evidence are required to be provided.

In the revised manuscript, we have provided the statistical data for these figures. Regrettably, due to limited resources in our animal facility, we do not have access to other equipment/methods beyond what has been presented in the manuscript.

2. Figure 1B Western blot lacks quantitative analysis of protein expression.

Quantification has been added in the revised manuscript.

3. The reference formatting contains errors: there should be a space between the preceding text and the parentheses, and multiple references within the same parentheses should be separated by semicolons.

The reference formatting has been corrected.

4. In Figure S2, the single-cell cluster identities are not clearly defined. Do these clusters include cell types beyond astrocytes, RGLs, nIPCs, and neuroblasts?

As expected by the Reviewer, other cell types were also identified in our single-cell analyses of P7 dentate gyrus. We did not include these cell types because we focused on dissecting the defects at the early stage of differentiation in the present study.

In the revised manuscript, we have presented all cell types identified in P7 and P14 wild-type and *Larp7* knockout dentate gyri, respectively (new data, Appendix Figure S2 for P7 timepoint, and Appendix Figure S5 for P14 timepoint). The new data is also reproduced here for your reference:

Appendix Figure S2:

In Appendix Figure S2 legend:

“(A) UMAP presentation of all cell types, identified by single-cell sequencing, from microdissected dentate gyrus from P7 wild-type (WT), *Larp7^{ff};nestin-Cre*, and *Larp7^{ff};Emx1-Cre* mutant mice.”

Appendix Figure S5:

In Appendix Figure S5 legend:

“(A) UMAP presentation of all cell types, identified by single-cell sequencing, from microdissected dentate gyrus from P14 wild-type (WT), and *Larp7^{ff};Emx1-Cre* mutant mice.”

5. Using multiple markers to independently identify NSCs, nIPCs, and neurons would strengthen the reliability of the results.

In the revised manuscript, we have clarified how cell clusters are defined as follows:

In Results, Line 206-208

“Ten major cell clusters were identified by unbiased clustering (Seurat), and their identities were assigned using previously defined markers (Appendix Fig. S2A and S2B) (Hochgerner et al., 2018).”

The new data is also reproduced here for your reference:

Appendix Figure S2:

In Appendix Figure S2 legends:

“(B) Single-cell expression of enriched marker genes for each cell group identified by unbiased clustering (Seurat). Cluster-specific, differentially expressed genes (top 5 per cluster) were shown. Genes in red are established markers by previous studies. Annotated cell types are grouped by columns, and genes are organized by their associated cell types. Scale bars denote the expression level of indicated genes per cell.”

Appendix Figure S5:

In Appendix Figure S5 legends:

“(B) Single-cell expression of enriched marker genes for each cell group identified by unbiased clustering (Seurat). Cluster-specific, differentially expressed genes (top 5 per cluster) were shown. Genes in red are established markers by previous studies. Annotated cell types are grouped by columns, and genes are organized by their associated cell types. Scale bars denote the expression level of indicated genes per cell.”

6. The ATAC-seq peak plot in Figure 6C lacks a vertical axis scale, which should be added to facilitate interpretation.

The description of y-axis has been added in the Figure legend for Fig. 6C in the revised manuscript as follows:

Line 1119-1121:

“The y-axis represents the average normalized signal density per genomic position, ranging from -2,000 bp to +2,000 bp from the TSS.”

7. Figure 7G lacks corresponding statistical analyses.

Statistical analyses have been added in Figure 7H in the revised manuscript.

8. The conclusion "We conclude that the disruption of the 7SK snRNP complex, thus the activation of P-TEFb, causes hypoplastic dentate gyrus" is not fully supported. In the *Hexim1*^{f/f};*Emx1*-Cre mouse model, only dentate gyrus size was assessed, other functional or structural readouts were not examined, and the potential effects of *Hexim2* deletion were not investigated.

Following your suggestion, we have carried out additional experiments with *Hexim1* knockout mice (new data, Figure 7H-7T in the revised manuscript). Please refer to our response to your “Major points, #10” above. These new data demonstrate that *Hexim1* and *Larp7* knockout mice and cells share phenotypic similarities. Thus, we did not investigate the effect of *Hexim2* deletion.

9. The statement in the abstract "Multi-omics and functional analyses reveal ..." is misleading. Although both transcriptomic and epigenomic analyses were performed, the datasets are presented separately without integration, so describing the study as a "multi-omics analysis" is not appropriate.

Thank you for pointing this out. The text has been changed to “Functional analyses” (Line 21).

10. The statement in the abstract "Here, we report that conditional ablation of either *Larp7* or *Hexim1* in the murine brain reduces the size and function of the hippocampal dentate gyrus during the neonatal period" is imprecise. The manuscript presents evidence only for *Hexim1* deletion affecting DG size, without evaluating neuronal function.

Following your suggestion, we have carried out additional experiments with *Hexim1* knockout mice (new data, Figure 7H-7T in the revised manuscript). Please refer to our response to your “Major points, #10” above.

Dear Prof. Li,

Thank you for the submission of your revised manuscript (EMBOJ-2025-122043R) to The EMBO Journal for our consideration, and for your patience during re-review. Your manuscript has now been seen by the three original referees who had previously assessed the original version of the study, and we have received their comments, which are appended below.

As you will see, all three referees are largely satisfied with the revision and recognize that the revised manuscript is substantially improved and the majority of the initially raised concerns have been adequately addressed. There are only two remaining points for minor improvements (from referees #1 and #2) that can be addressed textually in a final version of the manuscript. Please review the points of the referees below carefully and address them completely in a final revision. In addition, I kindly request you submit a point-by-point response to these remaining points with detailed explanations and a description of any changes to the manuscript. Please note that the referees' concerns must be sufficiently addressed both in the revised manuscript and the point-by-point response.

From the editorial side, there are also a few changes we need you to make in the final version of the manuscript, before we can move forward with its further processing at The EMBO Journal:

- Please change heading "Data and materials availability" to "Data availability".
- Please make sure that all datasets generated in this study (RNA sequencing, ATAC sequencing, CUT&Tag data) are deposited in appropriate repositories. All datasets must be publicly available at the time of publication and listed in the Data availability section of the final version of the manuscript. For each dataset, the repository, accession ID, and specific URL should be provided in this section. The links and tokens for confidential referee access can now be removed from the Data availability statement.
- Please also incorporate the code availability information and link in the revised Data availability section.
- Please change heading "Declaration of interests" to "Disclosure and competing interests statement".
- Please change heading "Materials and Methods" to "Methods".
- Please note that the institutional e-mail addresses of all co-corresponding authors must be provided in the online system and also on the title page of the revised manuscript. No institutional e-mail address has been provided for the co-corresponding author Shanling Liu.
- We also note that no ORCID IDs have been provided for the co-corresponding authors Shanling Liu and Xue Xiao. According to our journal's policy, an ORCID ID is mandatory for each co-corresponding author.
- We noticed that your manuscript has an unusually high number of "co-first" authors (4 authors). We kindly suggest you carefully review the actual contributions of each author (which will have to be specified for each co-author using the CRediT system - please see relevant point below) and reduce the number of co-first authors as appropriate to better reflect their actual contributions to the study and the manuscript.
- The author contributions statement should be removed from the manuscript file. Instead, we use CRediT to specify the contributions of each author in the journal submission system. Please feel free to use the free text box to provide more detailed descriptions during submission. See also our guide to authors for more information: <https://link.springer.com/partners/embo-press/editorial-policies#Authorship>.
- Please note that the References format needs to be updated to the style used by EMBO Press. In particular, "et al." must be used after the names of the first 10 co-authors of each citation. Please check our author guidelines for more information on our format: <https://link.springer.com/journal/44318/submission-guidelines#cms-Reference-guidelines>.
- Please note that EMBO press papers are accompanied online by:
 - A) a short (2 sentences) summary of the findings and their significance,
 - B) 2-5 short bullet points highlighting the key results, and
 - C) a synopsis image in .jpg or .png format that is exactly 550 pixels wide and 300-600 pixels high (the height is variable). Please note that all text needs to be legible at the final size.Please upload this information along with your revised manuscript (the text for A and B should be provided in a separate Word file).
- Thank you for uploading the Source Data for the Figures of your manuscript. We kindly request you re-organize the Source Data and re-upload them in a single ZIP folder per main Figure (named, for example, "Source Data for Figure 1.zip" etc.), with

clearly labeled subfolders for the individual Figure panels. The Source Data for the Appendix Figures can remain in a single ZIP folder, as they already are.

- During our standard Figure integrity checks, our team detected cell reuse between Figure 7S and Appendix Figure 7B - this reuse is not listed in the Figure legend. Please check these Figures carefully and correct them or detail the reuse explicitly in the Figure legends, if it is intentional and justified by the experimental setup. In any case, please clarify in your point-by-point response or in a cover letter.

- Furthermore, we noticed during our routine image checks that the blot images across the Figure set appear pixelated under analysis. This is a common result of converting original 16-bit TIFF images to RGB format for publication, and while not a cause for concern, it can sometimes give the impression of image alteration to critical readers. To resolve this, please upload the blot images at a higher resolution using the original captured sample 16-Bit TIFF. Please also update the source data to reflect this. This will enable us to confirm the integrity of the complete Figure set and enhance transparency for readers.

- Our data editors have checked your Figures and their legends, and raised the following queries. Please address all points below completely in your revised manuscript (all changes should be highlighted or "tracked"):

1. Please provide the exact p-values in the legends of Figures 1B, E, G, H, I, J, K; 3B, C; 4B, M, O; 5B, C; 6B, D; 7H, O, P, Q.
2. Please indicate the statistical test used for data analysis in the legends of Figures 3G, H; 5H, I; 6I, 7D.
3. Please note that information related to "n" is missing in the legends of Figures 1E, 2C, D, E, F, G, H, I; 3B, C; 4D, E, F, G, H, I, J, K, L, N; 5B, C; 6D, 7C, D, E, Q.
4. Please note that the error bars are not defined in the legends of Figures 6D, 7D.

- The manuscript sections need to be named and ordered as follows: Title page - Abstract - Introduction - Results - Discussion - Methods - Data Availability - Acknowledgements - Disclosure and Competing Interests Statement - References - Figure Legends - main Tables (if there are any) - Expanded View Figure Legends.

- Please also note that as part of the EMBO Press transparent editorial process, The EMBO Journal publishes online a Peer Review File along with each accepted manuscript. This File will be published in conjunction with your paper and will include the referee reports, your point-by-point responses and all pertinent correspondence relating to the manuscript. Your Author's Checklist will also be published at the end of the Peer Review File. Please let us know in case you want to remove any data or figures from your point-by-point responses before they are published as part of the Peer Review File. Retaining unpublished data in the Peer Review File means that these count as published and that the Peer Review File would need to be referenced in future publications. Please let the editorial office know in case you want to remove any data from this file (contact@embojournal.org).

We look forward to seeing a final version of your manuscript as soon as possible. Please let us know if you have any questions and use this link to submit your revision: Link Unavailable.

Best regards,

Ioannis

Referee #1:

I thank the authors for their attention to all the points raised in the first review. The only pending issue I see is that the explanation the authors provide to point #1 is inadequate. We know that primary microcephaly is not primarily caused by DNA damage-induced cell loss, but rather by the imbalance between symmetric and asymmetric cell division. What the authors propose is counterintuitive to this model and thus deserves careful explanation rather than simply blaming the difference on DNA damage and apoptosis as they did in their response letter. This must be addressed before the paper is accepted.

Referee #2:

In the revised manuscript, the authors sufficiently and adequately addressed the points raised by the reviewers. I have only a minor issue. Regarding my original points #2 and #5, the data indicating that HEXIM1 mRNA was increased in LARP7 fl/fl cell could indeed be direct evidence of P-TEFb-activation since it was reported that HEXIM1 is one of the immediate early genes transcribed after P-TEFb-activation (other data the authors mentioned in the rebuttal are rather circumstantial and indirect evidence). At the same time, this could also be one of the compensational mechanisms to limit the P-TEFb activity although the upregulated HEXIM1 probably cannot inhibit P-TEFb without LARP7. Perhaps, these topics could be included in Discussion?

Referee #3:

The work is of high quality and has important clinical implications not only in terms of explaining the pathogenicity of Alzami syndrome but also potentially age-related degeneration of the dentate gyrus. The topic is interesting and the authors address an important question at the interface of transcriptional regulation and neurodevelopment. I have carefully read the revised version, the authors have made revisions to the manuscript, many of the experiments are carefully conducted and the conclusions are of potential significance to the field. The questions I concerned have been addressed and performed additional experiments for further support. I have no more questions.

We are truly grateful to the anonymous Reviewers for their insightful suggestions and constructive feedback, which have helped improve this manuscript.

Referee #1:

I thank the authors for their attention to all the points raised in the first review. The only pending issue I see is that the explanation the authors provide to point #1 is inadequate. We know that primary microcephaly is not primarily caused by DNA damage-induced cell loss, but rather by the imbalance between symmetric and asymmetric cell division. What the authors propose is counterintuitive to this model and thus deserves careful explanation rather than simply blaming the difference on DNA damage and apoptosis as they did in their response letter. This must be addressed before the paper is accepted.

Following your suggestion, we added additional text in the revised manuscript as follows:

In Discussion, Line:422-428

“Of note, one prominent feature of Alazami syndrome patients is microcephaly. Damaging variants in primary microcephaly genes, such as those encoding centrosome proteins, DNA replication and repair factors, can cause the imbalance between symmetric and asymmetric division of neural progenitor cells. This defect is proposed to underlie the etiology of primary microcephaly (Phan and Holland, 2021). In a broad sense, our results are consistent with this notion in that *Larp7* deficiency also causes unbalanced differentiation of neural progenitor cells.”

Referee #2:

In the revised manuscript, the authors sufficiently and adequately addressed the points raised by the reviewers. I have only a minor issue. Regarding my original points #2 and #5, the data indicating that HEXIM1 mRNA was increased in LARP7 fl/fl cell could indeed be direct evidence of P-TEFb-activation since it was reported that HEXIM1 is one of the immediate early genes transcribed after P-TEFb-activation (other data the authors mentioned in the rebuttal are rather circumstantial and indirect evidence). At the same time, this could also be one of the compensational mechanisms to limit the P-TEFb activity although the upregulated HEXIM1 probably cannot inhibit P-TEFb without LARP7. Perhaps, these topics could be included in Discussion?

Following your suggestion, we added additional text in the revised manuscript as follows:

In Discussion, Line:482-484

“Even less known is how cells limit enhanced P-TEFb activity *in vivo*. We find that *Hexim1* is transcriptionally upregulated in *Larp7* knockout neural progenitor cells, potentially reflecting a cellular compensational mechanism.”

Referee #3:

The work is of high quality and has important clinical implications not only in terms of explaining the pathogenicity of Alzami syndrome but also potentially age-related degeneration of the dentate gyrus. The topic is interesting and the authors address an important question at the interface of transcriptional regulation and neurodevelopment. I have carefully read the revised version, the authors have made revisions to the manuscript, many of the experiments are carefully conducted and the conclusions are of potential significance to the field. The questions I concerned have been addressed and performed additional experiments for further support. I have no more questions.

We sincerely thank you for your effort and time invested in our manuscript.

Dear Prof. Li,

Congratulations on an excellent manuscript! I am very pleased to inform you that it has been accepted for publication in The EMBO Journal. Thank you for comprehensively addressing the initially raised referee concerns and our editorial requests for corrections and other changes.

You may qualify for financial assistance for your publication charges - either via a Springer Nature fully open access agreement or an EMBO initiative. Check your eligibility: <https://link.springer.com/journal/44318/how-to-publish-with-us>

If you have any questions, please do not hesitate to contact the Editorial Office. Thank you for your contribution to The EMBO Journal. Working with you has been a pleasure.

Best regards,

Ioannis

Please note that it is The EMBO Journal policy for the transcript of the editorial process (containing referee reports and your response letters) to be published as an online supplement to each paper. If you should prefer removal of any referee-only figures included in the point-by-point response(s), e.g. because they may still be used for future publication or because they have been reproduced from published work by others, please do let us know immediately via response email.

More information is available here: <https://link.springer.com/partners/embo-press/editorial-policies#Peer%20review>